# The biochemical mechanism of Rho GTPase membrane binding, activation and retention in activity patterning

Michael C Armstrong[1,10], Yannic R Weiß[1,10], Lila E Hoachlander-Hobby[2,3,4,10], Ankit A Roy [1,9,10], Ilaria Visco[1], Alison Moe [2,3,4,7], Adriana E Golding[2,3,4,8], Scott D Hansen [5,6], William M Bement [3,4✉] & Peter Bieling [1✉]

## Abstract

**Rho GTPases form plasma membrane-associated patterns that control the cytoskeleton during cell division, morphogenesis, migration, and wound repair. Their patterning involves transitions between inactive cytosolic and active membrane-bound states, regulated by guanine nucleotide exchange factors (GEFs), GTPase-activating proteins (GAPs), and guanine nucleotide dissociation inhibitors (GDIs). However, the relationships between these transitions and role of different regulators remain unclear. We developed a novel reconstitution approach to study Rho GTPase patterning with all major GTPase regulators in a biochemically defined system. We show that Rho GTPase dissociation from RhoGDI is rate-limiting for its membrane association. Rho GTPase activation occurs after membrane insertion, which is unaffected by GEF activity. Once activated, Rho GTPases are retained at the membrane through effector interactions, essential for their enrichment at activation sites. Thus, high cytosolic levels of RhoGDI-bound GTPases ensure a constant supply of inactive GTPases for the membrane, where GEF-mediated activation and effector binding stabilize them. These results delineate the route by which Rho GTPase patterns are established and define stage-dependent roles of its regulators.**

**Keywords** Rho GTPases; Small G Proteins; Cell Polarity; Membrane Signaling; Self-organization
**Subject Categories** Cell Adhesion, Polarity & Cytoskeleton; Membranes & Trafficking; Signal Transduction

## Introduction

The Rho GTPases—including the founding members Rho, Rac and Cdc42—are small, GTP-binding proteins essential for control of cell shape changes in processes such as cell division, cell migration, morphogenesis, and wound repair (Etienne-Manneville and Hall, 2002; Hodge and Ridley, 2016). These proteins organize into spatio-temporal activity patterns at the plasma membrane (PM), which locally control morphogenic processes powered by the actin cytoskeleton (Bement et al, 2024; Graziano and Weiner, 2014; Sit and Manser, 2011). Their signaling function depends on two complementary cycles undergone by the Rho GTPases, one enzymatic and one spatial (Hakoshima et al, 2003; Vartak and Bastiaens, 2010). In the enzymatic cycle, the GTPases catalytically transition between active, GTP-bound and inactive-GDP bound states and back again via GTP hydrolysis and exchange. In addition, the GTPases reversibly shuttle between the plasma membrane, where they exert their signaling function, and the cytosol, where they are considered to be inert. Membrane shuttling is not merely an equilibrium phenomenon; rather, it is closely tied to the GTPase's activity state, resulting in an energy-driven flux of GTPases between the membrane and the cytosol. This regulated, energy-driven flux between the two compartments is essential for GTPase patterning and a hallmark of what has been defined as a "spatial cycle." (Vartak and Bastiaens, 2010).

The two fundamental cycles of Rho GTPases arise from their regulated interaction with three core classes of regulators (Müller et al, 2020): (i) Guanine nucleotide exchange factors (GEFs), which activate Rho GTPases by stimulating exchange of GDP for GTP at the plasma membrane (Rossman et al, 2005), (ii) GTPase activating proteins (GAPs), which inactivate GTPases by stimulating GTP hydrolysis (Tcherkezian and Lamarche-Vane, 2007) and (iii) guanine nucleotide dissociation inhibitors (GDIs), which solubilize Rho GTPases (i.e. extract them from the PM) to create a large

[1]Department of Systemic Cell Biology, Max Planck Institute of Molecular Physiology, Dortmund, Germany. [2]Graduate Program in Cellular and Molecular Biology, University of Wisconsin-Madison, Madison, WI, USA. [3]Department of Integrative Biology, University of Wisconsin-Madison, Madison, WI, USA. [4]Center for Quantitative Cell Imaging, University of Wisconsin-Madison, Madison, WI, USA. [5]Department of Chemistry and Biochemistry, University of Oregon, Eugene, OR, USA. [6]Institute of Molecular Biology, University of Oregon, Eugene, OR, USA. [7]Present address: Department of Medicine, Medical College of Wisconsin, Milwaukee, WI 53226, USA. [8]Present address: Eunice Kennedy Shriver National Institute of Child Health and Human Development, National Institutes of Health, Bethesda, MD 20892, USA. [9]Present address: Department of Biological Sciences, Columbia University, New York, NY, USA. [10]These authors contributed equally: Michael C Armstrong, Yannic R Weiß, Lila E Hoachlander-Hobby, Ankit A Roy. ✉E-mail: wmbement@wisc.edu; peter.bieling@mpi-dortmund.mpg.de

reservoir of RhoGDI:Rho GTPase complexes in the cytosol (Garcia-Mata et al, 2011). Active, membrane-bound Rho GTPases then interact with effector proteins that control actin and myosin-2 assembly and dynamics at the plasma membrane (Bishop and Hall, 2000).

The fundamental participants in Rho GTPase regulation have been recognized for many years, as has the existence of the two cycles. Surprisingly, however, some fundamental features of the regulators and cycles remain mysterious. One of the most significant of these is the temporal relationship between the release of a given Rho by RhoGDI, the delivery of the GTPase to the PM, and the activation of the GTPase. In principle, the GDI could release the inactive GTPase into the cytosol whereupon it could insert into the plasma membrane and become activated by membrane-localized GEFs (Robbe et al, 2003). Alternatively, membrane-localized GEFs could interact with cytosolic GDI-GTPase complexes, displace the GDIs from the inactive GTPase, and simultaneously recruit the GTPase to and activate it at the PM (Boulter and Garcia-Mata, 2010; Fauré et al, 1999; Ugolev et al, 2008). The former model is consistent with imaging studies which suggest that activation of Rac occurs after its association with the PM (Das et al, 2015). The latter is consistent with the long-held idea that Rho GTPases are immediately extracted from the plasma membrane by RhoGDI following inactivation (Isomura et al, 1991), meaning that activation would have to occur concomitantly with recruitment to prevent immediate extraction.

Standing firmly in the way of both models is the high affinity of RhoGDI for inactive GTPases. That is, studies with different GTPases using different methods indicate that the $K_D$ is in the very low nM to high pM range (Gosser et al, 1997; Medina Gomez et al, 2024; Tnimov et al, 2012). In the face of such a tight interaction, and the high concentration of RhoGDI in most cells (~500–1500 nM (Beck et al, 2011; Michaelson et al, 2001; Wühr et al, 2014)) it is difficult to imagine either the efficient delivery of inactive GTPases to the PM following spontaneous dissociation of the GTPase from the GDI, or the displacement of the GDI from the GTPase by GEFs. Indeed, in vitro measurements indicate that promotion of GTP exchange by GEFs is essentially nonexistent in the presence of GDI (Kikuchi et al, 1992; Ozaki et al, 1996; Yaku et al, 1994). It has been suggested that this "GDI problem" may be ameliorated in vivo by RhoGDI phosphorylation (DerMardirossian et al, 2006, 2004), by the presence of GDI displacement factors (GDFs) and/or phospholipids which may weaken the GDI-GTPase interaction, thereby promoting release of the inactive GTPase into the cytosol, displacement of the GDI from the GTPase by GEFs, or both (Chuang et al, 1993; DerMardirossian and Bokoch, 2005; Fauré et al, 1999; Robbe et al, 2003). However, this crucial aspect of the GTPase cycle remains poorly understood.

The confusion surrounding the route by which Rho GTPases become activated at membranes means we also have a limited understanding of another fundamental feature of Rho GTPase signaling, namely, how Rho GTPase delivery and turnover are coupled to their spatial patterning. Early work indicated that Rho GTPases translocate from the cytosol to membranes in response to activating signals (Kranenburg et al, 1997; Michaelson et al, 2001; Philips et al, 1993). This suggests that partitioning between their membrane-bound and soluble, RhoGDI-bound forms in vivo is influenced by their activity state. However, we currently have no clear understanding of the relationships between active and total Rho GTPases at cellular membranes and how, exactly, their interconversion relates to the formation and maintenance of membrane-associated Rho GTPase patterns. Here we show that Rho GTPases accumulate at sites where they are activated inside cells. To mechanistically explain this phenomenon, we developed novel in vitro reconstitution approaches to study Rho GTPase signaling on model membrane systems that allowed us to explore the route by which Rho GTPases arrive and are activated at membranes and how this is coupled to pattern formation. We combine direct visualization of Cdc42 recruitment and activation in vitro with measurements of its membrane turnover both in vitro and in living cells. Our results indicate that (1) Rho GTPase arrival at the membrane is kinetically limited by spontaneous dissociation of their complexes with RhoGDI; (2) the arrival rate of the Cdc42 at the membrane is unaffected by GEF activity; (3) Cdc42 activation by GEFs occurs after arrival at the membrane; (4) the activation of the Cdc42 is responsible for its accumulation at the membrane; (5) interaction with effectors is a critical feature in the formation of Cdc42 patterns and the enrichment of Cdc42 at sites of its activation.

## Results

### Concentration of both active and total Cdc42 around single cell wounds

As an initial step toward understanding the relationship between Rho GTPase activation and patterning, we compared the cortical accumulation of active Cdc42 and total Cdc42 during cell repair in *Xenopus* oocytes which, like other cell types (Ammendolia et al, 2021), entails local activation of small GTPases including Cdc42 (Abreu-Blanco et al, 2014; Benink and Bement, 2005; Xu et al, 2021). We used mRNA encoding wGBD (to detect active Cdc42 (Benink and Bement, 2005)) and internally tagged-Cdc42 (IT-Cdc42) to report on total Cdc42 (Golding et al, 2019) (Fig. 1A). In this system, laser wounding of the cell results in rapid activation of Cdc42 in a circular zone around the wound site (Benink and Bement, 2005); following formation, the zone closes inward in conjunction with cortical flow powered by actomyosin (Mandato and Bement, 2001). Consistent with previous results (Golding et al, 2019), both active and total Cdc42 accumulate around wounds, partially, but not completely overlapping, such that total Cdc42 extended closer to the wound edge than active Cdc42 (Fig. 1B,C). The extent of active and total Cdc42 enrichment around the wound was measured (Fig. 1C–E; Movie EV1; see "Methods") 40 s after wounding ("pre-zone, P"), before a clear Cdc42 zone developed and at 60 s after wounding, at which point a zone was visible ("zone, Z"). These times were selected because they precede the onset of cortical flow, and thus let us exclude potential contributions of myosin-powered advection to Cdc42 patterning. We quantified protein enrichment around the wound by the "patterning index"—the ratio of signal within the nascent zone vs the signal 30 μm distal to the wound (Fig. 1F). This metric clearly revealed an increase in patterning of active Cdc42 (as revealed by wGBD) and total Cdc42 (as revealed by IT-Cdc42) between the pre-zone and the zone stage: quantification of the patterning index in the zone showed a near doubling of the amount of IT-Cdc42 and wGBD relative to the pre-zone (Fig. 1F).

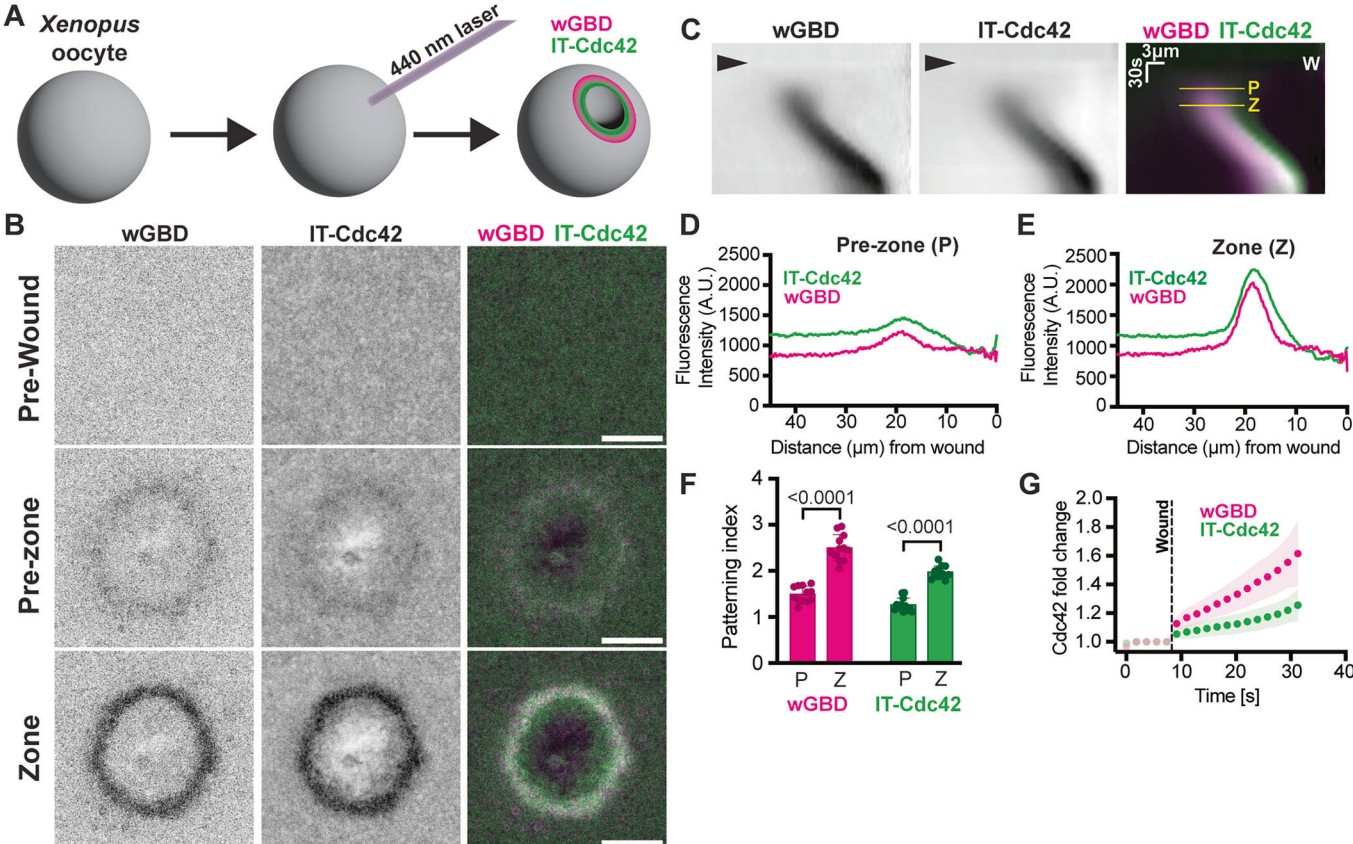

**Figure 1. Quantifying patterning of total and active Cdc42 in vivo.**

(A) Scheme of laser-induced wounding of *Xenopus laevis* oocyte and enrichment of wGBD and IT-Cdc42. (B) Representative micrograph of wGBD (mCh-wGBD) and IT-Cdc42 (IT-GFP-Cdc42) at the plasma membrane before wounding (top, Pre-wound), after wounding prior to zone formation (middle, Pre-zone), and after wounding when the zones are fully formed (bottom, Zone). Scale bar 20 μm. (C) Kymograph of the micrograph from (B) generated by radially averaging signal intensity around the wound over time (Moe et al, 2021; see "Methods"). The arrow denotes when the wound occurred, and W denotes the wound location. The yellow line "P" indicates the Pre-zone time point and the yellow line "Z" indicates the Zone time point. (D) Line scan generated from the kymograph in (C) to show IT-Cdc42 and wGBD fluorescence intensity at the Pre-zone time point as a function of distance from the wound center. (E) Line scan generated from the kymograph in (C) to show IT-Cdc42 and wGBD fluorescence intensity at the Zone time point as a function of distance from the wound center. (F) Patterning indices of IT-Cdc42 and wGBD at the Pre-zone time point compared to the Zone time point. Two-sample *t* tests were used to determine statistical significance between Pre-zone and Zone time points; (mean ± SD). wGDB P value = 4.20128E-09, IT-Cdc42 P value = 1.0572E-08. wGBD and IT-Cdc42 imaged in the same cells, n = 12 individual cells in N = 1 experiment. (G) Timing of IT-Cdc42 and wGBD enrichment around the wound. Mean (Shaded area = SD), n = 9 individual cells in N = 1 experiment per condition.

To independently monitor accumulation of total Cdc42, and to permit direct comparison of these in vivo results to results obtained in vitro (see below) we wounded oocytes microinjected with recombinantly purified complexes of RhoGDI1 with either geranylgeranylated, as a control, or non-prenylatable Cy3-Cdc42 (see "Methods"). Geranylgeranylated Cy3-Cdc42, but not non-prenylatable Cy3-Cdc42 accumulated around wounds (Fig. EV1A) and normalization using either background Cy3-Cdc42 or non-prenylatable Cy3-Cdc42 revealed that the degree of accumulation was similar to that obtained with IT-Cdc42 (1.8–2.2 fold Figs. 1E,F and EV1B–F).

Next, we set out to closely follow the accumulation of total and active Cdc42 over time and therefore increased the time resolution of our experiments. Samples were imaged in rectangular fields, which exploits the fact that the confocal raster moves faster in X than Y. While wGBD typically began accumulation immediately after wounding, the accumulation of IT-Cdc42 was generally slower and often delayed for several seconds after wounding (Figs. 1G and EV1G,H). We made similar observations for internally-tagged

RhoA (IT-RhoA), which also accumulated around wounds only after a rise in RhoA activity as detected by an activity probe (GFP-rGBD) (Fig. 1I), indicating that this behavior is not specific to a particular Rho GTPase. We drew two important conclusions from these experiments: Spatiotemporal patterning of Rho GTPases involves their accumulation at activation sites and this local enrichment appears to be delayed relative to their activation.

## GTPase:GDI dissociation limits GEF-dependent nucleotide exchange in a reconstituted system

To study the mechanism by which Rho GTPases enrich at membranes at sites where they are activated, we reconstituted Rho GTPase activation by GEFs from soluble complexes with RhoGDI in a membrane environment in vitro. To this end, we isolated stoichiometric complexes of geranylgeranylated Cy3-Cdc42 bound to Cy5-GDP and RhoGDI1 (Fig. 2A, see "Methods"). The proximity of the fluorophores allowed for Förster resonance energy

## Rho GTPase membrane binding and nucleotide exchange

### Dbl-family GEFs (ITSN)

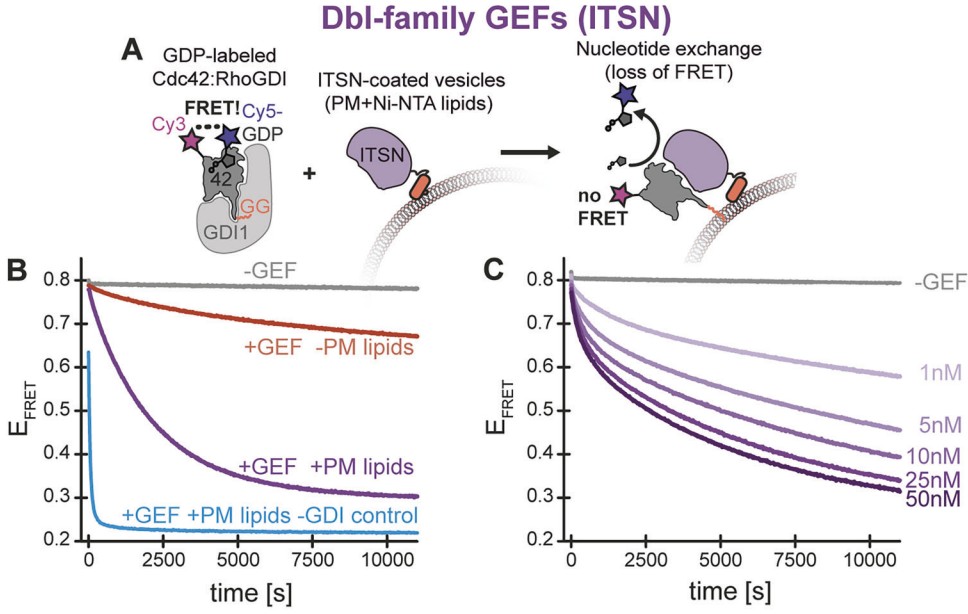

### DOCK-family GEFs (DOCK1:ELMO1)

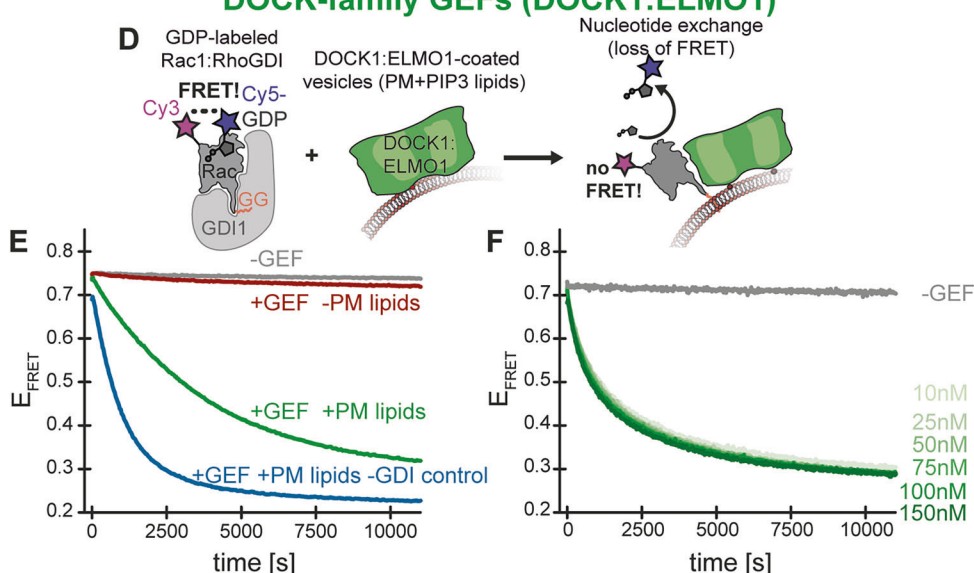

## Rho GTPase nucleotide exchange vs GDI dissociation

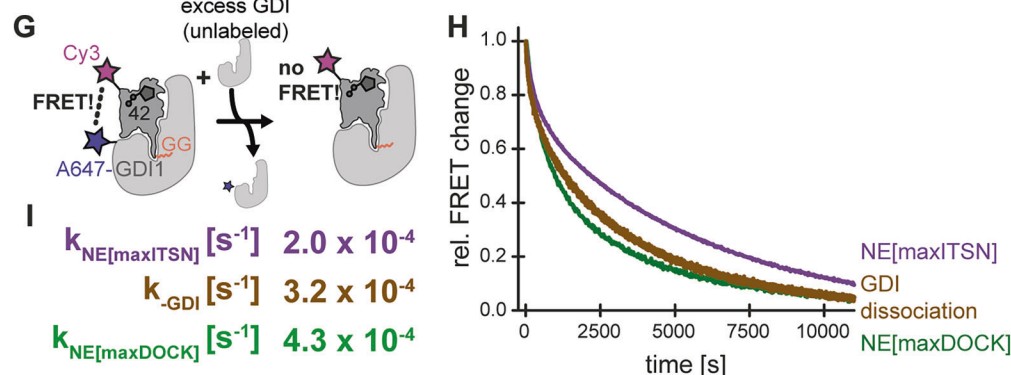

**Figure 2.** Rho GTPase activation from soluble RhoGDI complexes by membrane-bound GEFs in vitro kinetically coincides with GDI dissociation.

(A) Scheme of the FRET-based nucleotide exchange assays from Cy5-GDP:Cy3-Cdc42:RhoGDI complexes in the presence of ITSN$_{cat}$-coated LUVs. (B) Decrease of the acceptor/donor fluorescence intensity ratio as a function of time. At $t = 0$ s, Cy5-GDP:Cy3-Cdc42:RhoGDI1 or Cy5-GDP:Cy3-Cdc42 complexes (100 nM) were mixed with either ITSN$_{cat}$-His$_{10}$ (10 nM), either in solution or pre-bound to PM LUVs (125 μM total lipids containing 0.2% Ni$^{2+}$-NTA-DGS, or PM LUVs alone as indicated in the presence of excess unlabeled GDP. (C) At $t = 0$ s, Cy5-GDP:Cy3-Cdc42:RhoGDI1 (100 nM) were mixed in the presence of excess unlabeled GDP with either ITSN$_{cat}$-His$_{10}$ at concentrations as indicated, which was pre-bound to PM LUVs (125 μM) or PM LUVs alone. (D) Scheme of the FRET-based nucleotide exchange assays from Cy5-GDP:Cy3-Rac1:RhoGDI complexes in the presence of DOCK1:ELMO1-coated LUVs. (E) Time traces of the acceptor/donor fluorescence intensity ratio as a function of time. At $t = 0$ s, Cy5-GDP:Cy3-Rac1:RhoGDI1 or Cy5-GDP:Cy3-Rac1 complexes (100 nM) were mixed with either DOCK1:ELMO1 (10 nM), either in solution or pre-bound to PM LUVs (125 μM total lipids, 4% PtdIns(3,4,5)P$_3$), or PM LUVs alone as indicated in the presence of excess unlabeled GDP. (F) At $t = 0$ s, Cy5-GDP:Cy3-Cdc42:RhoGDI1 or Cy5-GDP:Cy3-Cdc42 complexes (100 nM) were mixed in the presence of excess unlabeled GDP with either DOCK1:ELMO1 complexes at concentrations as indicated that were pre-bound to PM LUVs (125 μM total lipids, 4% PtdIns(3,4,5)P$_3$), or PM LUVs alone. (G) Scheme of the FRET-based GDI dissociation assay. (H) Decrease of the normalized acceptor/donor fluorescence intensity ratio as a function of time (brown). At $t = 0$ s Cy3-Cdc42:A674-RhoGDI1 complexes (80 nM) are mixed with excess unlabeled RhoGDI1 (5 μM). Time courses of nucleotide exchange at saturating GEF concentrations either in the presence of ITSN$_{cat}$-His$_{10}$ (violet) or DOCK1:ELMO1 (green) as observed in (C) or (F) are shown for comparison. (I) Maximal rates of nucleotide exchange by ITSN$_{cat}$-His$_{10}$ (violet) or DOCK1:ELMO1 (green) compared to dissociation of RhoGDI1 (brown) obtained from mono-exponential fits to the data in (H). All traces shown represent the mean from three independent experiments ($N = 3$).

transfer (FRET). These complexes were then mixed with excess large unilamellar vesicles (LUVs) mimicking the phospholipid composition of the inner leaflet of the plasma membrane (PM lipids, see "Methods", (Lorent et al, 2020)). A catalytically active fragment of the DBL-family GEF Intersectin (ITSN$_{cat}$) was stably tethered to the LUVs via Ni-NTA:His$_{10}$-tag interactions to promote nucleotide exchange (Fig. 2A). Notably, localization of such minimal GEF fragments to the plasma membrane is sufficient to activate Rho GTPases in living cells (Levskaya et al, 2009; Oakes et al, 2017; van Unen et al, 2015). We monitored the nucleotide exchange of Cy5-GDP in Cy3-Cdc42 to unlabeled GDP, which was accompanied by a loss in FRET as indicated by changes in the ratio of the fluorescence intensities over time (Fig. 2B). In the absence of GEF, nucleotide exchange failed to occur, while the presence of soluble GEF resulted in modest exchange (Fig. 2B). Tethering the GEF to the surface of PM lipid LUVs strongly accelerated the reaction, however kinetics remained much slower than reactions carried out in the absence of RhoGDI1. To better understand which constraints limit the rate of nucleotide exchange under these conditions, we systematically varied the GEF concentration. Nucleotide exchange rates plateaued at moderate GEF concentrations, reaching a maximum of around 0.0002 s$^{-1}$, indicating that GEF activity was not the limiting factor (Fig. 2C).

Because natural GEFs are typically multi-domain proteins that often assemble into large complexes, we explored how these findings with GEF fragments related to full-length GEF complexes. Furthermore, we wondered about the generality of our findings for other Rho-type GTPases and other GEF families. We therefore switched the GTPase-GEF pair and turned to the heterodimeric DOCK1:ELMO1 complex, which is a GEF for Rac1 that belongs to the second large RhoGEF family, evolutionarily unrelated to Dbl-family proteins. To localize DOCK1:ELMO1 to membranes, we used PM lipid LUVs containing a small (4%) fraction of phosphatidylinositol 3,4,5-triphosphate (PI(3,4,5)P$_3$), its natural lipid input (Côté et al, 2005) (Fig. 2D). We then added dual-labeled Cy3-Rac1:Cy5-GDP:RhoGDI1 complexes in solution. Consistent with the results obtained with the catalytic domain of intersectin and Cdc42:RhoGDI1 complexes, we observed that nucleotide exchange from Rac1:RhoGDI1 complexes required the synchronous presence of the GEF and PM lipid LUVs, but remained much slower compared to reactions run in the absence of GDI (Fig. 2E). Moreover, increasing DOCK1:ELMO1 levels had no impact on the exchange rate, demonstrating that we were working with saturated

GEF concentrations (Fig. 2F). These results show that nucleotide exchange in Rho GTPases from soluble GTPase:GDI complexes is kinetically limited by a process other than GEF activity as such and that this feature is shared among distinct Rho GTPases and Rho GEFs. This is in line with prior observations that Rho GTPase:RhoGDI complexes are poor substrates for GEF-mediated activation on liposomes (Robbe et al, 2003).

The maximal rates of nucleotide exchange from GTPase:GDI complexes were unexpectedly similar (only twofold difference) for two distinct GEF-GTPase pairs. We therefore reasoned that they might be limited by the same biochemical process. As we observed this kinetic limit in the presence, but not the absence of RhoGDI1, we hypothesized that it might correspond to the dissociation of the GTPase from RhoGDI. Consistent with this possibility, we found that the rate of spontaneous dissociation of RhoGDI1 from prenylated Cdc42, which we independently measured by a FRET-based chase assay in the absence of GEFs and membranes (Fig. 2G), closely approximated the maximal nucleotide exchange rates we observed previously (Fig. 2H,I). Collectively, the results indicate that under our conditions, dissociation of the Rho GTPase from RhoGDI1 is the rate limiting step in GTPase activation by membrane-associated GEFs.

## RhoGDI dissociates before Rho GTPases bind to membranes

In principle, dissociation of the Rho GTPase from RhoGDI could occur prior to, or concomitant with, interaction of the GTPase:GDI complex with the membrane. To distinguish between these possibilities, we followed the recruitment dynamics of individual Cy3-Cdc42:A647-RhoGDI1 complexes by synchronous dual-color total internal reflection microscopy (TIRFM, see "Methods"). To ensure that we were able to visualize these dual-labeled complexes at the single-molecule level under our imaging conditions, we carried out control experiments on surface-tethered Cdc42:RhoGDI1 complexes. To this end, we introduced an additional biotin handle at the N-terminal Cy3 label of Cdc42 during complex preparation (see "Methods"). We then diluted biotin-Cy3-Cdc42:A647-RhoGDI1 complexes to very low concentrations (100 pM) and followed their immobilization on neutravidin-coated and biotin-PEG-functionalized coverslips by dual-color TIRFM with maximal time resolution (22 ms per frame, Fig. 3A). Given that these trace concentrations approximated the equilibrium dissociation constant

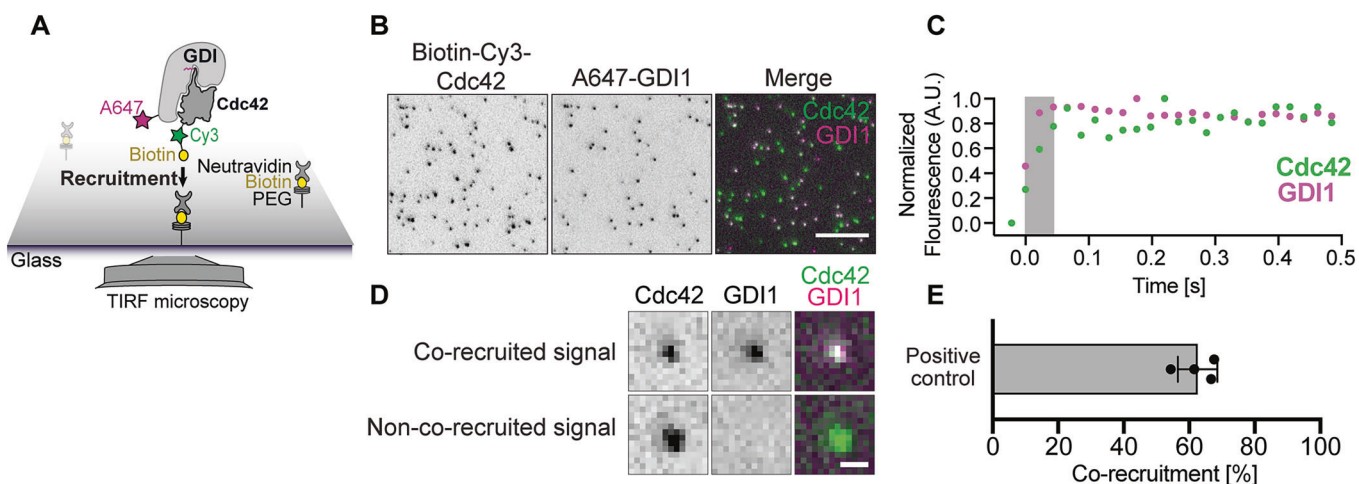

**Figure 3.   RhoGDI dissociation precedes Rho GTPase membrane binding in vitro.**

(A) Scheme of neutravidin-mediated recruitment of biotinylated, dual-labeled Cdc42:RhoGDI1 complexes to functionalized surfaces. (B) TIRFM images of Biotin-Cy3-Cdc42 (left) A647-RhoGDI1 (center) complexes (100 pM, ≤2 min after dilution) on neutravidin-coated biotin-PEG surfaces. Scale bar = 10 μm, $N = 4$ independent experiments, $n = 968$ observations. (C) Normalized fluorescence intensities of co-localized Cy3-Cdc42 (green) and RhoGDI1 (magenta) molecules imaged at maximal frame rate (22 ms) during recruitment at $t = 0$ s. Co-localization during the first three frames after recruitment was considered (grey box, see "Methods"). (D) Dual-color TIRFM images of a Cy3-Cdc42:A647-RhoGDI1 complex (upper) compared to a Cdc42 only (lower) recruitment event under conditions as (B). Scale bar = 1 μm. (E) Mean fraction ± SD of Cy3-Cdc42 molecules co-recruited with A647-RhoGDI1 ($n = 968$, $N = 4$, SD). (F) Scheme of two potential Rho GTPase recruitment mechanisms; either in complex with RhoGDI or free. (G) TIRFM images (Cy3-Cdc42, A647-GDI1, merge, and trajectories of Cdc42, left-to-right) of a PM SLB exposed to freshly-diluted Cy3-Cdc42:A647-GDI1 complexes (100 pM) in solution. Scale bar = 10 μm. (Image from (G) also used in Appendix Fig. S2B). (H) Dual-color TIRFM images at indicated times of a Cy3-Cdc42 molecule upon recruitment (upper, $t = 0$ s at recruitment) and dissociation (lower, $t = 0$ s at time of dissociation) on a PM SLB. Notice the absence of co-localized A647-RhoGDI1. Scale bar = 1 μm. (I) Mean fraction ± SD of Cy3-Cdc42 molecules co-recruited with A647-RhoGDI1 to PM SLBs ($n = 7851$, $N = 3$) compared to the positive control ($n = 968$, $N = 4$) in (E). T test was used to determine significance. (J) Probability (dots) ± SD (area) of Cy3-Cdc42 and A647-GDI1 co-localization as a function of Cdc42 membrane lifetime with $t = 0$ s as the moment of recruitment. Vertical grey box demarks the frames evaluated for co-recruitment. $n = 7851$, $N = 3$.

we previously determined for the Cdc42:RhoGDI1 interaction ($K_D = 90$ pM, (Medina Gomez et al, 2024)), we only imaged for <2 min after complex dilution. The high kinetic stability of the Cdc42:RhoGDI1 complex (Fig. 2I) ensures that a maximum of around 10% of GTPases can dissociate from RhoGDI1 during our measurements, even at these low concentrations. We observed rapid surface binding of both biotin-Cy3-Cdc42 and A647-RhoGDI1 under these conditions (Fig. 3B,C). To quantify the extent of co-localization at the moment of surface recruitment (Fig. 3C,D), we employed automated tracking and co-localization routines (see "Methods"). This analysis revealed that the dominant fraction (63%) of fluorescent Cdc42 molecules was recruited to the neutravidin-coated surface in complex with labeled RhoGDI1 (Fig. 3E). This high fraction of Cdc42 and RhoGDI1 co-recruitment confirmed our ability to detect hypothetical surface recruitment events involving the GTPase:GDI complex. The failure to detect even higher fractions of co-recruitment were likely due to a combination of incomplete fluorescent labeling of RhoGDI1, a low fraction of spontaneous RhoGDI1 dissociation from Cdc42 and rapid bleaching of RhoGDI1 at the high power densities required for single molecule imaging at maximal time resolution.

To investigate in which state Rho GTPases bind to membranes, we added freshly diluted Cy3-Cdc42:A647-RhoGDI1 complexes to supported lipid bilayers (SLBs) of PM lipid composition and monitored Cdc42 membrane recruitment (Fig. 3F). We confirmed that SLBs of PM lipid composition were fluid and continuous (Appendix Fig. S1). Individual molecules of Cy3-Cdc42 readily bound and inserted into the SLBs as indicated from their persistent two-dimensional movement in the plane of the bilayer (Fig. 3G; Movie EV2). In contrast, we observed only very few and highly transient events of A647-RhoGDI binding to the membrane (Fig. 3G; Movie EV2). To determine whether Cy3-Cdc42 first encounter the membrane in complex with A647-RhoGDI, we turned to automated tracking followed by custom developed co-recruitment quantification scripts (see "Methods"). Decisively, we observed that GDI-free Cdc42 represented the great majority of membrane recruitment events (99.71%) while Cdc42:RhoGDI1 complexes accounted for just a tiny fraction of the observed events (0.29%) (Fig. 3H,I). We note that the latter is 213-fold lower than the positive control. Further, these events likely just reflect random encounters between Cy3-Cdc42 and A647-RhoGDI1 on the membrane surface rather than bona fide delivery events, because the probability of Cdc42 co-localizing with RhoGDI1 at the membrane was nearly constant over its lifetime on the membrane and not strongly elevated at the initial moment of membrane recruitment

(Fig. 3J). These results indicate that dissociation of RhoGDI from Rho GTPases occurs in solution and precedes the interaction between Rho GTPases and membranes.

## Rho GEFs act on membrane-bound Rho GTPases and do not recruit GTPases from solution

Based on these observations, we wondered whether core GTPase regulators alter the sequence and or the kinetics of the biochemical events leading to Rho GTPase membrane binding and activation. GEFs, in particular, have been implicated in activating and enriching Rho GTPases at specific membrane locations in vivo, leading to the suggestion that GEFs recruit Rho GTPases directly from the cytosol. To investigate the impact of membrane-associated GEF activity on Rho GTPase activation and recruitment, we first tethered $ITSN_{cat}$ onto SLBs of PM lipid composition via Ni-NTA lipids and quantified its membrane density via single molecule spike-in experiments (Fig. EV2A,B). We intentionally used very high densities (~450 molecules per μm$^2$ translating to ~5% of surface area coverage, see "Methods") of $ITSN_{cat}$ to maximize its potential effect on Cdc42 recruitment (Fig. EV2C). To assess GEF functionality, we monitored the nucleotide exchange of individual Cy3-Cdc42:Cy5-GDP complexes on $ITSN_{cat}$-His$_{10}$-containing SLBs in the presence of unlabeled GTP in solution (Fig. 4A). We observed strongly reduced steady-state densities (Fig. 4B) and dwell times (Fig. 4C) of Cy5-GDP in the presence of surface-tethered $ITSN_{cat}$, indicating that these high GEF densities were capable of activating Cdc42 in less than a second after membrane binding.

To study the recruitment of Cdc42 to GEF-containing membranes, we introduced picomolar concentrations of freshly diluted, dual-labeled Cy3-Cdc42:A647-RhoGDI1 complexes to $ITSN_{cat}$-coated SLBs of PM lipid and visualized GTPase membrane binding via rapid dual-color TIRFM (Fig. 4D–F; Movie EV2). Consistent with the results above, the great majority of membrane recruitment events (99.7%) were represented by isolated Cdc42 recruitment events, lacking RhoGDI1 signal (Fig. 4G). We observed only marginal co-recruitment (0.3%) of Cdc42 and RhoGDI1 to SLBs in the absence of membrane-tethered $ITSN_{cat}$, which was only modestly increased by its presence (1.1%). While the highest co-recruitment probability of Cdc42 and RhoGDI1 coincided with the moment of membrane encounter, the probability remained around 1% (Fig. 4H). To investigate the impact of GEFs on recruitment of Cdc42 to membranes, we compared the membrane binding rate of Cy3-Cdc42 observed at the single molecule level in the presence or

## Nucleotide exchange on supported lipid bilayers

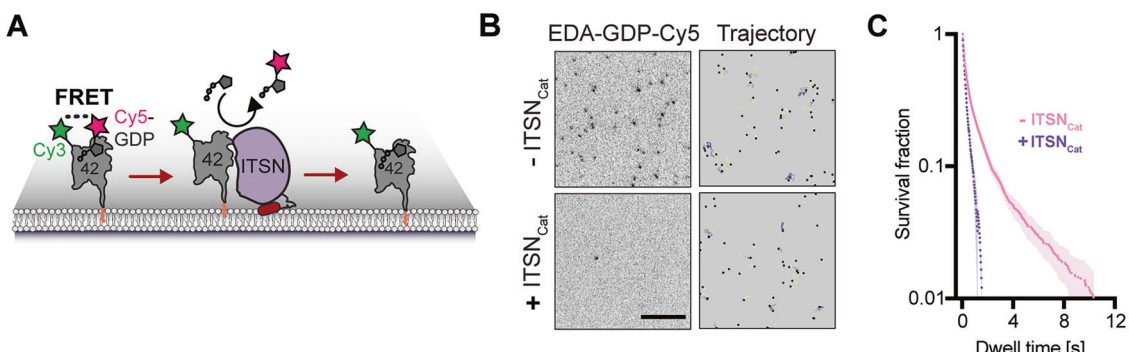

## Does GEF mediate Rho GTPase:GDI co-recruitment?

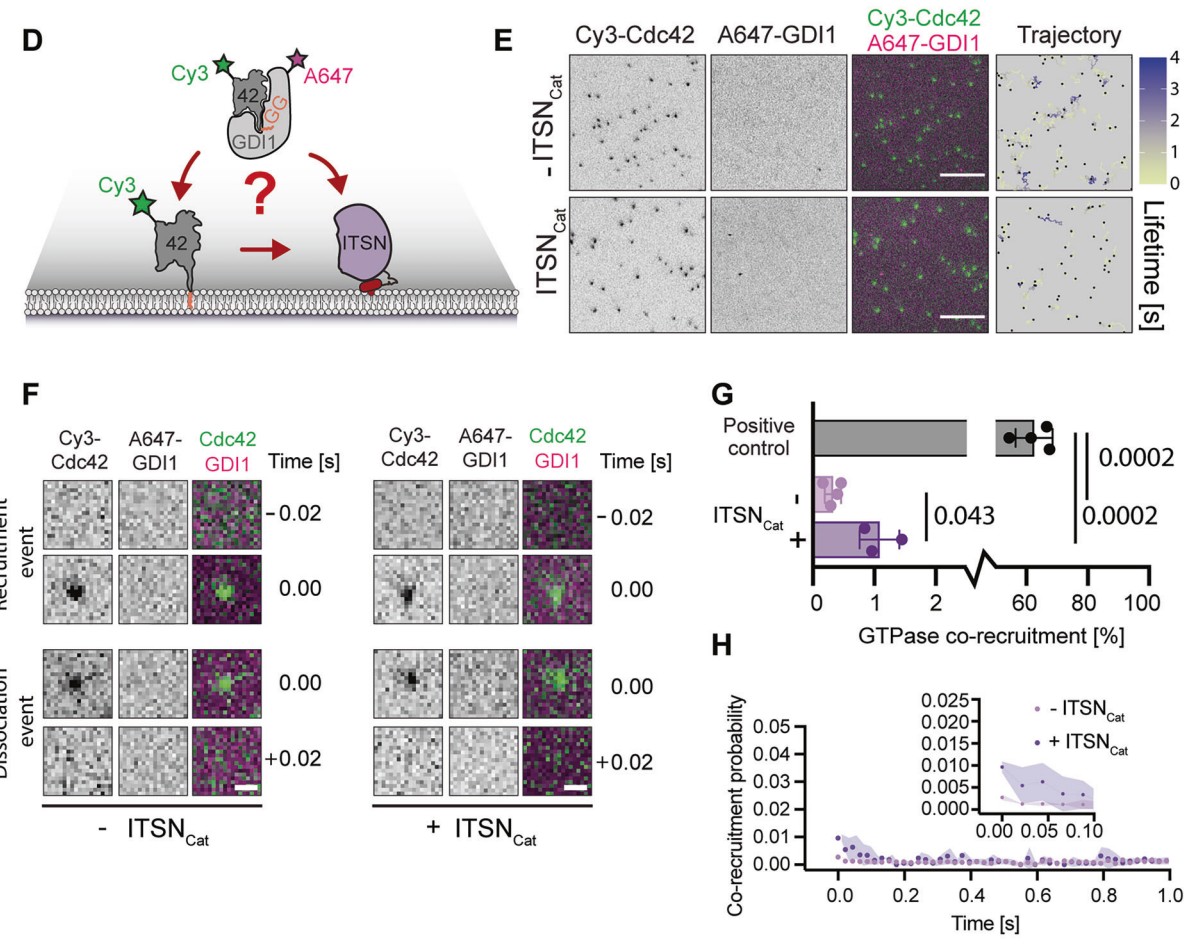

## Does GEF increase Rho GTPase recruitment?

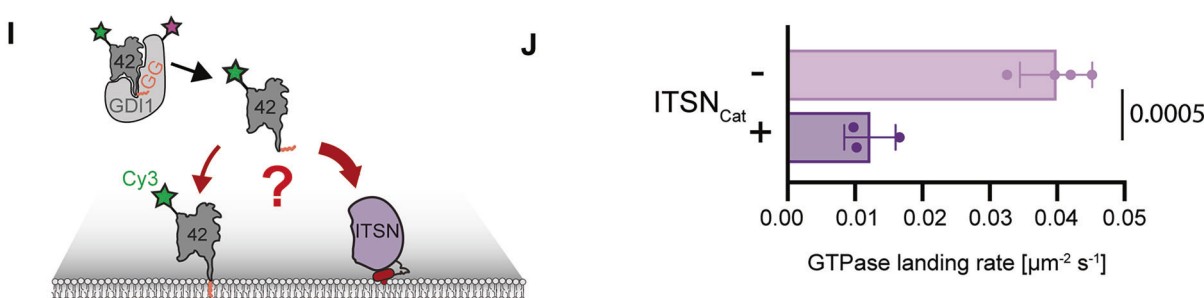

**Figure 4.   RhoGEFs act on membrane-bound Rho GTPases and not their soluble forms.**

(A) Scheme of the FRET-based single molecule nucleotide exchange assay on PM SLBs. (B) Cy5-GDP:Cy3-Cdc42 complexes (50 pM) recruited to PM SLBs (containing 2% $Ni^{2+}$-NTA-DGS) coated or not with high densities (350 molecules/$\mu m^2$) of $ITSN_{cat}$ imaged by sensitized Cy5 emission by TIRFM. Scale bar = 10 μm. (C) Survival fraction of Cy5-GDP as a function of time after membrane recruitment in the absence (pink) and presence (purple) of $ITSN_{cat}$. (-$ITSN_{cat}$ $n = 3348$, $N = 3$, +$ITSN_{cat}$ $n = 3023$, $N = 3$). (D) Scheme of two potential routes to GEF-mediated GTPase recruitment and activation. (E) TIRFM images (Cy3-Cdc42, A647-GDI1, merge, and trajectories of Cdc42, left-to-right) of a PM SLB (containing 2% DGS-NTA(Ni)) coated (bottom) or not (top) with high densities of $ITSN_{cat}$ exposed to freshly-diluted Cy3-Cdc42:A647-GDI1 complexes (100 pM) in solution. Scale bar = 10 μm. (F) Dual-color TIRFM images at indicated times over the lifetime of a Cy3-Cdc42 molecule on a PM SLB, coated (right) or not (left) with high densities of $ITSN_{cat}$ ($t = 0$ s at recruitment). Notice the absence of co-localized A647-RhoGDI1. Scale bar = 1 μm. (G) Mean fraction ±SD of Cy3-Cdc42 molecules co-recruited with A647-RhoGDI1 to PM SLBs in the absence or presence of $ITSN_{cat}$ compared to the positive control ($n = 968$, $N = 4$) (Fig. 3E). (−/+ GEF experiments $n = 12911$, $N = 4$ and $n = 2890$ $N = 3$ respectively). $T$ test was used to determine significance. (H) Probability (dots) ±SD (area) of Cy3-Cdc42 and A647-GDI1 co-localization as a function of Cdc42 lifetime on SLBs coated with (violet) or without (pink) $ITSN_{cat}$. $t = 0$ s is the moment of recruitment. Vertical grey box demarks the frames evaluated for co-recruitment. -$ITSN_{cat}$ $n = 12911$, $N = 4$, +$ITSN_{cat}$ $n = 2890$ $N = 3$. (I) Scheme of GEF-dependent and independent membrane recruitment of free Rho GTPases (J) Cy3-Cdc42 landing rates (mean ± SD) on PM SLBs in the absence (pink) or presence (purple) of $ITSN_{cat}$. -$ITSN_{cat}$ $n = 112911$, $N = 4$, +$ITSN_{cat}$ $n = 2890$, $N = 3$. $T$ test was used to determine significance.

absence of $ITSN_{cat}$ (Fig. 4I). Strikingly, we observed no increase, but a moderate (threefold) reduction in the rate of membrane binding, demonstrating that $ITSN_{cat}$ acts on membrane-bound Cdc42 but does not actively recruit Cdc42 from solution (Fig. 4J). This moderate reduction in membrane binding was likely due to steric exclusion at the high GEF densities used. We made similar observations for $ITSN_{cat}$ immobilized to PM lipid containing bilayers via C-terminal fusion to the Lactadherin C2 domain ($ITSN_{cat}$-2xC2, Appendix Fig. S2), which demonstrated that the technical means of GEF immobilization did not affect our findings. Moreover, we obtained qualitatively similar results for other Rho GTPases such as Rac1 in combination with full-length GEFs complexes from the DOCK-family (Fig. EV3). Finally, we wondered whether other cellular factors not included in the reconstitution might accelerate the membrane recruitment of Rho GTPases, potentially by destabilizing the Rho GTPase interaction with RhoGDI. We therefore decided to include concentrated porcine brain extracts in our SLB experiments (Hume et al, 2014). These extracts are fully capable of supporting rapid Rho GTPase signaling and branched actin assembly on PI(4,5)P₂-containing membranes (Koronakis et al, 2011; Ma et al, 1998) and should therefore likely contain such putative GDI displacement factors. However, we found that extracts supplemented with trace amounts of dual-labeled Cy3-Cdc42:A647-RhoGDI1 complexes neither promoted recruitment of Cy3-Cdc42 nor the co-recruitment of A647-RhoGDI1 (Appendix Fig. S3). In fact, Cdc42 recruitment rates were reduced by twofold, which was likely the consequence of competition with other endogenous peripheral membrane proteins from the extract for SLB binding. In summary, these findings demonstrate that Rho GEFs act specifically on membrane-associated, but not soluble, Rho GTPases and do not significantly contribute to the recruitment of Rho GTPases to membranes from solution. Therefore, the localized activity of Rho GEFs alone is insufficient to concentrate Rho GTPases directly at sites of their activation as observed in vivo (Fig. 1).

## Reconstitution of templated Rho GTPase activity patterns on model membrane systems

To dissect the local membrane enrichment of Rho GTPases under conditions in which they continuously cycle between active and inactive states, we aimed to recreate their spatial patterning in more complex in vitro reconstitutions. To mimic their local activation in

sub-regions of the membrane, we turned to a self-organizing phosphatidylinositol phosphate (PIP) lipid kinase-phosphatase system (Hansen et al, 2019). This system consists of the phosphatidylinositol-4-phosphate 5-kinase (PIP5K) and a competing 5′-phosphatase fused to a lipid binding domain (DrrA-OCRL) which spontaneously generate polarized, micrometer-sized membrane domains enriched for either PI(4)P or PI(4,5)P₂ lipids (Hansen et al, 2019). Consistent with previous results (Hansen et al, 2019), exposure of a SLB containing PM lipids and low amounts of PI(4)P and PI(4,5)P₂ (2% each of total lipids) to PIP5K and DrrA-OCRL resulted in patterns of mutually exclusive PIP domains, as revealed by TIRF microcopy using the fluorescently-labeled PI(4,5)P₂-binding PH domain of PLCδ (A647-PH) (Fig. 5A,B). Once formed, these PIP patterns remained relatively stable throughout our experiment.

To harness this system to study Rho GTPase patterning, we fused $ITSN_{cat}$ to the PI(4,5)P₂-binding PLCδ(PH) domain ($ITSN_{cat}$-PH) to create inducible patterns of local Cdc42 activation (Fig. 5C,D; Appendix Fig. S4; Movie EV3). Prior to $ITSN_{cat}$-PH addition, prenylated A647-Cdc42 bound homogenously to the SLB during the formation of PIP patterns (Fig. 5E,F) as did a Cdc42 activity sensor (A488-wCRIB) (Fig. 5F). Upon $ITSN_{cat}$-PH addition, Cdc42 was rapidly activated (Fig. 5F,G; Movie EV4). Initially, activation was predominantly in the PI(4,5)P₂ membrane areas recruiting the GEF, as indicated by increased A488-wCRIB binding (Fig. 5F,G). However, Cdc42 activity subsequently increased in the PI(4)P regions lacking GEF (Fig. 5F,G; Movie EV4). In contrast, the total amounts of membrane-bound A647-Cdc42 changed only slightly after GEF addition, regardless of the membrane area (Fig. 5F). To quantify the relative levels of active Cdc42 in the GEF-containing compared to the GEF-lacking membrane area, we used the signal of the PI(4,5)P₂ sensor to segment time-lapse images into mutually exclusive regions and then calculated the ratio of fluorescence intensities of either A488-wCRIB or A647-Cdc42 inside to outside of the GEF-containing area over time to calculate a patterning index (PI; see "Methods"). This analysis showed that GEF addition resulted in a transient peak in the patterning of active Cdc42, which gradually returned to levels closer to baseline (PI ≈ 1.35 with 1.0 indicating equal partitioning, Fig. 5H). This gradual decrease was likely driven by a combination of the diffusion of active Cdc42 out of the GEF-containing areas and activation of Cdc42 by low GEF activity in solution. These effects are expected to dominate over time, as they are only opposed by the low intrinsic hydrolysis activity of Cdc42.

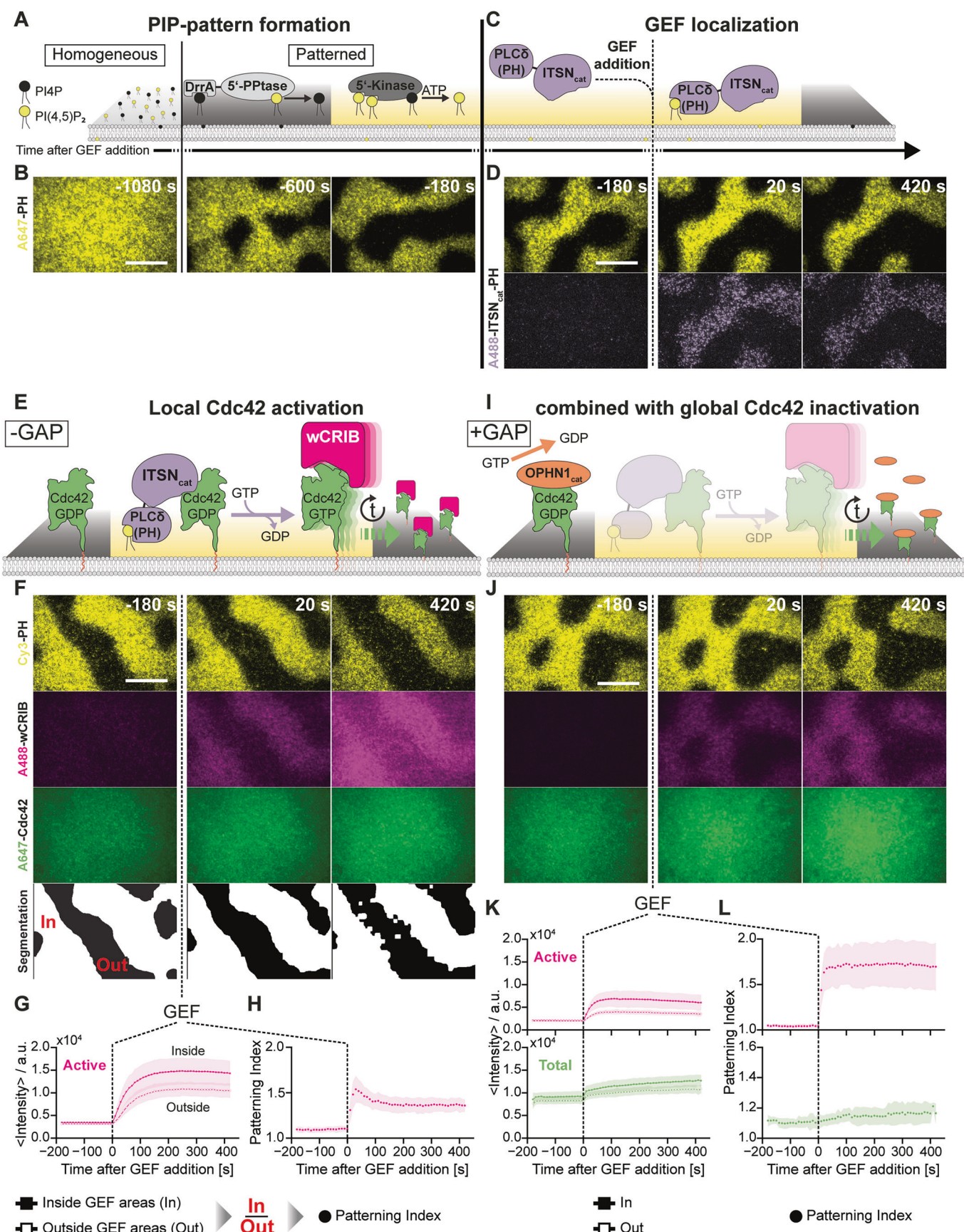

**Figure 5.  Reconstitution of membrane-templated Rho GTPase activity patterns in vitro.**

(A) Scheme of PI(4)P and PI(4,5)P$_2$ lipid pattern formation by PIP5K and DrrA-OCRL on PM SLBs. (B) Time-lapse TIRFM images of pattern formation on PM SLBs (initially containing 2% PI4P and PI(4,5)P$_2$ each) by PIP5K (20 nM) and DrrA-OCRL (here 4 nM, for all following experiments 6 nM were used) visualized by A647-PH (2 nM, yellow) at indicated times before GEF addition. (C) Schematic representation or (D) time-lapse multi-color TIRFM images of PIP lipid patterns on PM SLBs visualized by A647-PH (2 nM, yellow) at indicated times before or after addition of A488-ITSN$_{cat}$-PH (1 nM, purple) at $t$ = 0 s. (E) Scheme of the local activation of Cdc42 by ITSN$_{cat}$-PH in the PI(4,5)P$_2$-rich areas of PM SLBs, the resulting binding of the activity probe wCRIB and the diffusion of active Cdc42 into GEF-lacking areas over time. (F) Time-lapse multi-color TIRFM images of Cy3-PH (2 nM, yellow), A488-wCRIB (40 nM, magenta) and A647-Cdc42 (5.5 nM, green) on PIP patterns at indicated times before or after addition of ITSN$_{cat}$-PH (1 nM) at $t$ = 0 s. Bottom row shows segmentation based on the lipid pattern. (G) Average intensities inside (filled squares, $<I_{In}>$) and outside (hollow squares, $<I_{Out}>$) of the GEF-containing membrane areas or (H) patterning index (full circle, $PI = <I_{In}> / <I_{Out}>$) of A488-wCRIB (magenta) over time. (I–L) Correspond to (E–H), with the additional presence of OPHN1$_{cat}$ (20 nM). However, the segmentation is not shown in (J). Additionally, in (K) are the average intensities inside and outside of the GEF-containing membrane areas and in (L) the patterning index of A647-Cdc42 (green) over time. All numeric data represent the mean from three independent experiments (symbols) ±SD (shaded areas) ($N$ = 3). All scale bars are 20 µm. (Image from (J) also used in Appendix Fig. S5A).

To kinetically limit the activation of Cdc42 and thereby stabilize its activity pattern, we added a soluble RhoGAP –the catalytic domain of Oligophrenin-1 (OPHN1$_{cat}$)– to our system (Fig. 5I). This resulted in the near complete inactivation of Cdc42 before addition of the GEF (Fig. 5J,K; Movie EV4). However, GEF addition still led to a rapid increase in Cdc42 activity (Fig. 5J,K), albeit to reduced (~40%) levels compared to when OPHN1$_{cat}$ was absent, as expected (compare Fig. 5K–G). Remarkably, the patterning index of active Cdc42 activity now showed sustained activation that was significantly enriched in the GEF-containing area (PI ≥ 1.7) over long periods of time (Fig. 5L). Thus, we successfully reconstituted a minimal system capable of establishing and maintaining spatially-templated Rho GTPase activity patterns through the combination of a membrane-targeted GEF and soluble GAP. However, this patterning did not extend to total Cdc42, which remained evenly distributed across the whole bilayer, which manifested in a patterning index close to 1.0 for total Cdc42 (Fig. 5K,L). While GEF addition resulted in a minor increase in membrane-bound Cdc42 levels over time, this occurred equally in GEF-containing and GEF-lacking areas (Fig. 5K,L). Therefore, the combination of Rho GEFs and Rho GAPs alone is insufficient to directly couple the spatial and catalytic cycles of Rho GTPases and to enrich Rho GTPases at sites of activity. Hence, we concluded, that the mechanistic origin of this coupling must reside in other factors present in vivo.

## Effector proteins can enrich active Rho GTPases locally on membranes

One plausible candidate for coupling the catalytic and spatial cycles is RhoGDI, which has been suggested to selectively extract inactive Rho GTPases from membranes (Freisinger et al, 2013; Johnson et al, 2009), although other work suggests that its preference for inactive over active Rho GTPases is comparatively modest (Golding et al, 2019). To study the effect of RhoGDI on reconstituted patterning, we replaced free, prenylated Cdc42 by stoichiometric complexes of prenylated Cdc42 with RhoGDI1 (Fig. 6A). Experiments showed that we had to use 100-fold higher concentrations of Cdc42:RhoGDI1 complexes compared to free Cdc42 (600 nM vs 5.5 nM) to reach similar levels of membrane-bound GTPases (Appendix Fig. S5). This observation matches mass action estimations, which predict a free concentration of ~7 nM of both free Cdc42 and RhoGDI at equilibrium starting from a 600 nM 1:1 complex and a K$_D$ of ~90 pM (Medina Gomez et al, 2024). We

could not explore higher excess concentrations of free RhoGDI1, which would result in a drastic reduction of membrane-bound Cdc42 as predicted by mass action. At the conditions established, GEF addition induced a sustained Cdc42 activity pattern (Figs. 6B,C and EV4A–F; Movie EV5). However, enrichment of Cdc42 in GEF-containing regions was not observed (Figs. 6B,C and EV4D–F). These results suggest that RhoGDI cannot selectively destabilize inactive Cdc42 on membranes, at least at the low free concentrations that were compatible with retaining significant amounts of membrane-bound Cdc42.

We therefore turned to Rho GTPase effector proteins. While constituting a large and diverse class of proteins, effectors share a combination of key features that might allow them to selectively stabilize active Rho GTPases on membranes. In addition to their defining ability to discriminate between active and inactive GTPases, effectors also typically weakly bind either directly or indirectly to PM components and/or associate with Rho GTPases through more than a single interaction, either via multiple GTPase-binding domains or by assembling into higher-order oligomers (Bishop and Hall, 2000; Meca et al, 2019; Mosaddeghzadeh and Ahmadian, 2021). Thus, we investigated how physiological effectors endowed with such avidity-increasing features might affect Rho GTPase patterning. To this end, we replaced the Cdc42 activity sensor, A488-wCRIB, with full-length Atto488-N-WASP, which contains a polybasic stretch adjacent to the CRIB domain that binds to PI(4,5)P$_2$ lipids (Fig. 6D). Owing to this preferential binding to PI(4,5)P$_2$ lipids, N-WASP enriched in the PI(4,5)P$_2$-areas even before GEF addition, which manifested in its increased baseline patterning index (PI = 1.1) (Fig. 6E,F; Movie EV5) that further increased in response to GEF addition as expected (Fig. 6E,F). Strikingly, Cdc42 activation now coincided with its significant enrichment in the GEF-containing and N-WASP-concentrated areas (PI = 1.35, Fig. 6E,F). This directly shows that an increased membrane avidity via an additional effector-membrane interaction can be sufficient for membrane enrichment of active Rho GTPases. We hypothesized that an increase in GTPase-binding valency of effector proteins might elicit similar effects (Fig. 6G). To test this hypothesis, we mimicked effector dimerization by a wCRIB tandem construct (2xwCRIB). We observed that substitution of A488-wCRIB by A488-2xwCRIB resulted in an improved patterning index of the activity sensor, in line with similar observations in vivo (Mahlandt et al, 2021). More importantly, wCRIB dimerization also resulted in the significant enrichment of Cdc42 in regions of its activity (Fig. 6H,I; Movie EV5), which was also observed if

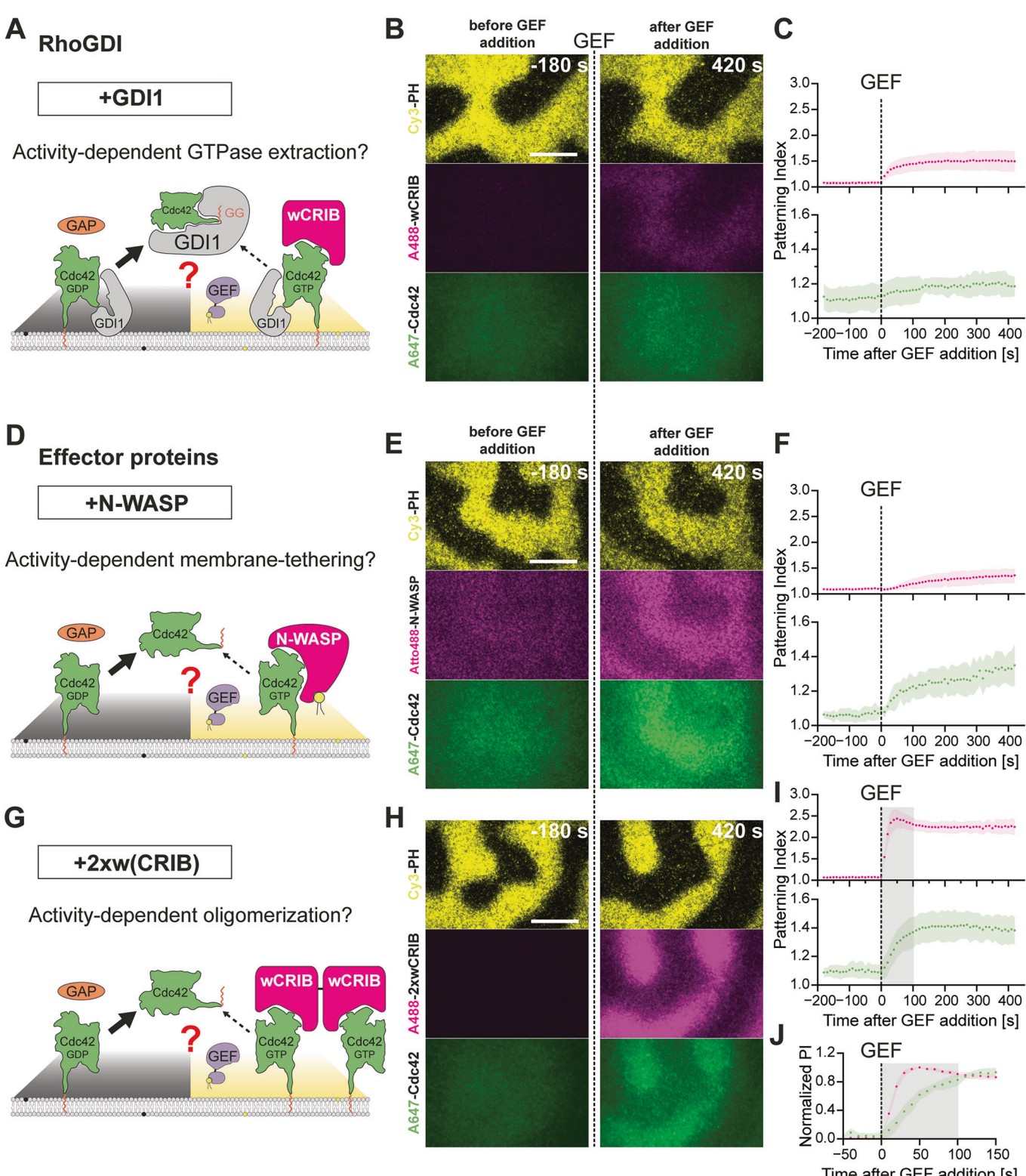

RhoGDI1 was present in addition to 2xwCRIB (Fig. EV4G,H). This was in direct contrast to experiments with wCRIB or RhoGDI1 alone. Importantly, the strong accumulation of Cdc42 was temporally delayed compared to the recruitment of 2xwCRIB

(Fig. 6J), which mimicked the delayed enrichment of total Cdc42 we previously observed around single cell wounds (Fig. 1G). Hence, we successfully reconstituted important in vivo-like characteristics of Rho GTPase patterning in vitro. These results show that effector

**Figure 6. Effector proteins can enrich Rho GTPases at membrane sites of their activity.**

(A) Scheme or (B) time-lapse multi-color TIRFM images of the effects of RhoGDI1 on templated Rho GTPase activity patterns. Images of Cy3-PH (2 nM, yellow), A488-wCRIB (40 nM, magenta) and A647-Cdc42:RhoGDI1 complexes (600 nM, green) on PIP patterns at indicated times before or after addition of ITSN$_{cat}$-PH (1 nM) at $t = 0$ s in the presence of OPHN1$_{cat}$ (20 nM). (Image from (B) also used in Fig. EV4D and Appendix Fig. S5A). (C) Patterning indices ($PI = <I_{In}> / <I_{Out}>$) of A488-wCRIB (magenta) or A647-Cdc42 (green) under conditions as in (B) over time. (D) Schematic representation or (E) time-lapse multi-color TIRFM images of the effects of full-length N-WASP on templated Rho GTPase activity patterns. Conditions as in (B) with A647-Cdc42:RhoGDI1 complexes replaced by A647-Cdc42 (5.5 nM, green) and A488-wCRIB replaced by Atto488-N-WASP (40 nM, magenta). (F) corresponds to (C) under conditions as in (E). (G) Schematic representation or (H) time-lapse multi-color TIRFM images of the effects of the synthetic dimerization mimic 2xwCRIB on templated Rho GTPase activity patterns. Conditions as in (E) with Atto488-N-WASP replaced by A488-2xwCRIB (20 nM, magenta). (I) corresponds to (C) under conditions as in (H). Additionally, the grey shaded box highlights the first 100 s after the addition of ITSN$_{cat}$-PH. (J) Normalized patterning indices of A488-wCRIB (magenta) or A647-Cdc42 (green) under conditions as in (H) over time with the grey shaded box highlighting the first 100 s after the addition of ITSN$_{cat}$-PH. All numeric data represent the mean from three independent experiments (symbols) ±SD (shaded areas) ($N = 3$). All scale bars are 20 µm.

proteins harboring avidity-increasing features are crucial agents to promote Cdc42 pattern formation on membranes.

## Effectors stabilize active Rho GTPases on membranes by two separate mechanisms

We considered two distinct mechanisms by which effectors might enrich active GTPases at the membrane. First, the additional interactions with the membrane could slow down the diffusion of effector:GTPase complex compared to free GTPases within the membrane, effectively trapping Rho GTPases at sites of their activity (Fig. 7A). Second, the increased membrane avidity of effector:GTPase complexes could simply prolong the dwell time of active Rho GTPases at the membrane (Fig. 7B). To test the first hypothesis, we studied the diffusion of Cdc42 by single molecule TIRFM in our reconstitutions. To this end, we reduced the fraction of labeled A647-Cdc42 molecules in our reconstitutions by 100-fold, so that individual molecules could be clearly visualized (Fig. 7C). In the presence of the simple activity sensor (wCRIB), individual A647-Cdc42 molecules moved with nearly similar speeds both inside and outside of the GEF-containing areas as indicated by both step size (Fig. 7D) and mean squared displacement analysis (Fig. 7E) (see "Methods"). The presence of both 2xwCRIB and full-length N-WASP lead to slowed Cdc42 diffusion, which was slightly more pronounced inside GEF-containing regions (Fig. 7D,E). Hence, effectors that increase the overall membrane avidity might slow the diffusion of active Rho GTPases.

To study the role of effector-mediated membrane stabilization, we turned to microfluidic flow-out experiments, which we previously established (Golding et al, 2019). Hence, we visualized the membrane dissociation kinetics of A647-Cdc42 undergoing activity patterning in the presence of either wCRIB (Fig. 7F; Movie EV6) or 2xwCRIB (Fig. 7F; Movie EV6) by TIRFM. Membrane dissociation was induced by a constant flow of buffer containing all reaction components except for prenylated Cdc42. We were unable to perform these experiments with full length N-WASP, because of the high quantities of proteins required in the flow-out mix. To accelerate membrane dissociation, we also added low amounts of free RhoGDI1 (22 nM) to the flow-out mix. While Cdc42 dissociated from the membrane with very similar rates inside and outside of the GEF-containing areas in the presence of wCRIB, effector dimerization lead to significantly prolonged (by 1.4-fold) GTPase dwell times inside the GEF-containing region (Fig. 7F,G). We therefore concluded that effectors can not only slow down, but also kinetically stabilize Rho GTPases on membranes. Both

mechanisms can contribute to enrichment at sites of Rho GTPase activity.

## Cdc42 activation coincides with prolongation of plasma membrane retention

All of our in vitro results suggest that accumulation of total Rho GTPases at sites of activation reflects increased retention rather than increased recruitment. This hypothesis predicts that activation of a Rho GTPase should precede accumulation of that Rho GTPase. To test this prediction in vivo, we generated Cdc42 internally tagged with photoactivatable GFP (IT-PA-GFP-Cdc42). Photo-activatable GFP harbors a point mutation that prevents it from fluorescing until photoactivation via illumination with 405 nm light (Patterson and Lippincott-Schwartz, 2002). This permitted us to monitor the retention of total Cdc42 at the plasma membrane by measuring the half-time to disappearance following photoactivation in different regions around wounds and different times after wounding (Fig. 8A). Consistent with the prediction made by the retention model, the half-time to disappearance following photoactivation was significantly shorter in cells prior to wounding or in regions away from the zone of Cdc42 activation around wounds than it was within the Cdc42 zone (Fig. 8B–K). Further, the half-time to disappearance in photoactivated regions made within the Cdc42 zone was correlated with the time after wounding (Fig. 8L; Movie EV7), as expected if activation promotes retention.

The above results show that Cdc42 retention is correlated with Cdc42 activity as the cell healing process unfolds. However, the wound response includes other changes that could potentially lead to increased Cdc42 retention, such as accumulation of phosphoi-nositide lipids around the wound (Vaughan et al, 2014). To directly assess the impact of increased Cdc42 activation on its retention in vivo, we fused the C2 domain of protein kinase Cβ (PKCβ-C2 to the catalytic domain of intersectin, based on the previous demonstration that PKCβ-C2-fusions are specifically targeted to the edges of single cell wounds (Moe et al, 2021). As expected, a C2-Itsn-CAT-BFP fusion is virtually undetectable at the plasma membrane prior to wounding, but immediately after wounding, it is recruited to the wound edge where produces a striking increase in both the rate and extent of wGBD and IT-Cdc42 recruitment to wounds relative to controls (Fig. EV5). Further, it also significantly increases the retention time of IT-PA-GFP-Cdc42, confirming the link between GEF activity, Cdc42 activation, and Cdc42 retention in vivo (Fig. 8M–T).

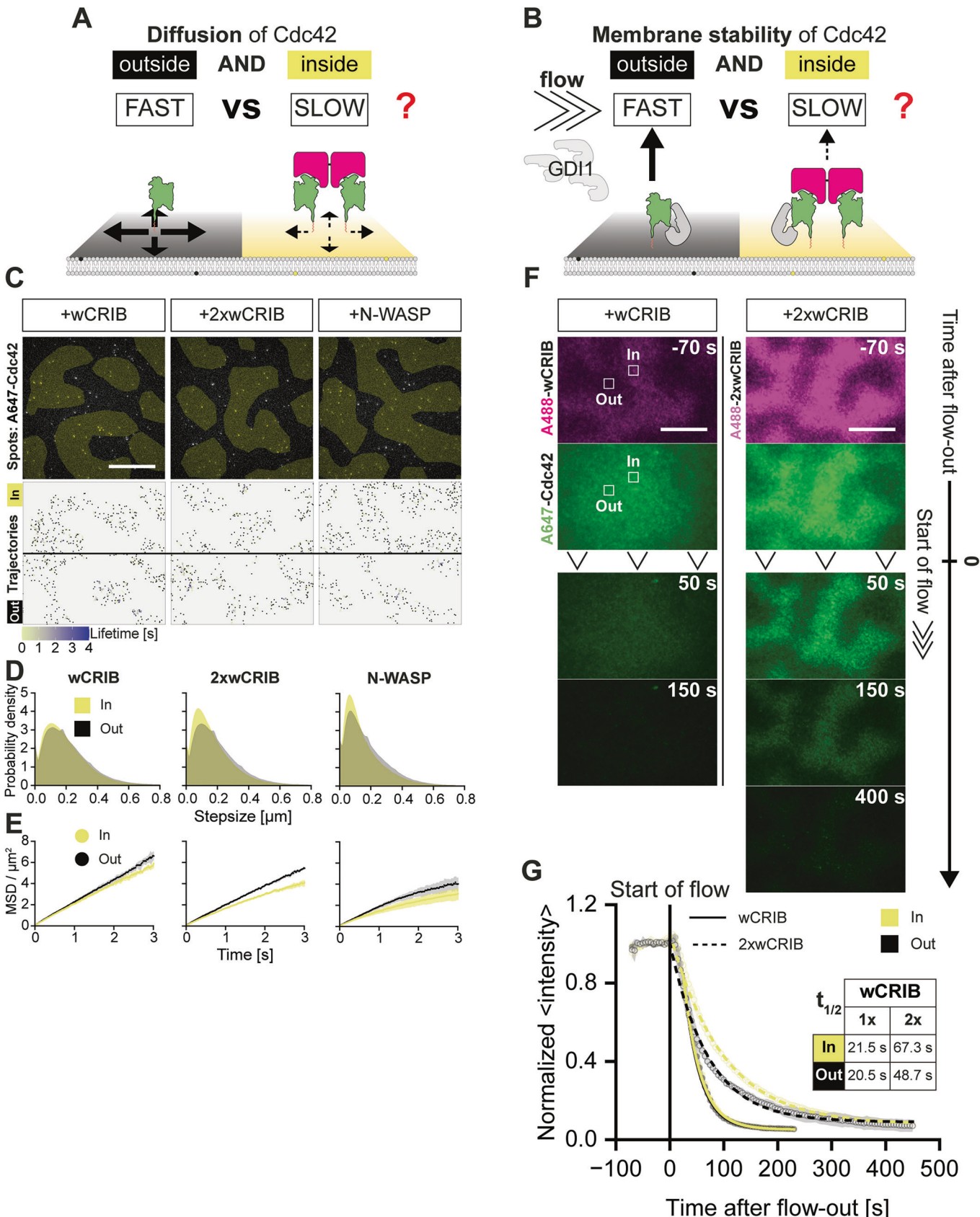

**Figure 7. The mechanism of effector-driven Rho GTPase enrichment at membranes in vitro.**

(A) Schematic representation of effector-mediated changes in diffusivity or (B) membrane stability of Cdc42 as possible mechanisms of active Rho GTPase membrane enrichment. (C) Still images (top) and trajectories (bottom) of single A647-Cdc42 molecules (5.5 nM total, 55 pM labeled) on PIP-templated activity patterns either inside (yellow) or outside (black) of GEF-containing regions in the presence of either wCRIB, 2xwCRIB or N-WASP as indicated. (D) Stepsize distribution or (E) MSD over time of Cdc42 trajectories inside (yellow) and outside (black) GEF areas at conditions as indicated. (F) Time-lapse multi-color TIRFM images of PIP-templated Cdc42 activity patterns in the presence of A488-wCRIB or A488-2xwCRIB at indicated times before or after buffer flow-out at t = 0 s. Flow-out buffer contained all proteins at the same concentrations before flow onset, except for A647-Cdc42 that was omitted, and additional free RhoGDI1 (22 nM) to accelerate Cdc42 membrane dissociation. White boxes (4.1 μm × 4.1 μm) indicating ROIs used for analysis. (G) Normalized average intensities of A647-Cdc42 inside (yellow) or outside (black) GEF-containing areas as a function of time after buffer wash-out in the presence of either A488-wCRIB (solid lines) or A488-2xwCRIB (dashed lines). Half-times were obtained by mono-exponential fits to the data. All numeric data represent the mean from three independent experiments (symbols) ±SD (shaded areas) (N = 3). All scale bars are 20 μm.

## Discussion

The sequence of biochemical events leading to the membrane association, activation and local accumulation of Rho GTPases from the soluble pool of GTPase:GDI complexes has remained controversial for more than two decades. The development of biochemically defined in vitro reconstitution systems has allowed us to delineate a clear hierarchy of individual steps in this process. The sequence begins with dissociation of the inactive GTPase from the GDI, followed by insertion of the dissociated, inactive GTPase into the membrane (Fig. 9). The membrane-associated GTPase is then activated by membrane-bound GEFs and the activation promotes retention of the GTPase in the membrane (Fig. 9).

Remarkably, the results confirm that RhoGDI severely limits the process of membrane insertion and activation while also revealing that GEFs neither promote the displacement of the GTPase from the GDI nor accelerate the recruitment of the GTPase to the membrane. If these findings seem peculiar, they have the virtue of being consistent with structural studies showing that RhoGEFs and RhoGDI share partially overlapping binding sites at the Rho GTPases switch regions (Hoffman et al, 2000; Worthylake et al, 2000) and that "hand-over" complexes between soluble GTPase:GDI and membrane-bound GEF are unlikely to form (Medina Gomez et al, 2024). Further, the fact that GEFs cannot directly recruit Rho GTPases to the membrane even in the absence of RhoGDI is in line with the low affinities and highly transient nature of the catalytic interaction between Rho GEFs and Rho GTPases (Guo et al, 2005; Jaiswal et al, 2011).

How then is the kinetic barrier imposed by GDI binding to the GTPase overcome? The unexpected answer seems to be that it isn't. Rather, the extremely large cytosolic reservoir of soluble RhoG-DI:GTPase complexes ensures that a stable pool of GDI-free GTPases should be present at all times. That is, RhoGDI is present at ~1.2–1.5 μM and Rho, Rac and Cdc42 are present at ~400–500 nM each in the typical eukaryotic cell (Beck et al, 2011; Michaelson et al, 2001; Wühr et al, 2014). The GTPase membrane landing rates we observe from trace amounts of GTPase:GDI complexes (~0.02 s$^{-1}$ μm$^{-2}$ at c = 100 pM, Figs. 4J and EV3; Appendix Figs. S2 and 3) should translate to rates of >10 s$^{-1}$ μm$^{-2}$ at physiological GTPase:GDI levels. Hence, high Rho GTPase membrane densities of more than 100 molecules per μm$^2$ can be generated in principle within seconds. This point is also supported empirically by our observation that distinct Cdc42 patterns do not begin to form on SLBs in our reconstituted system until the concentration of GDI:Cdc42 complexes begins to approach 400–500 nM.

Thus, the picture that emerges is of inactive Rho GTPases arriving at the plasma membrane at a constant rate. In the absence of GEF activity, the arrival is balanced by departure, presumably mediated by GDI-based extraction. In the presence of GEF activity, the GTPase is activated, prompting a decrease in its departure rate. While the results presented here indicate that this basic scheme is sufficient to account for local GTPase patterns that are quantitatively on par with those seen in vivo, this should not be taken to mean that other factors such as GDI phosphorylation or GDFs play no role in vivo. Support for both types of regulatory mechanism has been obtained in several in vivo models of Rho GTPase signaling in vivo (DerMardirossian et al, 2006, 2004; Dovas et al, 2010; Takahashi et al, 1997; Yamashita and Tohyama, 2003). Our results do, however, indicate that rather than promoting the handoff of the GTPase from the GDI to the GEFs as in some models, GDI phosphorylation and GDFs are more likely to expand the pool of dissociated GTPase in the cytoplasm, reduce GDI-dependent extraction of the GTPase from the membrane, or both.

The demonstration that GEFs operate on inactive GTPases that are already at the membrane and that the high concentration of GDI:GTPase complex in the cytosol provide a steady source of membrane-bound, inactive GTPases has other important implications: first, a pre-existing pool of inactive GTPases at membranes should, in principle, accelerate GEF-based GTPase activation, since release from the GDI has already occurred. Consistent with this possibility, we find that active Cdc42 accumulates as quickly as we can image after wounding while Cdc42 itself is slower to accumulate. Second, if membrane-associated, inactive Rho GTPases are the normal substrate for Rho GEFs, it follows that there is nothing to prevent a GEF from immediately reactivating a GTPase following its inactivation by a GAP. That is, traditional models of the two Rho GTPase cycles envision that following inactivation, Rho GTPases are extracted from membranes by RhoGDI such that there is an obligatory return of the GTPase to the cytosol prior to reactivation (Cho et al, 2019). However, the results presented here indicate that a back-and-forth hand off of a GTPase from a GAP to a GEF is not only possible, but probable, at least in situations where GEFs and GAPs are operating in close proximity (e.g. (Michaud et al, 2022)).

Our findings can also be interpreted in light of decades of theoretical and experimental research on Cdc42 polarity in yeasts, which has distilled several critical features (Goryachev and Leda, 2017; Woods and Lew, 2019). Polarization in these systems depends on (i) positive feedback through activity-dependent recruitment of a GEF, facilitated by scaffolding proteins (Irazoqui et al, 2003; Rapali et al, 2017) and effector kinases (Kozubowski

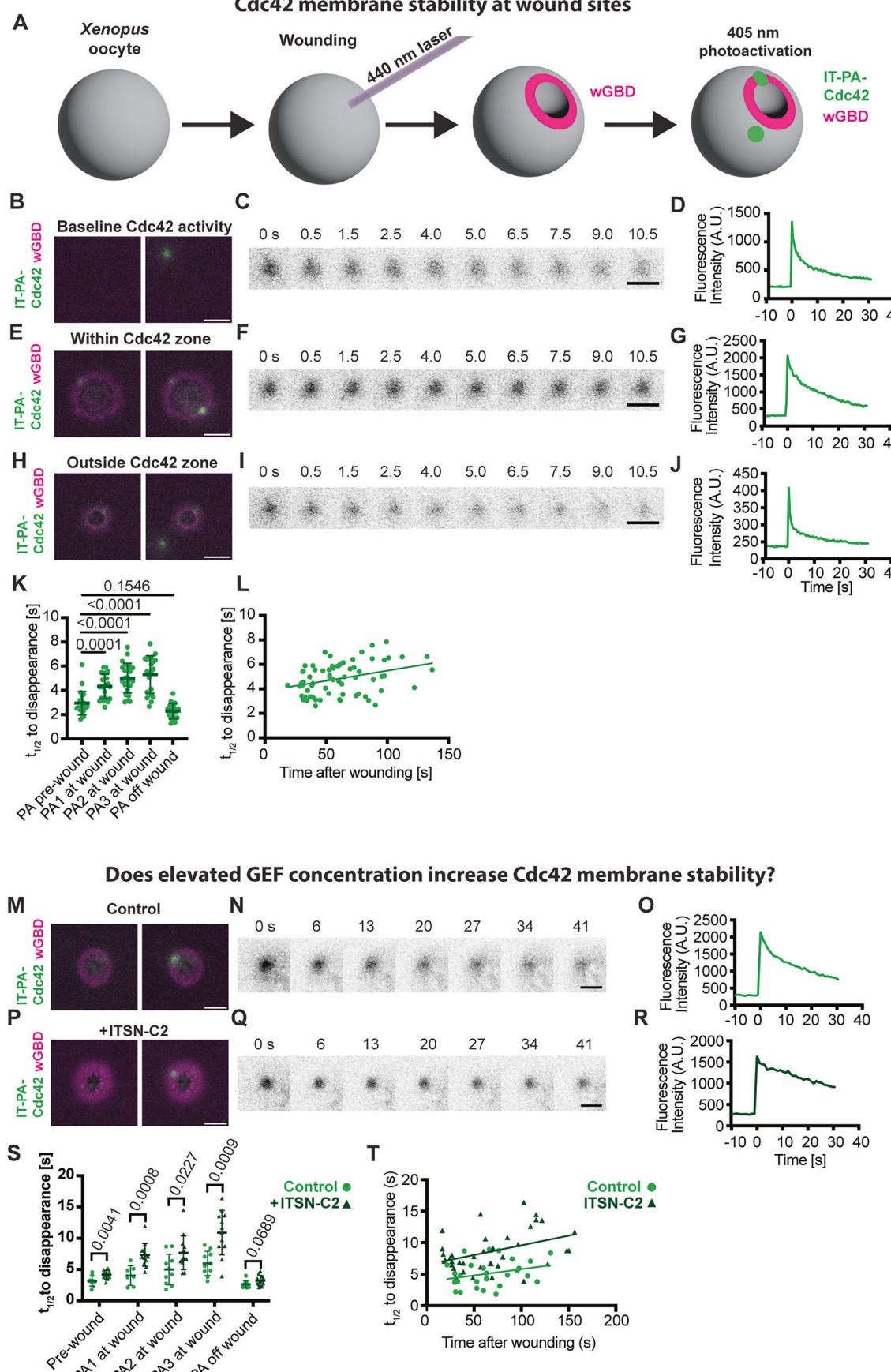

**Cdc42 membrane stability at wound sites**

**Does elevated GEF concentration increase Cdc42 membrane stability?**

**Figure 8. Cdc42 activation precedes Cdc42 retention.**

(A) Scheme showing photoactivation of IT-PA-Cdc42 at a site of baseline/background Cdc42 activity and a site of increased Cdc42 activity around a wound. (B) Representative image of baseline Cdc42 activity (wGBD) before (left) and immediately after (right) photoactivation of IT-PA-Cdc42. Scale bar 20 μm. (C) Montage of IT-PA-Cdc42 from (B) showing disappearance over time. (D) Plot of IT-PA-Cdc42 fluorescence intensity from (B) over time after photoactivation. (E–G) Micrograph (scale bar 20 μm), montage scale bar 10 μm, and plot of IT-PA-Cdc42 photoactivation within the Cdc42 activity zone around a wound. (H–J) Micrograph (Scale bar 20 μm), montage (scale bar 10 μm), and plot of IT-PA-Cdc42 photoactivation outside the Cdc42 activity zone around a wound. (K) Plot of the $t_{1/2}$ to disappearance (mean ± SD) of IT-PA-Cdc42 prior to wounding (PA pre-wound; $n = 2$,) within the Cdc42 zone for the earliest photoactivation (PA1 at wound; $n = 24$), within the Cdc42 zone for a later photoactivation (PA2 at wound; $n = 24$), within the Cdc42 zone for the latest photoactivation (PA3 at wound; $n = 20$), and outside the Cdc42 zone (PA off wound; $n = 22$). One way ANOVA with Dunnett comparison was used to determine significance. PA pre-wound vs. PA2 at wound $P$ value = 1.09682E-08, PA pre-wound vs. PA3 at wound $P$ value = 5.60967E-10. (L) Correlation between $t_{1/2}$ to disappearance of photoactivation within the Cdc42 zone and time after wounding. A positive correlation was found ($P = 0.0043$). (M–O) Micrograph (Scale bar 20 μm), montage (scale bar 10 μm), and plot of IT-PA-Cdc42 photoactivation within the Cdc42 activity zone around a wound. (P–R) Micrograph (scale bar 20 μm), montage (scale bar 10 μm), and plot of IT-PA-Cdc42 photoactivation within the Cdc42 activity zone with ITSN-C2 mediated increased Cdc42 activity. (S) Plot comparing the $t_{1/2}$ to disappearance (mean ± SD) of IT-PA-Cdc42 in control and ITSN-C2 conditions prior to wounding (PA pre-wound; control $n = 10$, ITSN-C2 $n = 13$) within the Cdc42 zone for the earliest photoactivation (PA1 at wound; control $n = 7$, ITSN-C2 $n = 13$), within the Cdc42 zone for a later photoactivation (PA2 at wound; control $n = 10$, ITSN-C2 $n = 13$), within the Cdc42 zone for the latest photoactivation (PA3 at wound; control $n = 10$, ITSN-C2 $n = 12$), and outside the Cdc42 zone in wounded cells (PA off wound; control $n = 10$, ITSN-C2 $n = 13$). Two-sample $t$ tests were used to determine significance. (T) Correlation between $t_{1/2}$ to disappearance of photoactivation within the Cdc42 zone and time after wounding for control and ITSN-C2 conditions. No correlation was found for control ($P = 0.1523$), but a positive correlation was found for ITSN-C2 ($P = 0.0106$).

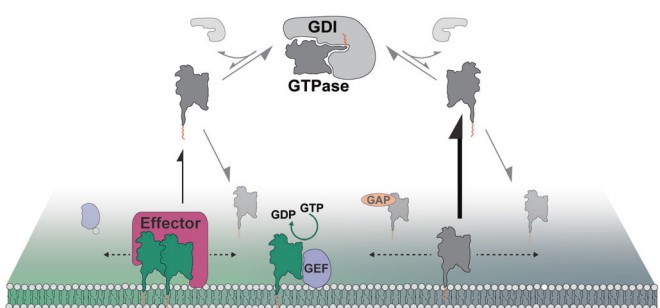

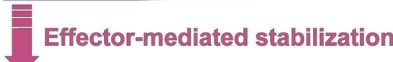

**Figure 9. Model figure.**

Model of route to Rho GTPase pattern formation.

et al, 2008), and (ii) a selective reduction in membrane mobility and turnover of active Cdc42 compared to its inactive form (Slaughter et al, 2009; Woods et al, 2016). This reduced mobility has mostly been attributed to the preferential extraction of inactive Cdc42 by GDI (Freisinger et al, 2013; Slaughter et al, 2009), though it may simply result from effector-mediated stabilization, as it has been observed even in the absence of RhoGDI (Bendezú et al, 2015; Woods et al, 2016). Importantly, Cdc42 activation in fission yeast leads to its enrichment at activity sites (Bendezú et al, 2015), mirroring our observations in Xenopus oocytes (Golding et al, 2019, Fig. 1). We therefore believe that observations in yeast are largely consistent with and in part mechanistically explained by our findings.

The results demonstrate that a primary driver of Rho GTPase pattern formation is stabilization of the active GTPases at the membrane and show that local GTPase activation alone cannot maintain a pattern in the face of lateral diffusion, a point previously made based on theoretical considerations (Bement et al, 2006). Rather, pattern establishment and maintenance requires a combination of both GAPs and effectors. The potential contributions of GAPs (Bement et al, 2006; Goryachev and Pokhilko, 2006) and effectors to Rho GTPase pattern formation has been inferred previously based on experiments performed in vivo; the reconstitution experiments presented here directly confirm the ability of GAPs and effectors to support pattern formation in a minimal system. Strikingly, both GAPs and effectors operate on active, rather than inactive GTPases, which accounts for the fact that pattern formation follows activation.

Are these results relevant to other members of the Ras superfamily of GTPases? We think it likely. For example, our findings are congruent with recent reconstitutions of Rab GTPase signaling systems, which suggest a similar order of biochemical events, in which RabGDI dissociation from the Rab GTPase precedes membrane association, which is followed by GEF-mediated activation (Bezeljak et al, 2020). This equivalence is remarkable given the evolutionary divergence of the two systems and lack of similarity in structure between Rho and Rab GDIs in particular.

The enrichment of active Rho GTPases at membranes we reconstituted in vitro (1.4-fold, Fig. 6) is slightly less pronounced than observed in cells (2-fold, Fig. 1). This is likely due to the absence of cytoskeletal assembly and remodeling downstream of effectors. Supramolecular structures such as membrane-proximal actin networks should be able to strongly enhance Rho GTPase enrichment via effector-mediated stabilization and immobilization. Another plausible factor for local Rho GTPase enrichment missing in our reconstitution is their potential clustering via anionic phospholipids in nanodomains (Remorino et al, 2017), which has been proposed to serve a scaffolding role (Budnar et al, 2019; Meca et al, 2019). Such nanoscale clustering likely requires actin-driven advection of phospholipids (Garcia-Parajo and Mayor, 2024), which is challenging to recapitulate in vitro.

Importantly, all of these stabilizing and immobilizing effects occur as a consequence of Rho GTPase activation and result from the association of effector proteins, as first suggested from in vivo observations (Budnar et al, 2019). Activity-induced stabilization of Rho GTPases on membranes critically rely on specific biochemical features such as multivalency, weak membrane binding and potentially actin association of effector proteins. Multivalency in

particular has been previously identified as a critical determinant of robustly detecting endogenous Rho GTPase activity by localization-based sensors (Mahlandt et al, 2021), but should also be expected to affect Rho GTPase turnover itself as we demonstrate here. Other biochemical determinants of effectors, such as the affinity for GTPases, which is known to vary considerably among different effectors (Bishop and Hall, 2000), or their affinity for other membrane components, can also be expected to influence the degree of membrane retention. Generally, increased affinity should decrease membrane off-rate more significantly, depending on which interaction is kinetically limiting. Experimentally exploring how distinct effector properties shape Rho GTPase signaling responses offers an exciting avenue for future research.

What is the significance of Rho GTPases concentrating at membrane sites of their activity? Despite the enrichment being moderate in magnitude, we believe that it constitutes an important feature of Rho GTPase signaling systems because of the following. First, many Rho GTPase effectors likely exert their function in a manner that depends non-linearly on their membrane density (De Seze et al, 2023; Sun et al, 2017). Such switch-like behavior renders the local Rho GTPase density an important control parameter for effector signaling output. More broadly, theoretical models demonstrate that the self-organization of Rho GTPase activity into spatial patterns necessitates feedback of higher-order non-linearities (Goryachev and Leda, 2017). This means that simple positive feedback via GTPase-mediated GEF recruitment alone is insufficient for pattern formation (Lo et al, 2014). The mechanistic requirements for autonomous patterning are most directly met by coupling of the spatial and catalytic cycle of Rho GTPases. It is this coupling, which enables the establishment of mutually exclusive zones of high activity resulting in high membrane stability/low diffusion and low activity with low stability/high lateral mobility. We demonstrate here how such coupling can be mechanistically realized in a straightforward manner. This opens exciting opportunities for reconstituting and dissecting the self-organized patterning of Rho GTPases in the future.

# Methods

### Reagents and tools table

| Reagent/resource | Reference or source | Identifier or catalog number |
|---|---|---|
| **Experimental models** | | |
| *Xenopus laevis* stage VI immature oocytes | *Xenopus* 1, Michigan The National *Xenopus* Resource, Marine Biological Laboratory, Massachusetts | |
| **Recombinant DNA** | | |
| pETM60-10xHis-NusA-TEV-GGGG-2xwCRIB | This study | |
| pETM60-10xHis-NusA-TEV-GGGG-wCRIB | This study | |
| pETM66-NusA-His-TEV-SortaseA(delta59) | This study | |
| pETMSumo3-10xHis-TEV-Sumo3-GGGGG-OPHN1$_{cat}$ | This study | |

| Reagent/resource | Reference or source | Identifier or catalog number |
|---|---|---|
| pETMSumo3-10xHis-TEV-Sumo3-GGGGG-PLCδ(PH)-ITSN$_{cat}$ | This study | |
| pETMSumo3-6xHis-TEV-Sumo3-GGGGG-PLCδ(PH) | Hansen Lab, Oregon | |
| pETMSumoH10-His10-Sumo3-GGGGG-ITSN(DHPH 1229-1577) | This study | |
| pETMz2-6xHis-zTag-TEV-Cdc42(wt) | This study | |
| pETMz2-6xHis-zTag-TEV-Cys-RhoGDI1(BT,wt) | This study | |
| pETMZ2-His6-zTag-TEV-Cdc42 | This study | |
| pETMZ2-RhoGDI1wt | This study | |
| pFastBac1-6xHis-MBP-10xAsn-TEV-GGGG-PI4P5K | Hansen Lab, Oregon | |
| pFB-6xHis-TEV-NWASP | This study | |
| pMal-C2-MBP-TEV-His6-TEV(Protein,S129V)-Arg6 | Hansen Lab, Oregon | |
| pMPB-6xHis-MBP-10xAsn-TEV-GGGG-DrrA-GGGG-OCRL | Hansen Lab, Oregon | |
| pCS2_mCherry-wGBD | Benink and Bement, 2005; Moe et al, 2021 | |
| pCS2_IT-GFP-Cdc42 | Golding et al, 2019 | |
| pCS2_IT-PAGFP-Cdc42 | This study | |
| pCS2_RhoGDI | Golding et al, 2019 | |
| pCS2_ITSN1 DHPH domain-PKCβ C2 domain | This study | |
| pCS2_IT-GFP-RhoA | Golding et al, 2019 | |
| pCS2_dTomato-rGBD | This study | |
| **Antibodies** | | |
| **Oligonucleotides and other sequence-based reagents** | | |
| **Chemicals, enzymes and other reagents** | | |
| 1,2-dioleoyl-sn-glycero-3-[(N-(5-amino-1-carboxypentyl) iminodiacetic acid)succinyl] 18:1 DGS-NTA(Ni) | Avanti Polar lipids (Sigma-Aldrich (München)) | 790404 |
| 1,2-dioleoyl-sn-glycero-3-phospho-(1'-myo-inositol-3',4',5'-trisphosphate) (PI(3,4,5)P3 | Avanti Polar lipids (Sigma-Aldrich (München) | 850156 |
| 1,2-dioleoyl-sn-glycero-3-phosphoethanolamine-N-[methoxy(polyethylene glycol)-5000] 18:1 PEG5000 PE | Avanti Polar lipids (Sigma-Aldrich (München) | 880230 |
| 1,2-dioleoyl-sn-glycero-3-phospho-(1'-myo-inositol-4',5'-bisphosphate) (PI(4,5) P2) | Avanti Polar lipids (Sigma-Aldrich (München) | 850155 |
| 1,2-dioleoyl-sn-glycero-3-phosphoethanolamine-N-(lissamine rhodamine B sulfonyl) (Rhod PE) | Avanti Polar lipids (Sigma-Aldrich (München) | 810150 |

| Reagent/resource | Reference or source | Identifier or catalog number |
|---|---|---|
| Alexa Fluor® 647-Maleimide (AF647) | Thermo Fisher Scientific (München) | A20347 |
| Blocker Casein (β-Casein) | Thermo Fisher Scientific (München) | 37528 |
| Catalase | Sigma-Aldrich (München) | C1345 |
| Chloroform | Sigma-Aldrich (München) | 650498 |
| Cholesterol (ovine) | Avanti Polar lipids (Sigma-Aldrich (München)) | 700000 |
| EDA-GDP-Cy5 | Jena Bioscience (Jena) | NU-840-Cy5 |
| EDTA-Free cOmplete™ protease inhibitor tablets | Roche (Basel) – Merck (Darmstadt) | 4693132001 |
| Geranylgeranyl pyrophosphate ammonium salt | Sigma-Aldrich (München) | G6025 |
| Glucose oxidase | Serva (Heidelberg) | 22778 |
| Hellmanex III | Hellma GmbH (Müllheim) | 9-307-011-4-507 |
| L-α-phosphatidylcholine (PC, soy) | Avanti Polar lipids (Sigma-Aldrich (München) | 441601 |
| L-α-phosphatidylinositol-4,5-bisphosphate (Brain, Porcine) (PI(4,5)P$_2$ | Avanti Polar lipids (Sigma-Aldrich (München)) | 840046 |
| Leupeptin | Serva (Heidelberg) | 51867 |
| PBS | Corning (Manassas, VA, USA) | 46-013-CM |
| Pepstatin | Roth (Gosen, IN, USA) | 2936.1 |
| Phosphocreatine | Sigma-Aldrich (München) | P7936 |
| Sulpho-Cy3-NHS | Lumiprobe (Hannover) | 11320 |
| Atto488-Maleimide (Atto488) | ATTO-TEC (Siegen) | AD488-41 |
| AF488-NHS ester | Lumiprobe (Hannover) | 11820 |
| AF647-NHS ester | Lumiprobe (Hannover) | 16820 |
| 1,2-dioleoyl-sn-glycero-3-phospho-L-serine 18:1 PS (DOPS) | Avanti Polar lipids (Sigma-Aldrich (München)) | 840035 |
| 1,2-dioleoyl-sn-glycero-3-phosphoethanolamine 18:1 (Δ9-Cis) PE (DOPE) | Avanti Polar lipids (Sigma-Aldrich (München)) | 850725 |
| Hydrogen peroxide (30%) | Roth (Gosen, IN, USA) | 8070.2 |
| Phosphatase inhibitor cocktail 2 | Sigma-Aldrich (München) | P5726 |
| 1,2-dioleoyl-sn-glycero-3-phospho-(1'-myo-inositol-4'-phosphate) (PI(4)P) | Avanti Polar lipids (Sigma-Aldrich (München)) | 850151 |
| Phosphatase inhibitor cocktail 3 | Sigma-Aldrich (München) | P0044 |
| Type 1 Collagenase | Gibco/ThermoFisher Scientific (Waltham, MA, USA) | 17100017 |
| **Software** | | |
| Adobe Illustrator | Adobe | |
| ApE | Davis and Jorgensen, 2022 | |
| CodonCode aligner | CodonCode corporation | |

| Reagent/resource | Reference or source | Identifier or catalog number |
|---|---|---|
| Expasy-ProtParam | SIB, Swis | |
| Fiji | Schindelin et al, 2012 | |
| ImageJ | Schneider et al, 2012 | |
| OligoCalc | Kibbe, 2007 | |
| OriginPro | OriginLab Corporation | |
| PDB | RSCB | |
| PyMol | Schrödinger, LLC | |
| Python | Python software foundation | |
| R | R Core Team | |
| SnapGene | Insightful science | |
| TrackMate | Tinevez et al, 2017 | |
| UCSF Chimera | Pettersen et al, 2004 | |
| Uniprot | EMBL-EBI | |
| **Other** | | |
| Coverslips for sticky slides – Glass 1.5H | Ibidi GmbH | 10812 |
| HiPrep 26/10 column | GE Healthcare | GE17-5087-01 |
| HisTrap Excel | GE Healthcare | 17-3712-06 |
| HiTrap chelating column | GE Healthcare | GE17-0409-03 |
| LM10 Microfluidizer | Hyland Scientific | N/A |
| MonoQ 5/50 GL | Cytiva | 17-5166-01 |
| MonoS 5/50 GL | Cytiva | 17-5168-01 |
| Pierce™ Detergent removal columns | Thermo Fisher Scientific (München) | 69705 |
| RC900 Rotary evaporator | KNF | S-328 |
| SC920G Diaphragm pump | KNF | S-327 |
| Sticky slides VI 0.4 | Ibidi GmbH | 80608 |
| Superdex 200 XK 16/60 | GE Healthcare | GE28-9893-35 |
| Superdex 200 XK 26/60 | GE Healthcare | GE28-9893-36 |
| Superdex 75 10/300 GL increase | GE Healthcare | GE29-1487-21 |
| Superdex 75 XK 16/60 | GE Healthcare | GE28-9893-33 |
| Vivaspin 15 R - 30000 MWCO | Sartorius | VS15RH21 |
| Vivaspin 15R - 5000 MWCO | Sartorius | VS15RH11 |
| Vivaspin 500 – 10000 MWCO | Sartorius | VS0102 |
| Vivaspin 6 - 30000 MWCO | Sartorius | VS0621 |
| Vivaspin 6 - 5000 MWCO | Sartorius | VS0611 |
| Zeba Spin column | Thermo Fisher | 89883 |
| PLI-100 Medical Systems Pico-Injector | Harvard Apparatus | MA1 65-0001 |

## Generation of in vivo constructs

### Cloning and mRNA synthesis

mCherry-wGBD and mTomato-rGBD were created as previously described (Benink and Bement, 2005; Moe et al, 2021). IT-GFP-Cdc42, IT-GFP-RhoA and RhoGDI were created as previously described (Golding et al, 2019). IT-PAGFP-Cdc42 was made with photoactivatable GFP (Patterson and Lippincott-Schwartz, 2002), which was inserted to internally tag Cdc42 as previously described (Golding et al, 2019). ITSN-C2 was created by fusing the DHPH domain of human ITSN1 to the C2 domain of PKCβ, similar to a C2 fusion construct used in (Moe et al, 2021). mRNA was generated using the mMESSAGE mMACHINE SP6 Transcription Kit (Ambion) and purified using the RNeasy MinElute Cleanup Kit (QIAGEN). IT-PAGFP-Cdc42 mRNA was polyadenylated via the Poly(A) Tailing Kit (Invitrogen) and cleaned up via chloroform:-phenol:isoamyl alcohol extraction. mRNA transcripts were run on a 1% agarose-formaldehyde denaturing gel with Millenium Marker and mRNA standards with known concentrations, used to quantify the mRNA via linear forecasting.

## Expression and purifications of recombinant protein from E. Coli

### Rho GTPases, RhoGDI1, GGTaseI and RabGGTase beta

For bacterial expression, coding sequences for human Cdc42 (Uniprot: A0A494C1M1) and human Rac1 (Uniprot: P63000) were ordered as codon-optimized GeneStrands (Eurofins) with and N-terminal 5xglycine-tag for labeling by Sortase-mediated peptide ligation and cloned into a pETMz2 vector containing a TEV cleavable His6-z-tag.

Rho GTPases were expressed in BL21 (DE3) E. coli cells induced with 1 mM IPTG at 37 °C for 4 hr. Cells were harvested after expression, centrifuged at $4000 \times g$ for 20 min, and pellets were flash-frozen in liquid nitrogen and stored at −80 °C. Frozen pellets were resuspended in a $3 \times$ volume of lysis buffer (100 mM KPi pH 7.4, 400 mM NaCl, 5 mM $MgCl_2$, 0.1 mM GDP, 0.5 mM β-mercaptoethanol, 1 mM PMSF and ~10 μg/mL DNAseI) and lysed with a high pressure homogenizer at 4 °C. Lysate was clarified by centrifugation at $100,000 \times g$ for 1 h. Affinity purification of the clarified lysate was performed on a 5 ml HiTrap Chelating HP column loaded with cobalt and equilibrated in lysis buffer. Protein was eluted in lysis buffer containing 500 mM imidazole, peak fractions were pooled and the His6-z-tag was removed by TEV cleavage at 4 °C overnight. The cleaved protein was desalted into lysis buffer via HiPrep 26/10 desalting column and recirculated over 5 ml HiTrap Chelating HP to remove His-TEV and the cleaved His-z-tag. The flow through was pooled, concentrated and gel filtered in storage buffer (50 mM HEPES pH 7.5, 50 mM NaCl, 2 mM $MgCl_2$, 2 mM DTT, 10% glycerol) over a Superdex 75 16/60 column. Peak fractions were pooled, concentrated and glycerol was added to 20%. Proteins were SNAP-frozen in liquid nitrogen and stored at −80 °C.

Bovine RhoGDI1 (Uniprot: P19803) was purified by a series of GST-affinity, reverse affinity and gel filtration steps as described in (Golding et al, 2019). GGTaseI consisting of the farnesyltransfer-ase/geranylgeranyltransferase type-1 subunit alpha (Uniprot: Q04631) and the geranylgeranyl transferase type-1 subunit beta (Uniprot: P53610) from R. norvegicus was purified by a series of

IMAC and gel filtration steps as described in (Gavriljuk et al, 2013). Geranylgeranyl transferase type-2 subunit beta from R. norvegicus (RabGGTase beta, Uniprot: Q08603) was purified by a series of IMAC and gel filtration steps as described in (Gavriljuk et al, 2013).

## ITSN$_{cat}$-His$_{10}$ and ITSN$_{cat}$-2xC2

The catalytic GEF module consisting of the tandem DH-PH domains (aa:1229-1576) of human ITSN (Uniprot: Q15811) was synthesized as a codon-optimized GeneStrand (Eurofins) with an N-terminal 5xglycine-tag for labeling by Sortase-mediated peptide ligation and either a C-terminal His$_{10}$-tag, creating ITSN$_{cat}$-His$_{10}$, or a C-terminal 2xC2-tag consisting of a tandem phosphatidylserine-binding C2 (aa: 270–427) domain of bovine Lactadherin (Uniprot: Q95114), creating ITSN$_{cat}$-2xC2. Both GeneStrands were cloned into a pETMz2 vector containing a TEV cleavable His$_6$-z-tag via Gibson assembly. Chemically competent E. coli Rosetta (DE3) cells were transformed with either ITSN$_{cat}$ construct and induced overnight at 18 °C with 0.15 μM IPTG. Cells were harvested after expression, centrifuged at $4000 \times g$ for 20 min, and pellets were SNAP-frozen in liquid nitrogen and stored at −80 °C. Frozen pellets were resuspended in a $3 \times$ volume of lysis buffer (50 mM KPi pH 7.3, 400 mM NaCl, 0.5 mM βME, 2 mM PMSF, 15 mM benzamidine and ~10 μg/mL DNAseI) and lysed with a high pressure homogenizer at 4 °C. Lysate was clarified by centrifugation at $100,000 \times g$ for 1 h. Affinity purification of the clarified lysate was performed on a 5 ml HiTrap Chelating HP column loaded with cobalt and equilibrated in lysis buffer. Protein was eluted in a 60 ml gradient of 0–45% elution buffer (50 mM Kpi pH 7.3, 400 mM NaCl, 500 mM imidazole, 0.75 mM βME, 15 mM benzamidine) and fractions were pooled based on SDS-PAGE. Pooled fractions were cleaved over night with TEV. ITSN$_{cat}$-His$_{10}$ or ITSN$_{cat}$-2xC2 were then separated from TEV and the cleaved His$_6$-z-tag via gel filtration over a Superdex 200 16/60 column into storage buffer (20 mM HEPES 7.5, 150 mM NaCl, 0.5 mM TCEP). Peak fractions were pooled, concentrated and glycerol was added to 20%. Proteins were SNAP-frozen in liquid nitrogen and stored at −80 °C.

## OPHN1$_{cat}$

The catalytic Rho GAP domain (aa: 375–583) of murine Oligophrenin-1 (Uniprot: Q99J31) was synthesized as a codon-optimized GeneStrand (Eurofins) with an N-terminal 5xglycine-tag for labeling by Sortase-mediated peptide ligation were cloned into a pETMSumo3 vector containing a SenP2-cleavable His10-Sumo-tag via Gibson assembly. Chemically competent E. coli Rosetta (DE3) cells were transformed with OPHN1$_{cat}$ construct, induced with 0.25 mM IPTG overnight at 18 °C. Cells were harvested after expression, centrifuged at $4000 \times g$ for 20 min, and pellets SNAP-frozen in liquid nitrogen and stored at −80 °C. Frozen pellets were resuspended in a $3 \times$ volume of lysis buffer (100 mM KPi pH 7.4, 400 mM NaCl, 5 mM $MgCl_2$, 0.5 mM βME, 1 mM PMSF and ~10 μg/mL DNAseI) and lysed with a high pressure homogenizer at 4 °C. Lysate was clarified by centrifugation at $100,000 \times g$ for 1 h. Affinity purification of the clarified lysate was performed on a 5 ml HiTrap Chelating HP column loaded with cobalt and equilibrated in lysis buffer. Protein was eluted in lysis buffer containing 500 mM imidazole, peak fractions were pooled and the His10-Sumo3-tag

was removed by SenP2 cleavage at 4 °C overnight. The cleaved protein was desalted into lysis buffer via HiPrep 26/10 desalting column and recirculated over a 5 ml HiTrap Chelating HP column to remove His-SenP2 and the cleaved His-Sumo3-tag. The flow through was pooled, concentrated and gel filtered in storage buffer (50 mM HEPES pH 7.5, 50 mM NaCl, 2 mM MgCl$_2$, 2 mM DTT, 10% glycerol) over a Superdex 75 16/60 column. Peak fractions were pooled, concentrated, and glycerol was added to 20%. Proteins were SNAP-frozen in liquid nitrogen and stored at −80 °C.

## wCRIB and 2xwCRIB

The isolated CRIB domain (aa: 201–268) or the corresponding tandem CRIB domain separated by a GAPSGGGATAGAGGGGPAG linker of human WASP (Uniprot: P42768) was synthesized as codon-optimized GeneStrands (Eurofins) with an N-terminal 5xglycine-tag for labeling by Sortase-mediated peptide ligation and cloned into a pETM60 vector (EMBL Protein Expression and Purification Facility) containing a TEV-cleavable N-terminal NusA-tag. Chemically competent *E. coli* Star ™ pRare pACYC cells were transformed with wCRIB or 2xwCRIB constructs and induced with 50 µM IPTG and incubated overnight at 15 °C. Cells were harvested after expression, centrifuged at 3500 × g for 20 min, and pellets SNAP-frozen in liquid nitrogen and stored at −80 °C. Frozen pellets were resuspended in a 3 × volume of lysis buffer (50 mM HEPES pH 7.5, 100 mM NaCl, 50 mM arginine, 50 mM glutamic acid, 0.4 mM βME, 1 mM PMSF, 15 µg/mL Benzamidine, 10% Glycerol, ~10 µg/mL DNAse1, 1 EDTA free protease inhibitor tablet/100 mL). After cell lysis with a high-pressure homogenizer at 4 °C, the lysate was hard spun (142,000 × g, 4 °C, 1 h). Affinity purification of the clarified lysate was performed on a 5 ml HiTrap Chelating HP column loaded with cobalt and equilibrated in wash buffer (50 mM HEPES pH 7.5, 100 mM NaCl, 50 mM Arginine, 50 mM Glutamic acid, 0.4 mM βME, 15 µg/mL Benzamidine, 10% glycerol). Protein was eluted in wash buffer containing 500 mM imidazole, peak fractions were pooled and the His6-NusA-tag was removed by TEV cleavage at 4 °C overnight. The cleaved protein was desalted via HiPrep 26/10 desalting column equilibrated with MonoS low salt buffer (5 mM HEPES pH 7.5, 10 mM NaCl, 7.5 mM arginine, 7.5 mM glutamic acid, 0.4 mM βME, 10% glycerol). Desalted protein was subjected to cation exchange chromatography using a MonoS 15/50 GL column (GE Healthcare) and eluted over a 30 CV linear gradient of MonoS high salt buffer (5 mM HEPES pH 7.5, 750 mM NaCl, 7.5 mM arginine, 7.5 mM glutamic acid, 0.4 mM βME, 10% Glycerol). Pooled fractions were gel filtered via a HiLoad Superdex 75 16/600 column (GE Healthcare) equilibrated with storage buffer (50 mM HEPES pH 7.5, 100 mM NaCl, 50 mM Arginine, 50 mM Glutamic acid, 1 mM TCEP, 10% glycerol). Peak fractions were pooled, concentrated, final glycerol concentration increased to 20% and SNAP-frozen in liquid nitrogen and stored at −80 °C.

## DrrA-OCRL

The chimeric 5′phosphatase consists of the PI4P-binding domain (aa: 544-647) of DrrA/SidM from *L. pneumophila* (Uniprot: Q29ST3) fused via a GSGGGGGTS linker to the catalytic 5′ phosphatase domain (aa: 234-539) of phosphoinositide 5-phosphatase OCRL (Uniprot: A0A2R8YG38), which was cloned into a modified pET expression vector containing an N-terminal His6-MBP-Asn$_{10}$-TEV-Gly$_5$-tag. The protein was expressed and

purified as described in (Hansen et al, 2019). Briefly, BL21 (DE3) *E. coli* cells were transformed and induced with 0.1 mM IPTG overnight at 18 °C. Cells were harvested after expression, centrifuged at 4000 × g for 20 min, and pellets SNAP-frozen in liquid nitrogen and stored at −80 °C. Frozen pellets were resuspended in a 3 × volume of lysis buffer 50 mM NaPi pH 8.0, 300 mM NaCl, 0.4 mM βME, 1 mM PMSF, 100 µg/mL DNase) and lyzed using a microfluidizer. Lysate was clarified by centrifugation at 100,000 × g for 1 h. Affinity purification of the clarified lysate was performed on a 5 ml HiTrap Chelating HP column loaded with cobalt and equilibrated in lysis buffer. Protein was eluted in lysis buffer containing 500 mM imidazole, peak fractions were pooled and the His$_6$-MBP-Asn$_{10}$-tag was cleaved by TEV cleavage at 4 °C overnight. The cleaved protein was desalted into lysis buffer via a HiPrep 26/10 desalting column and recirculated over a 5 ml HiTrap Chelating HP column to remove His-TEV and the cleaved His$_6$-MBP-Asn$_{10}$-tag. Flow-through containing DrrA-OCRL was then buffer exchanged into MonoS buffer (20 mM HEPES pH 7, 100 mM NaCl, 1 mM DTT) using a HiPrep 26/10 desalting column and loaded onto a mL MonoS (5/50 GL) cation exchange column equilibrated in MonoS buffer. Protein was eluted by a 0–100% linear gradient (0.1–1 M NaCl). Peak fractions containing DrrA-OCRL were pooled, concentrated and gel filtered into storage buffer (20 mM Tris [pH 8.0], 200 mM NaCl, 10% glycerol, 1 mM TCEP) over a Superdex 75 10/300 GL column, pooled, concentrated and glycerol was added to 20%. Protein was SNAP-frozen in liquid nitrogen and stored at −80 °C.

## Sortase

The catalytic domain (aa: 60–206) from Staphylococcus aureus Sortase (Uniprot: Q2FV99) was cloned into a pETM66 vector containing an N-terminal His6-NusA-tag and expressed in BL21 (DE3) *E. coli* cells induced with 0.25 mM IPTG overnight at 18 °C. Cells were harvested after expression, centrifuged at 4000 × g for 20 min, and pellets SNAP-frozen in liquid nitrogen and stored at −80 °C. Frozen pellets were resuspended in a 3 × volume of lysis buffer (50 mM KPO$_4$ pH 7.5, 400 mM NaCl, 0.75 mM βME a 0.1–1 g DNase powder, 1 mM PMSF) and lysed with a high pressure homogenizer at 4 °C. Lysate was clarified by centrifugation at 100,000 × g for 1 h. Affinity purification of the clarified lysate was performed on a 5 ml HiTrap Chelating HP column loaded with cobalt and equilibrated in lysis buffer. Protein was eluted in lysis buffer containing 500 mM imidazole, peak fractions were pooled and gelfiltered over a Superdex 200 XK26/60 column into storage buffer (20 mM HEPES, pH 7.5, 150 mM KCl, 0.5 mM DTT, 10% v/v glycerol). Peak fractions were pooled, concentrated and glycerol was added to 20%. Protein was SNAP-frozen in liquid nitrogen and stored at −80 °C. SNAP-frozen in liquid nitrogen and stored at −80 °C.

## Expression and purification of recombinant protein from insect cells

### DOCK1:ELMO1 complex

Coding sequences of full length human DOCK1 (Uniprot: Q14185) and ELMO1 (Uniprot: Q92556) were amplified from corresponding cDNA clones and cloned via Gibson assembly into modified pLIB vectors (Weissmann et al, 2016), containing N-terminal 5xglycine-tag

for labeling by Sortase-mediated peptide ligation and either a TEV-cleavable StrepII- (ELMO1for DOCK1) or a TEV-cleavable His$_6$- (for ELMO1) tag and into DH10Bac-competent *E. coli* (Thermo Fisher). One positive clone per construct was selected by blue-white screening and used to generate Bacmid DNA. Sf-9 insect cells ($1 \times 10^6$ cells) were transfected with the purified bacmid DNA together with FuGene6 transfection reagent and incubated for 5 days at 27 °C. After two passages, the cell suspension was spun down (RT, $100 \times g$, 10 min), the supernatant was filtered, 10% FBS added and stored at 4 °C. The heterodimeric complex was expressed by co-infecting insect TnaO38 cells with viruses for both proteins for 3 days at 27 °C. Cells were harvested by centrifugation ($868 \times g$, 20 min, 18 °C), washed with $1 \times$ PBS pH 7.2, centrifuged (as above), and SNAP-frozen and stored at −80 °C. Frozen pellets were resuspended with a $5 \times$ pellet volume of lysis buffer (75 mM KPi pH 7.8, 300 mM NaCl, 10% glycerol, 5 mM βME, 0.015 mg/ml benzamidine, 1 mM PMSF, $1 \times$ complete protease inhibitor cocktail). After cell lysis by a microfluidizer the lysate was hard spun, filtered through a 0.45-μm syringe filter and recirculated over a 5 ml StrepTrap HP column. Fractions were pooled after SDS-PAGE and TEV-cleaved at 4 °C overnight. Proteins were gel filtered into storage buffer (50 mM HEPES (pH = 7.5), 150 mM KCl, 0.5 mM TCEP, 10% glycerol) over a Superose 6 column, pooled, concentrated and glycerol was added to 20%. Protein was SNAP-frozen in liquid nitrogen and stored at −80 °C.

### Full length N-WASP

Full length human N-WASP (Uniprot: O00401) was cloned from a cDNA clone into a modified pLib vector (Weissmann et al, 2016) containing a TEV-cleavable His$_6$-tag and transformed into DH10Bac-competent *E. coli* (Thermo Fisher). A positive clone was selected by blue-white screening and used to generate Bacmid DNA. Sf-9 insect cells ($1 \times 10^6$ cells) were transfected with the purified bacmid DNA together with FuGene6 transfection reagent and incubated for 5 days at 27 °C. After two passages, the cell suspension was spun down (RT, $100 \times g$, 10 min), the supernatant was filtered, 10% FBS added and stored at 4 °C. TnaO38 suspension culture ($c = 1 \times 10^6$ cells/mL) was infected with virus, incubated for 4 days and pelleted ($868 \times g$, 20 min, 18 °C). Pellets were resuspended in lysis buffer (25 mM Tris pH 7.5, 150 mM NaCl, 0.5 mM EDTA, 5 mM βME, 15 μg/mL Benzamidine, 10% glycerol, ~10 μg/mL DNAse1, 1 EDTA-free cOmplete protease inhibitor tablets/50 mL) and homogenized at 4 °C using a microfluidizer. Lysate was hard spun ($142,000 \times g$, 4 °C, 1 h) and recirculated over a HisTrap excel column equilibrated with wash buffer (25 mM Tris pH 7.5, 150 mM NaCl, 0.5 mM EDTA, 5 mM βME, 15 μg/mL Benzamidine, 10% glycerol) for 2-3 h. The protein was eluted over a 6 CV linear gradient (0–100%) of elution buffer (25 mM Tris pH 7.5, 150 mM NaCl, 5 mM βME, 500 mM imidazol, 10% glycerol). Tag cleavage was achieved by overnight TEV cleavage on ice and cleaved protein was gel filtered over a HiLoad Superdex 200 16/600 column equilibrated with storage buffer (25 mM Tris pH 7.5, 150 mM NaCl, 1 mM TCEP, 20% glycerol). The protein was passed through a HiPrep 26/10 Desalting column (GE-Healthcare) equilibrated with MonoS low salt (5 mM HEPES pH 7.5, 10 mM NaCl, 7.5 mM arginine, 7.5 mM glutamic acid, 0.4 mM βME, 10% Glycerol). Desalted protein was subjected to cation exchange chromatography using a MonoS 15/50 GL column (GE Healthcare) and eluted over a 30 CV linear gradient of MonoS high salt buffer (5 mM HEPES pH 7.5, 750 mM NaCl, 7.5 mM arginine, 7.5 mM glutamic acid, 0.4 mM βME, 10% glycerol). Peak fractions were pooled and buffer exchanged via Nap5 columns equilibrated in storage buffer prior to maleimide labeling.

### PIP5K

The kinase domain (aa: 1–421) of human Phosphatidylinositol 4-phosphate 5-kinase type-1 beta (Uniprot: O14986-3) in frame with a N-terminal his6-MBP-(Asn)10 -TEV-GGGGG fusion, cloned into a FastBac1 vector, was expressed and purified as essentially as described before (Hansen et al, 2019). After transformation into DH10Bac-competent *E. coli* (Thermo Fisher), a positive clone was selected by blue-white screening and used to generate Bacmid DNA. Sf-9 insect cells ($1 \times 10^6$ cells) were transfected with the purified bacmid DNA together with FuGene6 transfection reagent and incubated for 5 days at 27 °C. After two passages, the cell suspension was spun down (RT, $100 \times g$, 10 min), the supernatant was filtered, 10% FBS added and stored at 4 °C. TnaO38 suspension culture ($c = 1 \times 10^6$ cells/mL) was infected with virus and incubated for 2 days. Cells were harvested by centrifugation ($868 \times g$, 18 °C, 20 min), washed with $1 \times$ PBS pH 7.2, centrifuged, and SNAP-frozen and stored at −80 °C. Frozen pellets were resuspended with a $5 \times$ pellet volume of lysis buffer (50 mM NaPi pH 8.0, 10 mM imidazole, 400 mM NaCl, 1 mM PMSF, 5 mM βME, 100 μg/mL DNase, 1 Roche protease inhibitor cocktail EDTA free (Roche) per 100 mL lysis buffer, 0.25% v/v Phosphatase inhibitor cocktail 2 (Sigma) and 0.25% v/v Phosphatase inhibitor cocktail 3 (Sigma)) using a microfluidizer. Lysate was hard spun ($142,000 \times g$, 4 °C, 1 h) and recirculated over a Ni$^{2+}$-sepharose excel column equilibrated with wash buffer (50 mM NaPi pH 8.0, 30 mM imidazole, 400 mM NaCl, and 5 mM βME). After recirculation, the column was washed with wash buffer until baseline and eluted with wash buffer containing 500 mM imidazole. Peak fractions were pooled and TEV-cleaved at 4 °C overnight. Cleaved protein was desalted into MonoS buffer (20 mM Tris pH 8.0, 100 mM NaCl, 1 mM DTT, 50 μM ATP, 50 μM MgCl$_2$) via a HiPrep 26/10 Desalting column was then loaded onto a MonoS 5/50 GL cation exchange column equilibrated in MonoS Buffer. Proteins were eluted by a 0–100% linear gradient (0.1–1 M NaCl). Peak fractions containing PIP5Kb were pooled, concentrated and gelfiltered into storage buffer (20 mM Tris pH 8.0, 200 mM NaCl, 10% glycerol, 50 μM ATP, 50 μM MgCl$_2$ and 1 mM TCEP over a HiLoad Superdex 200 16/600 column, pooled, concentrated, SNAP-frozen and stored at −80 °C.

### Protein prenylation, labeling and complex formation

#### *In vitro prenylation of bacterially expressed Rho GTPases*
Rho GTPases were in vitro prenylated by mixing prenylation reactants in prenylation reaction buffer (50 mM HEPES pH 7.5, 50 mM NaCl, 2 mM MgCl$_2$, 2% w/v CHAPS and 2 mM DTT) in the following order: 10 μM ZnSO$_4$, 10% v/v glycerol, 57.9 μM Cdc42 or Rac1, 9 μM GGTase1, 148 μM geranylgeranyl pyrophosphate (GGPP) (Golding et al, 2019). The reaction was wrapped in aluminum foil to protect from light and was rotated on a disc rotator at 4 °C overnight. Geranlygeranylated Rho GTPases were separated from their non-prenylated counterparts and free

GGTaseI via gel filtration over a Superdex 75 16/60 column equilibrated in prenylation buffer with CHAPS concentration reduced to 0.5%. The peak fractions were collected, pooled and concentrated to <250 μL. Geranylgeranylated proteins were then taken into one or more of the following steps; nucleotide exchange, sortase mediated peptide ligation or complex formation with RhoGDI or RabGGTase beta.

## Loading of dye-nucleotide conjugates in Rho GTPases

Nucleotide exchange of the GTPase was completed by first destabilizing the native "dark" nucleotide via incubation of GTPase with 10-fold molar excess of EDTA and EDA-GDP-Cy5 or EDA-GDP-A647 for 30 min on ice. The EDA-GDP-Cy5 or EDA-GDP-A647 was then stabilized in the GTPase via the addition of 20-fold molar excess of $MgCl_2$ (Golding et al, 2019). Protein was passed over a Nap5 column equilibrated in storage buffer (50 mM HEPES pH 7.5, 50 mM NaCl, 2 mM $MgCl_2$, 2 mM DTT, 20% glycerol) and either SNAP-frozen and stored at −80 °C or carried forward to labeling or complex formation.

## Synthesis of dye-peptide or Biotin-dye-peptide conjugates for Sortase-mediated protein labeling

For Sortase-mediated peptide labeling, NHS-Dyes were first conjugated to either CLPETGG or LPETGG peptides (Genscript). 40 mM NHS-Dye, 20 mM peptide and 30 mM triethylamine were mixed in DMSO and incubated overnight in a mixing heat block in the dark at 42 °C, 750 rpm. The reaction was quenched by addition of Tris-Cl pH 8.0 to a final concentration of 100 mM and incubated for 12 h at room temperature. These dye-peptide conjugates were then either stored at −20 °C for use in Sortase-mediated labeling reactions or, in the case of dye-CLPETGG conjugates, additionally biotinylated via maleimide chemistry. For this purpose, C-dye-LPETGG conjugates were mixed with a 6-fold molar excess biotin-PEG11-maleimide in maleimide labeling buffer (50 mM HEPES pH 7.5, 50 mM NaCl and 2 mM $MgCl_2$). The reaction was incubated at 16 °C for 2 h. The reaction was quenched with 10 mM DTT and incubated overnight 16 °C. Biotin-dye-peptide conjugates were stored at −20 °C until further use in Sortase-mediated labeling reactions.

## Sortase-mediated protein labeling

Substrate protein containing the N-terminal penta-glycine motif (>250 μM) was mixed with dye-LPETGG or biotin-biotin-C-dye-LPETGG conjugates (10× molar excess conjugate over substrate), sortase (1/3 molar concentration of substrate protein), 6 mM $CaCl_2$, 107 mM Tris pH 8 and were mixed in the labeling buffer (150 mM KCl, 0.5 mM TCEP) and incubated overnight at 16 °C in a centrifuge at 16,000 × g. Labeled proteins were separated from free dye-peptide conjugates and sortase by gel-filtration over either Superdex 75, Superdex 200 or Superose 6 10/300 columns equilibrated storage buffer of the respective substrate protein (as described in "Methods"; protein purification). Peak fractions were pooled, concentrated and glycerol was added to 20%. Proteins were SNAP-frozen in liquid nitrogen and stored at −80 °C or carried forward to complex formation.

## Maleimide labeling

RhoGDI was mixed in maleimide labeling buffer buffer (50 mM HEPES pH 7.5, 50 mM NaCl and 2 mM $MgCl_2$) with a sixfold molar excess A647-maleimide. The reaction was incubated for 2 h at 16 °C spinning at 16,000 × g. DTT was added to a final concentration of 10 mM, incubated for 10 min on ice and the reaction mix was gel filtered over a Superdex 75 10/300 column in RhoGDI gel filtration buffer (50 mM HEPES pH 7.5, 50 mM NaCl, 2 mM $MgCl_2$, and 2 mM DTT). Labeled RhoGDI peak fractions were selected by via 280 and 650 nm absorbance. Peak fractions were pooled, concentrated and glycerol was added to 20%. Proteins were SNAP-frozen in liquid nitrogen and stored at −80 °C or carried forward to complex formation.

Full length N-WASP was mixed in maleimide labeling buffer buffer (50 mM HEPES pH 7.5, 50 mM NaCl and 2 mM $MgCl_2$) with a 7× molar excess of Atto488-maleimide and incubated overnight on ice. DTT was added to a final concentration of 5 mM, incubated for 10 min on ice and the reaction mix was gel filtered over a HiLoad Superdex 200 10/300 GL column equilibrated with storage buffer (25 mM Tris pH 7.5, 150 mM NaCl, 1 mM TCEP, 20% glycerol). Protein concentrated, SNAP-frozen and stored −80 °C.

Non-prenylatable Cdc42 (Cdc42-CVIL) was labeled as described for RhoGDI however Cy3-Maleimide was used in place of A647-maleimide. Labeled, non-prenylatable Cdc42 peak fractions were selected via 280 and 555 nm absorbance. Peak fractions were pooled, concentrated and glycerol was added to 20%. Proteins were SNAP-frozen in liquid nitrogen and stored at −80 °C or carried forward to complex formation.

## Formation and isolation of GTPase:GDI complexes

To form a stoichiometric complex, equimolar ratios of prenylated, labeled Rho GTPases were mixed with labeled RhoGDI and incubated on ice for 10 min to allow for complex formation. The complex was then gel filtered using a Superdex 75 10/300 column pre-equilibrated in gel filtration buffer (50 mM HEPES pH 7.5, 50 mM NaCl, 2 mM $MgCl_2$, and 2 mM DTT). The complex peak was identified by 280, 555 and 650 nm absorbance. Peak fractions were pooled and concentrated to 100 μL. Complete detergent removal was ensured utilizing a detergent removal kit (Pierce). The column was placed in a 2 mL Eppendorf tube and spun at 1500 × g for 1 min to remove the column storage buffer. In total, 400 μL of gel filtration buffer were added, and the columns spun as above and the flow through discarded. This was repeated an additional two times. 400 μL of gel filtration buffer supplemented with 0.5 mg/mL blocker casein were added to the column and incubated at room temperature for 2 min. Columns were spun at 1500 × g for 2 min and the flow through discarded. Washing with gel filtration buffer was repeated three times as above and the columns transferred to a new 1.5 mL low-protein-binding Eppendorf tube. In all, 100 μL of GTPase:GDI complex were added and incubated at room temperature in the dark for 2 min. Columns were then spun for 2 min at 1500 × g and the flow-through was collected. The flow-through was then passed over a second detergent removal column prepared as described above. 20% glycerol (v/v) was added to the flow-through and was SNAP-frozen and stored at −80 °C.

## Preparation of oocytes

### Handling of oocytes

Ovaries were surgically removed from adult *Xenopus laevis* frogs via a protocol approved by the University of Wisconsin–Madison Institutional Animal Care and Use Committee. Ovaries were incubated in $1\times$ Barth's solution (88 mM NaCl, 1 mM KCl, 2.4 mM NaHCO$_3$, 0.82 mM MgSO$_4$, 0.33 mM Ca(NO$_3$)$_2$, 0.68 mM CaCl$_2$, 10 mM HEPES, with additional 25ug/ml ampicillin, 6ug/ml tetracycline, and 50ug/ml gentamicin sulfate, pH 7.4. Ovaries were manually cut into small clumps, washed with $1\times$ Barth's, and incubated in $1\times$ Barth's with 8 mg/ml type I collagenase for 1 h at 16 °C on a rotating plate (60 rpm). Dissociated oocytes were washed with $1\times$ Barth's and incubated in 1x Barth's at 16 °C. At least 2 h post-collagenase treatment, stage VI immature oocytes were separated and the follicle cell layer surrounding the oocytes was manually removed with forceps. Defolliculated oocytes were stored in $1\times$ Barth's at 16 °C until injection.

## Preparation of phospholipids, LUVs and SUVs

### Handling of phospholipids

All lipids used in this work were purchased from Avanti® Polar Lipids. The following lipids were used: 1,2-dioleoyl-sn-glycero-3-phosphocholine (18:1 (Δ9-Cis) PC (DOPC)), L-α-phosphatidylcholine (Soy PC (95%)), 1,2-dioleoyl-sn-glycero-3-phosphoethanolamine (18:1 (Δ9-Cis) PE (DOPE)), 1,2-dioleoyl-sn-glycero-3-phospho-L-serine (sodium salt) (18:1 PS (DOPS)), 1,2-dioleoyl-sn-glycero-3-phospho-(1′-myo-inositol-4′-phosphate) (ammonium salt) (synthetic PI(4)P; Avanti; 850151), 1,2-dioleoyl-sn-glycero-3-phospho-(1′-myo-inositol-4′,5′-bisphosphate) (ammonium salt) (synthetic PI(4.5)P$_2$; Avanti; 850155), L-α-phosphatidylinositol-4,5-bisphosphate (Brain PI(4,5)P$_2$), 1,2-dioleoyl-sn-glycero-3-phospho-(1′-myo-inositol-3′,4′,5′-trisphosphate) (PI(3,4,5)P$_3$; Avanti; 850156), 1,2-dioleoyl-sn-glycero-3-phosphoethanolamine- N-[methoxy(polyethylene glycol)-5000] (18:1 PEG5000 PE), Cholesterol (ovine, ovine wool >98%), 1,2-dioleoyl-sn-glycero-3-[(N-(5-amino-1-carboxypentyl)iminodiacetic acid)succinyl] (18:1 DGS-NTA(Ni)). All phospholipids were purchased from Avanti as either stock solutions in chloroform or chloroform:methanol:water:ammonia (45:35:7.7:2.3, v/v/v/v), or powder, and stored at -20 °C. Powdered lipids were resuspended fresh prior to use in chloroform (Soy PC, cholesterol) or chloroform:methanol:water (20:9:1, v/v/v) (PI(3,4,5)P$_3$, synthetic PI(4)P and synthetic PI(4,5)P$_2$). Any water used in the preparation of lipids was ultrapure generated by the Milli-Q IQ7003 with Q-POD dispenser and the LC-Pak (Merck Milli-Q).

## Description of phospholipid compositions

Throughout this paper we utilize a mixture of phospholipids (30.5% DOPC, 26.5% DOPE, 22.5% DOPS, 0.5% PEG5000-PE and 20% cholesterol, with % being molar percent) that we refer to as "PM lipids". This composition was chosen to mimic the inner leaflet of the plasma membrane (Lorent et al, 2020). The introduction of other additional phospholipids necessary for some experiments is detailed below and in the corresponding figure legends and has always been done by reducing the molar fraction of PC (Soy/DOPC) accordingly. For bulk GEF experiments in the presence of ITSN$_{cat}$-His$_{10}$, 0.2% Ni$^{2+}$-NTA-DGS was used in PM lipids. For bulk GEF experiments in the presence of DOCK1:-ELMO1, 4% PtdIns(3,4,5)P$_3$ was used in PM lipids. For single molecule recruitment experiments on SLBs in the presence of ITSN$_{cat}$-His$_{10}$, 2% Ni$^{2+}$-NTA-DGS was used in PM lipids for buffer based experiments, and 0.25% for cell lysate experiments. Single molecule GEF experiments in the presence of DOCK1:ELMO1, 4% PtdIns(3,4,5)P$_3$ was used in PM lipids. All single molecule Cdc42 experiments without spatially patterned SLBs were done in the presence of 4% PI(4,5)P$_2$. All experiments on spatially patterned SLBs by the enzymatic kinase-phosphatase system contained 2% PI(4,5)P$_2$ and 2% PI(4)P in PM lipids at the beginning of the experiment.

## Glassware for preparation of lipids

All glassware involved in lipid preparation was submerged in 5% Hellmanex III for 45 min at 60 °C in a dry oven. Glassware was rinsed five times in ultrapure water, and submerged in 50% 2-propanol:50% ultrapure water and sonicated for 30 min. After sonication, glassware was rinsed five times with ultrapure water and placed back in the dry oven at 60 °C to completely dry. Round-bottom flasks were washed as described above, however, after the final rinsing, both the flasks and stoppers were submerged in piranha solution (5 parts sulfuric acid to 3 parts hydrogen peroxide 30%)) for ≥90 min. The flasks were rinsed with ultrapure water as above and placed in a clean beaker in the 60 °C dry oven to dry overnight.

## Preparation of LUVs and SUVs

A total amount of 1 mM of the desired lipid species were mixed from their stock solutions in a total volume of 1 ml of chloroform in round-bottom flask and attached to a rotary evaporator. Lipids were dried to a thin film under vacuum rotating at 110 rpm in a 35 °C water bath. Once the solvent completely evaporated, the flasks were placed in a desiccator under vacuum for 30 min. Dried lipids were resuspended to a final concentration of 1 mM in PBS by gentle vortexing for 2 min.

For the preparation of LUVs, the resuspended lipid mixture was transferred to cryovials and snap frozen in liquid nitrogen. Once completely frozen, lipids were rapidly thawed in a water bath at 37 °C, this freeze thaw cycle was repeated a total of 7 times. LUVs were then extruded with a 100 nm filter 11 times and stored at 4 °C.

For the preparation of SUVs, the resuspended lipid mixture was extruded with a 30 nm filter 11 times and stored at 4 °C.

## Preparation of cell lysate

Cell lysate was prepared as described in (Hume et al, 2014). Briefly, fresh pig brains were placed in ice cold PBS and stored on ice. Brains were peeled to remove Meninges membrane and attached blood vessels of both the brain hemispheres as well as the cerebellum. Around 200 g of brain were placed into a blender along with 200 mL of extraction buffer (20 mM HEPES pH 7.4, 100 mM KCl, 1 mM EGTA, 0.1 mM EDTA, 2 mM MgCl$_2$ 2.5 mM DTT, 0.5 mM ATP, 10 mM Leupetin, 10 mM Pepstatin, 10 mM Chymostatin, cOmplete protease inhibitor tablets (1 per 50 mL) and phosphatase inhibitor cocktail 2 + 3, 4 µL per mL). The brains

were blended on the lowest setting for 1 min and then the blender container was placed in ice for 3 min. Following this, the container was placed back on the blender and switched to high powers for 40 s then quickly placed back on ice for 3 min. This was repeated two more times. Following blending, fresh DTT was added to a final concentration of 2.5 mM and the slurry was spun at $20,000 \times g$, 4 °C for 30 min. The supernatant was decanted through a nylon cheesecloth filter into a clean beaker (on ice). The supernatant was then spun once more at $142,000 \times g$ at 4 °C for 45 min. The SN was then added to 10,000 MWCO dialysis tubing and dialyzed against extraction buffer (20 mM HEPES pH 7.4, 100 mM KCl, 1 mM EGTA, 0.1 mM EGTA, 2 mM MgCl$_2$ 2.5 mM DTT, 0.5 mM ATP) containing 50% PEG MN 20,000 while stirring continuously at 4 °C to a final volume of around 50 mL. Lysate was SNAP frozen in liquid nitrogen and stored at −70 °C. For experiments, lysate was supplemented using 20× energy (300 mM phosphocreatine, 40 mM ATP, 40 mM MgCl$_2$) and 20× salt mixes (1.2 M KCl, 100 mM EGTA, 400 mM 3-phosphogycerate).

## In vivo assays

### Microinjection of oocytes

mRNA was injected approximately 16 h before imaging and protein was injected approximately 1 h before imaging. Both mRNA and protein were injected using a PLI-100 Medical Systems Pico-Injector equipped with a microneedle calibrated to inject a 40 nL volume. mRNA was used at the following concentrations: mCherry-wGBD (33.3 ng/μl), RhoGDI (62.5 ng/μl), IT-GFP-Cdc42 (62.5 ng/μl), polyadenylated IT-PAGFP-Cdc42 (125 ng/μl) ITSN-C2 (125 ng/ul). In vitro prenylated Cy3-Cdc42:RhoGDI protein and non-prenylatable Cy3-Cdc42:RhoGDI protein were both used at a 4.56 μM labeled concentration.

## Bulk biochemical assays

### GEF assays

GEF reactions were prepared by separately preparing (1) 2× GEF mix and (2) 2× GTPase mix. For the 2× GEF mix, ITSN$_{cat}$-His$_{10}$ or DOCK1:ELMO1 GEFs were diluted to 2× working concentration as stated in the figure legends (generally 20 nM) in GEF Buffer (50 mM HEPES pH 7.5, 50 mM NaCl, 2 mM MgCl$_2$, 10 mg/mL blocker casein, 1 mM βME, 0.2 mM GDP) in the presence or absence of PM-lipid LUVs (250 μM total lipids), which contained either 0.25% Ni$^{2+}$-NTA-DGS or 4% PtdIns(3,4,5)P$_3$ for membrane recruitment of GEF, which was carried out at RT for 5 min. For the 2x GTPase mix, either dual labeled GTPase:GDI complexes (Cy5-GDP:Cy3-Cdc42:RhoGDI1 or Cy5-GDP:Cy3-Rac1:RhoGDI1) or dual-labeled free GTPases (Cy5-GDP:Cy3-Cdc42 or Cy5-GDP:Cy3-Rac1) were diluted to 200 nM in GEF buffer. The reaction was then immediately initiated by mixing equal volumes of 2× GEF mix with 2× GTPase mix in a 96-well plate. Ratiometric FRET was synchronously measured at 10 s intervals by exciting at 554 nm and measuring emission at 610 and 667 nm via a Tecan Spark fluorescence plate reader.

## FRET-based RhoGDI dissociation assays

FRET-based RhoGDI dissociation assays were essentially performed as described in (Medina Gomez et al, 2024). 80 nM of Cy3-Cdc42 was complexed in situ with 120 nM of A647-RhoGDI1 in GEF buffer (50 mM HEPES pH 7.5, 50 mM NaCl, 2 mM MgCl$_2$, 10 mg/mL blocker casein, 1 mM βME) in a quartz cuvette. Ratiometric FRET was measured by exciting at 550 nm and measuring emission at 563 and 665 nm via a PTI photospectrophotometer (Photon Technology International, Birmingham NJ, USA). Once a baseline was established, 5 μM unlabeled RhoGDI was added and the FRET signal was measured at intervals of 2 s for 20,000 s.

## TIRF microscopy-based assays

### Chamber and supported lipid bilayer preparation

Coverslips (Coverslips for sticky-Slides 1.5H Glass, Ibidi) were incubated in 5% Hellmanex III at 60 °C in the dry oven for 45 min. They were then rinsed five times with ultrapure water and submerged in piranha solution (5 parts sulfuric acid to 3 parts hydrogen peroxide 30%)) overnight. Coverslips were rinsed with ultrapure water and dried with an argon line. Sticky slides VI (Ibidi) were plasma cleaned for 3 min and stuck to the coverslips by applying pressure with a Q-tip. These "chambers" were placed on a pre-heated metal block at 45 °C for 2.5 min, lipid mixture was added and incubated for additional 2.5 min. NaCl (500 mM final concentration) was flushed into the chamber and pipetted gently from each end to ensure mixing. Chambers were then incubated for 30 min at 45 °C. Following incubation, chambers were washed, each with 5 mL of PBS followed by 1 mL of GEF buffer (50 mM HEPES pH 7.5, 50 mM NaCl, 2 mM MgCl$_2$) or kinase buffer (20 mM HEPES pH 8.0, 150 mM NaCl, 5 mM MgCl$_2$, 0.5 mM EGTA)). The heat block was switched off and allowed to cool to RT over several hours.

## Membrane fluidity

Membrane fluidity was assessed via FRAP (see "Methods"; "Data analysis"). Supported lipid bilayers were prepared as described above with a lipid composition of 26.49% DOPC, 26.5% DOPE, 22.5% DOPS, 2% PI(4)P, 2% PI(4,5)P2, 0.5% PEG5000, 20% cholesterol and 0.01% 18:1 Lissamine rhodamine-PE (molar fraction). TIRF images were captured at 1-s time intervals for 5 s to acquire a baseline. SLBs were photobleached for 3 s in a defined area at 100% laser power using the Visitron system (Visitron systems). Imaging was continued as above for a total of 2 min.

## Biotin-mediated recruitment controls

Functionalized glass coverslips were prepared as described in (Bieling et al, 2010). This technique whereby glass is covalently modified with a dense, protein-blocking PEG polymer brush allows for subsequent covalent coupling of functional chemical groups. In our case, this resulted in a glass surface functionalized with PLL-PEG-biotin. Counter slides were plasma cleaned for 3 min, and the coverslip was placed with the functionalized side down and secured with double-sided tape to the counter slide. The chambers were then passivated using PLL-PEG with a grafting ratio of 3.5 with a final concentration of ~25 mg/mL. The PLL-PEG passivation agent was diluted tenfold in MilliQ water, flowed into the chamber, incubated at room temperature for 3 min and washed with 100 μL MilliQ water.

In all, 1 nM of Neutravidin in GEF buffer (50 mM HEPES pH 7.5, 50 mM NaCl, 2 mM MgCl$_2$) was flushed into the chamber and incubated at room temperature for 2 min. Chambers were washed with 200 μL GEF buffer once more and the reaction mix, containing 100 pM Cy3-Cdc42:RhoGDI-AF647, 200 μM nucleotide (GDP or GTP) and 10 % (v/v) oxygen scavenger system (3.75 mg/mL glucose oxidase, 3.75 mg/mL catalase, 40 mg/mL glucose) was added to the chamber and imaged). Dual color single molecule TIRF movies were captured at 22 ms exposure, continuous illumination for 40 s.

## Single molecule TIRFM GTPase recruitment assays

Supported lipid bilayers (as described above) were washed with 1 mL of fresh GEF buffer (50 mM HEPES pH 7.5, 50 mM NaCl, 2 mM MgCl$_2$). For +ITSN$_{cat}$-H10 experiments, 400 nM of ITSN-DHPH-His$_{10}$ was added and incubated for 15 min to bind to NTA(Ni) lipids. For +ITSN$_{cat}$-2xC2 experiments, 80 nM of ITSN-DHPH-His$_{10}$ was added and incubated for 5 min at room temperature to bind to DOPS lipid. For ELMO1:DOCK1 experiments, 80 nM of ELMO1:DOCK1 was added and incubated for 5 min at room temperature to bind to PI(3,4,5)P$_3$ lipid. After incubation, unbound GEF was washed out with 1 mL GEF buffer supplemented with 2 mM β-mercaptoethanol and 0.2 mg/mL blocker casein. 100 pM of freshly diluted GTPase:GDI complex was mixed in GEF buffer containing 10 % (v/v) oxygen scavenger system and 200 μM GTP, and added to the SLB chamber and imaging started immediately. Single molecule, simultaneous dual color movies were captured at 22 ms exposure, continuous illumination.

## Single molecule TIRFM GEF-mediated nucleotide exchange assays

GEF was recruited to supported lipid bilayers as described above. Cy3-GTPase:GDP-Cy5:GDI complex (50 pM) was added to the chamber in the presence of 200 μM of unlabeled GTP and imaging began immediately. Imaging via sensitized emission of GDP-Cy5 allowed for the capture of the moment of nucleotide exchange due to high time resolution image series (1 frame exposure (22 ms)). Movies were captured using continuous illumination for 40 s.

## Lipid pattern formation and visualization

Following supported lipid bilayer preparation (as described above), chambers were washed with 1 mL imaging buffer (20 mM HEPES pH 8.0, 150 mM NaCl, 5 mM MgCl$_2$, 0.5 mM EGTA, 1 mM GTP, 1 mM ATP, 20 mM BME, 200 μg/mL β-casein, 1.25 mg/mL glucose oxidase, 0.2 mg/mL catalase and 400 mg/mL glucose). For lipid pattern formation, 20 nM PIP5K and 6 nM DrrA-OCRL were added to the flow chamber, if not stated otherwise.

To visualize the developing lipid pattern, we included 2 nM A488-DrrA and 2 nM A647-PH to observe PI(4)P and PI(4,5)P$_2$, respectively. Bilayers were imaged in TIRFM for 40 min with 100 ms exposure at 10-s time intervals.

## GEF localization and densities

Lipid patterns were formed and allowed to stabilize for 10–15 min in the presence of "base mix" (2 nM A647-PH, 20 nM PIP5K and 6 nM Drra-OCRL) in fresh imaging buffer (20 mM HEPES pH 8.0, 150 mM NaCl, 5 mM MgCl$_2$, 0.5 mM EGTA, 1 mM GTP, 1 mM ATP, 20 mM BME, 200 μg/mL β-casein, 1.25 mg/mL glucose oxidase, 0.2 mg/mL catalase and 400 mg/mL glucose). TIRFM imaging was started with 100 ms exposure at 10 s time intervals to acquire a baseline and after 3 min, imaging was paused. "Base mix" with 1 nM A488-ITSN$_{cat}$-PH in fresh imaging buffer was then flushed into the chamber and imaging resumed for a total of 10 min.

For GEF density measurements microscope settings were adjusted to an exposure of 22 ms. Additionally, the labeled fraction of (A488-)ITSN$_{cat}$-PH was reduced (5-, 10-, 20- or 50-fold) to reach single molecule regime.

## Membrane-templated Rho GTPase patterns (bulk and single molecule)

Lipid patterns were formed and allowed to stabilize for 10–15 min in the presence of supplemented "base mix" (20 nM PIP5K, 6 nM DrrA-OCRL, 2 nM Cy3-PLCδ(PH)) in fresh imaging buffer (20 mM HEPES pH 8.0, 150 mM NaCl, 5 mM MgCl$_2$, 0.5 mM EGTA, 1 mM GTP, 1 mM ATP, 20 mM BME, 200 μg/mL β-casein, 1.25 mg/mL glucose oxidase, 0.2 mg/mL catalase and 400 mg/mL glucose). The supplements varied with experiments and are detailed in figure legends where appropriate. Supplements used were: (1) Cdc42; provided either alone (5.5 nM) or in 1:1 complex with RhoGDI (600 nM) (formed in situ). (2) An activity readout (either 40 nM A488-wCRIB, 20 nM A488-2xwCRIB or 40 nM Atto488-N-WASP). (3) A soluble GAP (20 nM OPHN1$_{cat}$). Bilayers were imaged for 3 min to establish a baseline, after which imaging was paused. Supplemented "base mix" with 1 nM ITSN$_{cat}$-PH in fresh imaging buffer was then flushed into the chamber and imaging was resumed for a total of 10 min.

For single molecule experiments of Cdc42 on patterned SLBs, the labeled fraction of (A647-) Cdc42 was reduced (100-fold) to reach single molecule regime. Membrane-templated Cdc42 patterns were allowed to form for 5 min (as described above). Next, single molecule TIRFM movies were captured at 22 ms exposure and continues illumination for 1 min.

## Cdc42 dwell time measurement

Lipid patterns were formed and allowed to stabilize for 10–15 min in the presence of supplemented "base mix" (20 nM PIP5K, 6 nM DrrA-OCRL, 2 nM Cy3-PLCδ(PH)) in fresh imaging buffer (20 mM HEPES pH 8.0, 150 mM NaCl, 5 mM MgCl$_2$, 0.5 mM EGTA, 1 mM GTP, 1 mM ATP, 20 mM BME, 200 μg/mL β-casein, 1.25 mg/mL glucose oxidase, 0.2 mg/mL catalase and 400 mg/mL glucose). Fixed supplements were (1) Cdc42; provided alone (5.5 nM). (2) A soluble GAP (20 nM OPHN1$_{cat}$). Some supplements varied with experiments and are detailed in figure legends where appropriate. Varying supplements were the activity readout (either 40 nM A488-wCRIB or 20 nM A488-2xwCRIB). After flushing in fresh supplemented "base mix" with 1 nM ITSN$_{cat}$-PH in fresh imaging buffer, membrane templated Cdc42 patterns were allowed to form for 5–10 min. Next, TIRF images were acquired every 5 s with 100 ms exposure for 70 s to establish a baseline. A flow of supplemented "base mix" in fresh imaging buffer without Cdc42 but with 22 nM GDI was initiated (5 μL/s) via a syringe

pump whilst continuing to image as above for a minimum of 200 s total.

## Data analysis

### Membrane fluidity

Time traces of the normalized fluorescence intensity in Fluorescence recovery after photobleaching (FRAP) regions were measured in Fiji (Schindelin et al, 2012) and were fitted with a mono exponential growth function in Orgin (OriginLab Corporation).

## Segmentation of Cdc42 patterns

The lipid patterns visualized by the A647-PH marker were segmented using our custom scripts that execute denoising, equalization, and pattern segmentation (autoSegmentation_dice-N-splice_LipidPatch.py and autoSegmentation_LipidPatch.py). Initially, the A647-PH channel timeseries underwent denoising using a Non-Local Means algorithm from Scikit-Image's restoration module. Post-denoising, histogram equalization was implemented to uniformly readjust contrast levels across all image sections. An adaptive Histogram Equalization method (CLAHE) from OpenCV was employed for this purpose. Lastly, the histogram-equalized image timeseries was segmented through application of a Random Walker Segmentation algorithm, featuring two distinct labels: one for lipid pattern regions and another for areas exterior to the pattern. Preliminary markers for these regions were designated by assigning markers based on the histogram's splitting points above (inside the pattern) and below (outside the pattern) the 50th percentile. Following segmentation, regions underwent expansion from their initial markers, subsequently removing any errant pixels via execution of five iterations of binary closing and binary opening using Scipy's ndimage module. Segmentation scripts are publicly accessible through our GitHub repository.

## Calculation of patterning index

### In vitro

Using the segmentation map we generated from the lipid channel, we designated an "inside" (high $PI(4,5)P_2$) and "outside" (high $PI(4)P$) region for each frame. Using the ImageJ plugin Fiji, we then took a mean intensity value of both regions and divided values from inside by the outside for both total- and active-Cdc42. The resulting number was what we term "patterning index (PI)".

### In vivo

Fiji (Schindelin et al, 2012) was used to process imaging data, create kymographs of time lapse movies, and quantify fluorescence intensity over time. For patterning experiments, radial averaging was used to measure fluorescence intensity around the wound over time (Moe et al, 2021). Patterning indices were measured by dividing the peak fluorescence of active and total Cdc42 early in repair by the respective background fluorescence at least 30 μm from the wound center. Two-sample t-tests were used to determine statistical significance between pre-zone and zone patterning indices. GraphPad Prism was used to plot fluorescence intensity data and generate graphs.

## Single molecule tracking of dual color TIRF microscopy data

Dual color single molecule TIRFM data was analyzed using Fiji (Schindelin et al, 2012). Data from NIS-Elements was first imported using the BioFormats plugin. Images were processed using the Fiji plugin Trackmate (Ershov et al, 2022). Individual spots were identified and incorporated into tracks, the statistics of which were exported and used as input files for our custom colocalization scripts. The source codes for these scripts are hosted publicly on GitHub, however briefly, GTPase and GDI channel "spot statistics" files from Trackmate were taken as an input for our colocalization script. The SpotColocalization.py script analyzed data from both channels and generated output files that contained fields that identified which tracks and spots were colocalized. Spots were considered to be colocalized if their centroid distances were less than or equal to 0.5 μm. The output file was used as an input for our the getStat_TracksColocalized.py script, calculated metrics such as landing rate and fractions of colocalized tracks. Arguments were set to their defaults for both scripts except for the field of view which was set to 0.75 to only analyze the top-left 75% of the field of view. Colocalization probabilities at different time points in the lifetime of individual tracks were calculated with calc_ColocalizationProbability_positionSpecific.py. The source codes for these scripts are hosted publicly on GitHub. In the case of control experiments, the control argument was set to "True". Briefly, this means that all spots detected in the first frame are counted, rather than exclusively new landing events, as in experimental analysis. Details of script arguments can be found in the GitHub repository (see "Source code availability").

## Single molecule tracking of total Cdc42

Following segmentation, A647-Cdc42 was tracked using the Fiji plugin, Trackmate (Ershov et al, 2022) as described above to generate a spot statistics file for both "inside" and "outside" regions. Statistics for Cdc42 tracks were used as an input for our custom scripts, StepSize-distribution.R and MSD-distribution.py to determine metrics such as step size and MSD distributions. The source codes for these scripts are hosted publicly on GitHub.

## Calculation of GEF density for single molecule TIRFM assays

$ITSN_{cat}$-H10 density was calculated by mixing a small fraction (10–100 pM) of AF488-$ITSN_{cat}$ with unlabeled $ITSN_{cat}$ totaling 400 nM and incubated for 15 min, room temperature, on supported lipid bilayers (prepared and assembled as described above) containing 2% molar fraction of Ni-NTA-DGS lipids. Following incubation, chambers were washed with 1 mL of GEF buffer, and images were taken with 22 ms exposure times in multiple locations on the bilayer. $ITSN_{cat}$-2xC2 was calculated as with above however using 80 nM total $ITSN_{cat}$-2xC2, incubated for 5 min. ELMO-DOCK density was calculated as with $ITSN_{cat}$-2xC2, however no unlabeled ELMO-DOCK was used in the incubation step. ELMO-DOCK was incubated for 5 min on supported lipid bilayers containing 4% $PI(3,4,5)P_3$ prior to washing as above. These concentrations of labeled GEF resulted in single molecule densities

which allowed the counting of spots via the Fiji plugin Trackmate (Ershov et al, 2022) with the density of GEF calculated by assuming a linear relationship and extrapolating to total GEF.

For calculating $ITSN_{cat}$-PH density in experiments with patterned SLBs, a similar approach was used as above, however a total concentration of 1 nM $ITSN_{cat}$-PH was used with 20 - 200 pM of A488-$ITSN_{cat}$-PH.

## Estimation of surface coverage

A measurement was made between the most distal amino acids of the crystal structures of $ITSN_{cat}$ (PDB: 1KI1) and ELMO1:DOCK1 (PDB: 6TGB) in PyMol (The PyMOL Molecular Graphics System, Version 3.0 Schrödinger, LLC). This measurement was used to estimate the surface area of a single GEF molecule (assuming a circular footprint). This area was multiplied by the number of GEF molecules (as calculated above) and expressed as a fractional percentage of the surface area.

## Calculation of dwell times of photoactivatable Cdc42 during oocyte wounding

For photoactivation experiments, registration in FIJI was used to correct for advection-induced movement of photoactivated spots around the wound prior to measuring fluorescence intensity over time. GraphPad Prism was used to generate plots of fluorescence intensity over time and perform nonlinear regression with one phase decay to calculate $t_{1/2}$ to disappearance. In these calculations, $t = 0$ s refers to the moment of photoactivation and $t_{1/2}$ to disappearance is the time it takes for the photoactivated region to diminish to half of the peak intensity. GraphPad Prism was also used to measure the correlation between $t_{1/2}$ to disappearance and time after wounding. One-way ANOVA with a Dunnett comparison was used to measure statistical significance between experimental conditions and a control condition. Two-sample t-tests were used to determine statistical significance of $t_{1/2}$ to disappearance between control and ITSN-C2 conditions.

## Source code availability

All data analysis was conducted using Python 3.10.13. Numpy 1.24.4 and Pandas 2.1.2 were used for data manipulation and analysis. R version 4.2.2 was utilized for the execution of R scripts. Shell scripts were executed in Zsh version 5.9 for batch processing and file handling. Image analysis and segmentation were performed using Sci-kit Image 0.22.0, OpenCV 4.7.0, and Scipy 1.11.3. Parallelization was implemented where applicable with the Multiprocessing package to enhance processing efficiency. Single molecule spots were tracked with TrackMate. Colocalized GTPase and GDI spots were identified using the SpotColocalization.py script. Colocalization statistics were calculated using getStat_TracksColocalized.py and colocalization probabilities at specific positions throughout the lifetime of tracks were determined with calc_ColocalizationProbability_positionSpecific.py. Lipid patterns were segmented using autoSegmentation_dice-N-splice_LipidPatch.py, and subsequent analysis was carried out with GTPase_patterning_analysis.py. All scripts used for the analysis as well as accessory scripts can be accessed at the following GitHub Repository: https://github.com/iamankitroy/GTPase-Patterning.

## Microscopy

### TIRF microscopy

All TIRFM experiments were performed at 23 °C using a customized Nikon TIRF Ti2 microscope (Nikon Instruments). Image acquisition was achieved by dual EM CCD Andor iXon cooled cameras (Cairn Research) controlled by Nikon NIS – Elements software (Nikon Instruments). Imaging was performed through an Apo TIRF × 60 oil DIC N2 objective using a custom multi-laser launch system with 488, 532, 561 and 639 nm lasers (AcalBFi LC). Single molecule movies were captured in simultaneous dual color using a T-splitter which allowed for simultaneous hardware-triggering of lasers. Continuous illumination resulted in a 22 ms exposure time. Bulk experiment movies were captured with a range of exposures between 20 and 100 ms with interleaved imaging.

## FRAP-TIRF microscopy

FRAP-TIRFM experiments were conducted at room temperature using a Nikon Ti1 microscope (Nikon Instruments) controlled by the Visiview software (Visitron Systems). Images were captured using Evolve Delta cameras from Photometrics (AZ, USA). A circular ROI was drawn in the Visiview software on the bilayer at which photobleaching took place. 100% laser power was applied for 3 s using the FRAP module within the Visitron software (Visitron systems). Following FRAP, TIRF images were captured at 1 s time intervals for a total of 2 min.

## Confocal microscopy

Oocytes were imaged using a Nikon Eclipse Ti inverted laser-scanning confocal microscope and wounded with a 440 nm dye laser pumped by a Laser Science Inc. MicroPoint 337 nm nitrogen laser. 4D time lapse movies consist of 5 focal planes (1 μm step size) compiled via maximum intensity projections. Photoactivation experiments (1 focal plane) were performed using 405 nm light to photoactivate a specified region upon user-initiated firing. Photo-activated regions have a gaussian distribution.

# Data availability

Source data for all main figures have been deposited at BioStudies and are available via the following link: (https://www.ebi.ac.uk/biostudies/studies/S-BSST1811?key=8b769085-e10b-42e0-beea-4216bc563ef9). All custom scripts used for data analysis as well as accessory scripts can be accessed at the following GitHub Repository (https://github.com/iamankitroy/GTPase-Patterning).

The source data of this paper are collected in the following database record: biostudies:S-SCDT-10_1038-S44318-025-00418-z.

# Peer review information

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

## Acknowledgements

We thank Philippe Bastiaens for continuous support, useful discussions and help in shaping the manuscript and Svenja Koch for expert technical assistance. This work was supported by funds from the Max Planck Society (to PB), the Human Frontier Science Program (HFSP, grant no. CDA00070/2017-2), the German Research Foundation (DFG, grant no. BI 1998/2-1 to PB), and the National Institutes of Health (RO1GM052932 and R35GM156206 to WMB).

## Author contributions

**Michael C Armstrong**: Conceptualization; Resources; Data curation; Formal analysis; Supervision; Funding acquisition; Validation; Investigation; Visualization; Methodology; Writing—original draft; Project administration; Writing—review and editing. **Yannic R Weiß**: Conceptualization; Resources; Data curation; Formal analysis; Supervision; Funding acquisition; Validation; Investigation; Visualization; Methodology; Writing—original draft; Project administration; Writing—review and editing. **Lila E Hoachlander-Hobby**: Resources; Data curation; Formal analysis; Validation; Investigation; Visualization; Methodology; Writing—original draft; Project administration; Writing—review and editing. **Ankit A Roy**: Resources; Data curation; Software; Formal analysis; Validation; Investigation; Visualization; Methodology; Writing—original draft; Project administration; Writing—review and editing. **Ilaria Visco**: Resources; Data curation; Formal analysis; Validation; Investigation; Visualization; Methodology; Writing—original draft; Project administration; Writing—review and editing. **Alison Moe**: Resources; Data curation; Software; Formal analysis; Validation; Visualization; Methodology; Writing—original draft; Project administration; Writing—review and editing. **Adriana E Golding**: Resources; Data curation; Formal analysis; Validation; Investigation; Visualization; Methodology; Writing—original draft; Project administration; Writing—review and editing. **Scott D Hansen**: Resources; Methodology; Project administration. **William M Bement**: Conceptualization; Resources; Data curation; Formal analysis; Supervision; Funding acquisition; Validation; Investigation; Visualization; Methodology; Writing—original draft; Project administration; Writing—review and editing. **Peter Bieling**: Conceptualization; Resources; Data curation; Formal analysis; Supervision; Funding acquisition; Validation; Investigation; Visualization; Methodology; Writing—original draft; Project administration; Writing—review and editing.

Source data underlying figure panels in this paper may have individual authorship assigned. Where available, figure panel/source data authorship is listed in the following database record: biostudies:S-SCDT-10_1038-S44318-025-00418-z.

## Funding

## Disclosure and competing interests statement

The authors declare no competing interests.

# Expanded View Figures

**Figure EV1.   Cy3-Cdc42 follows same patterning as IT-Cdc42 around wounds.**

(A) Micrograph of Cy3-Cdc42 during Pre-wound, Pre-zone, and Zone time points in the repair process. Scale bar 20 μm. (B) Kymograph of the micrograph from (A) generated by radially averaging signal intensity around the wound over time (Moe et al, 2021; see "Methods"). The arrow denotes when the wound occurred, and W denotes the wound location. The yellow line "P" indicates the Pre-zone time point and the yellow line "Z" indicates the Zone time point. Vertical and horizontal scale bars are 30 s and 3 μm, respectively. (C) Line scan generated from the kymograph in (B) to show Cy3-Cdc42 fluorescence intensity at the Pre-zone time point as a function of distance from the wound center. (D) Line scan generated from the kymograph in (B) to show Cy3-Cdc42 fluorescence intensity at the Zone time point as a function of distance from the wound center. (E) Patterning index of Cy3-Cdc42 at the Pre-zone time point compared to the Zone time point. A two-sample t-test was used to determine statistical significance; $n = 13$. *P* value = 6.48965E-07. (F) Patterning index of Cy3-Cdc42 compared to IT-Cdc42 (IT-Cdc42 data were taken from Fig. 1). A two-sample t-test was used to determine statistical significance. Cy3-Cdc42 $n = 13$ individual cells, IT-Cdc42 $n = 12$ individual cells, in $N = 1$ experiment. (G) Example of wGBD and IT-Cdc42 timing in cells where both accumulate rapidly. (H) Example of wGBD and IT-Cdc42 timing in cells where both accumulate more slowly. (I) Timing of IT-RhoA and rGBD enrichment around the wound (mean, $N = 11$ per condition, shaded area = SD).

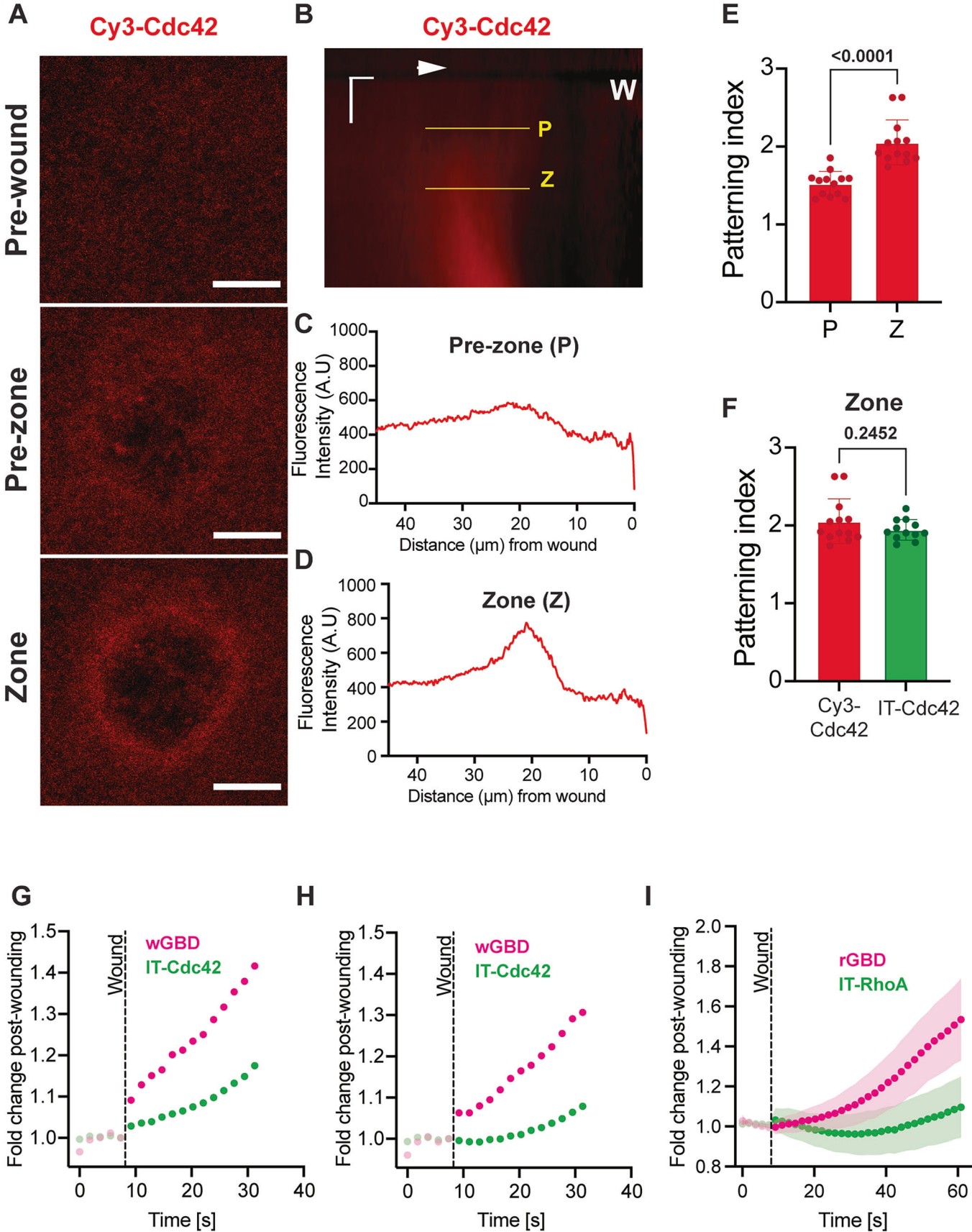

## Nucleotide exchange on supported lipid bilayers

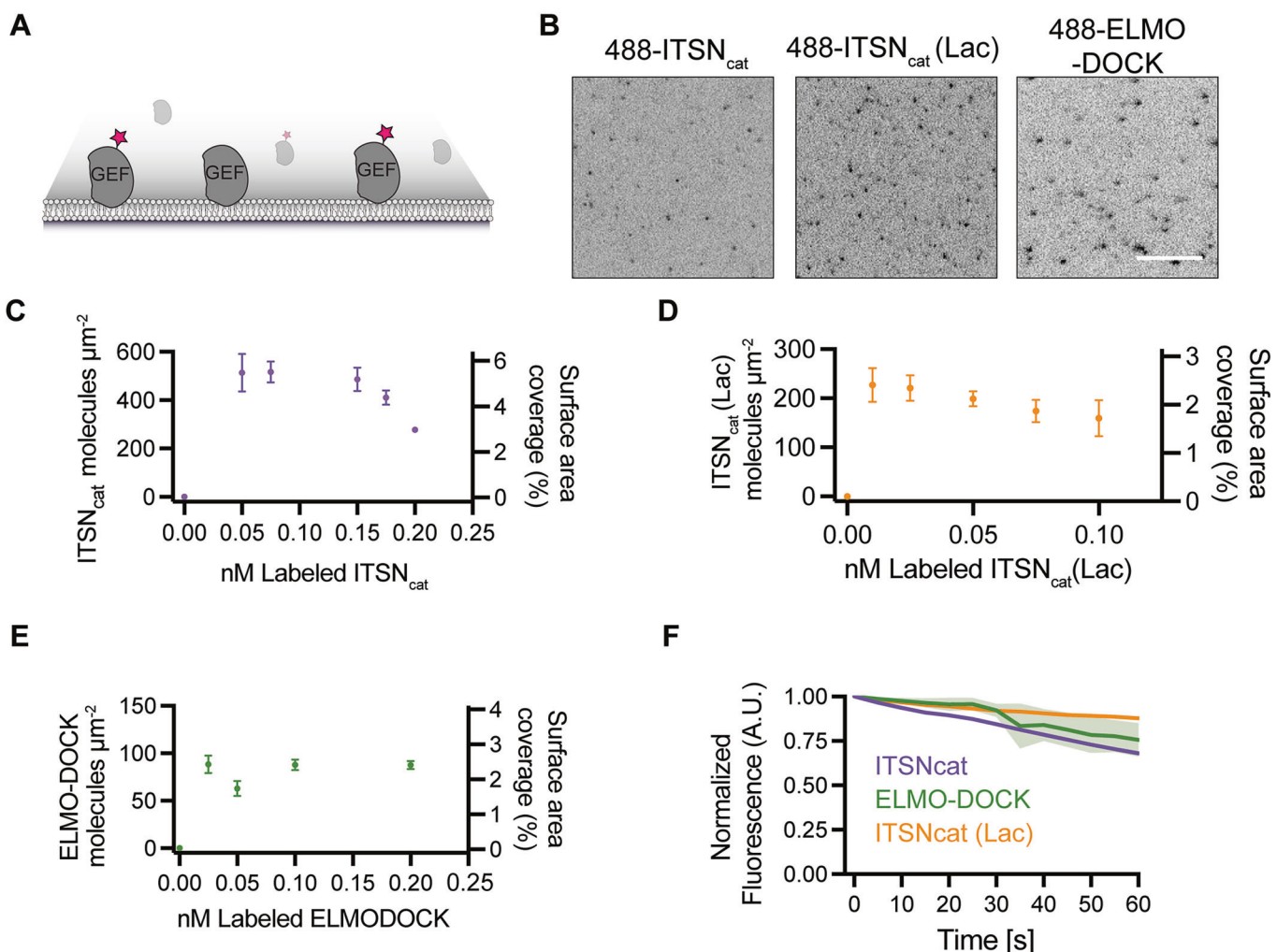

**Figure EV2.  Achieving bilayer surface coverage of active GEF.**

(A) Scheme of the single molecule quantification of RhoGEF density on SLBs. (B) 488-ITSN$_{cat}$ (left) 488-ITSN$_{cat}$(Lac) (center) and 488-ELMO:DOCK (right) recruited to SLBs (containing 2% DGS-NTA(Ni), 22.5% PS and 2% PI(3,4,5)P$_3$ respectively) Scale bar = 10 µm. (C) Quantification of ITSN$_{cat}$ density on SLBs (454 molecules/µm$^2$, 4.8% surface coverage), left axis. Estimated bilayer surface area coverage of total ITSN$_{cat}$ right axis (mean ± SD). $N = 1$ experiment, $n = 3$–13 fields of view imaged per condition, total spots = 49823. (D) Quantification of ITSN$_{cat}$(Lac) density on SLBs. (203 molecules/µm$^2$, 2.2% surface coverage), left axis. Estimated bilayer surface area coverage of total ITSN$_{cat}$(Lac) right axis (mean ± SD) $N = 1$ experiment, $n = 3$–15 fields of view imaged per condition, total spots = 34044. (E) Quantification of ELMO-DOCK density on SLBs. (84 molecules/µm$^2$, 2.3% surface coverage), left axis. Estimated bilayer surface area coverage of total ELMO-DOCK right axis (mean ± SD) $N = 1$ experiment, $n = 2$–11 fields of view imaged per condition, total spots = 17985. (F) Survival fraction of 400 nM 488-ITSN$_{cat}$, 400 nM 488-ITSN$_{cat}$(Lac) and 80 nM 488-ELMO-DOCK as a function of time during wash off (shaded area, SD). Flow rate 10 µL/s.

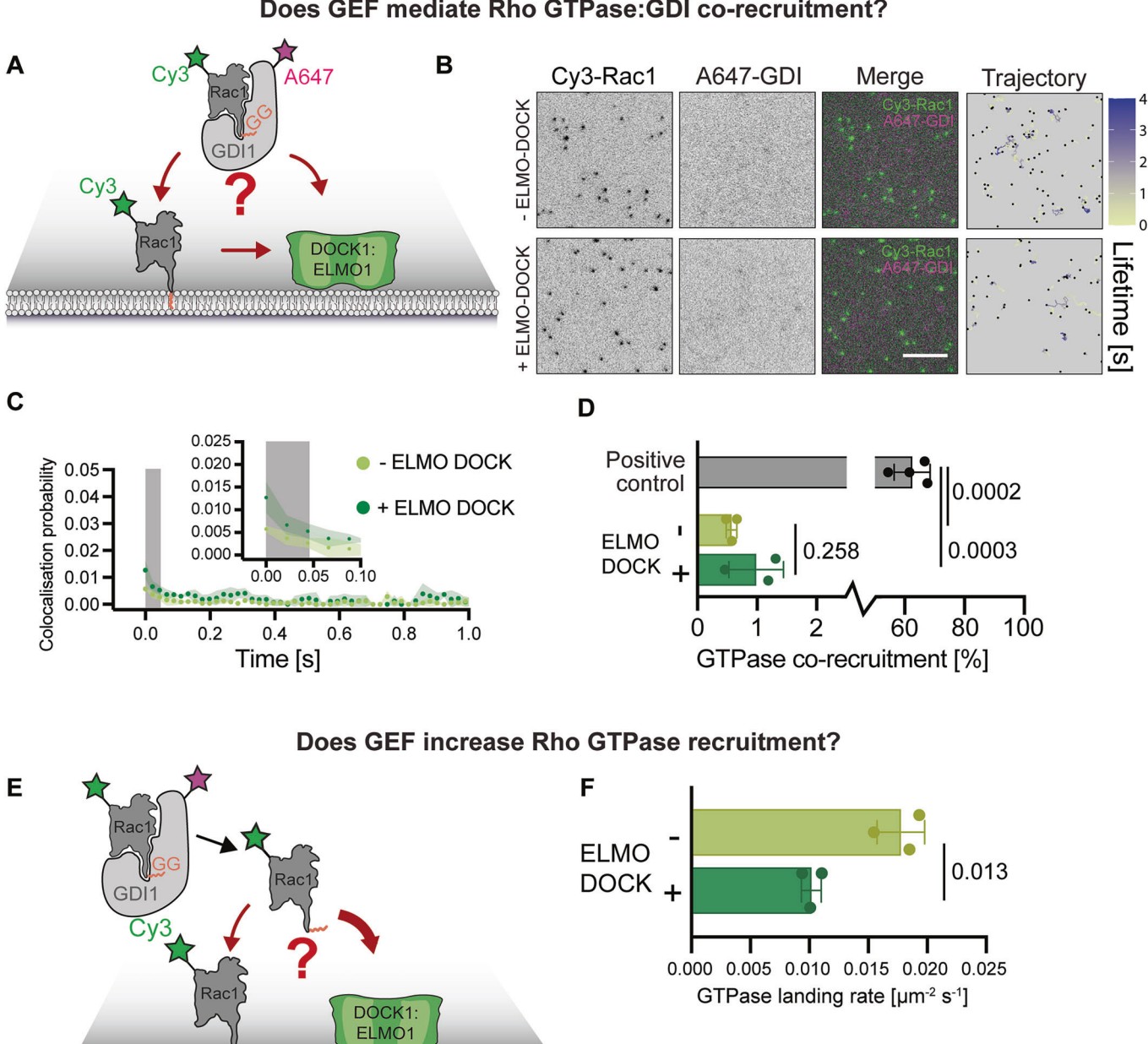

**Figure EV3.    DOCK family GEF does not provide mechanism for Cdc42:GDI1 recruitment.**

(A) Scheme of two potential routes to GEF-mediated GTPase recruitment and activation. (B) TIRFM images of single molecule recruitment of Rac1 from complex (100 pM) (Cy3-Rac1, A647-GDI1, merge, and trajectories of Cdc42, left-to-right) on a PM SLB (containing 4% PI(3,4,5)P$_3$) in the absence (top) or presence (bottom) of 80 nM ELMO-DOCK. Scale bar = 10 μm. (C) Probability (dots) ± SD (area) of Cy3-Rac1 and A647-GDI1 co-localization as a function of Rac1 lifetime on SLBs coated with (dark green) or without (light green) ELMO-DOCK. $t = 0$ s is the moment of recruitment. Vertical grey box demarks the frames evaluated for co-recruitment. – ELMO-DOCK $N = 3$, $n = 5068$, + ELMO-DOCK $N = 3$, $n = 2984$. (D) Mean fraction ± SD of Cy3-Rac1 molecules co-recruited with A647-RhoGDI1 to PM SLBs in the absence or presence of ELMO-DOCK compared to the positive control (Fig. 3E). ( ± GEF experiments (– ELMO-DOCK $N = 3$, $n = 5068$, + ELMO-DOCK $N = 3$ $n = 2984$, control $N = 4$ $n = 968$, SD)). Statistical comparisons via T-test. (E) Scheme of GEF-dependent and independent membrane recruitment of free Rho GTPases. (F) Cy3-Rac1 landing rates (mean ± SD) on PM SLBs in the absence (light green) or presence (dark green) of ELMO-DOCK. – ELMO-DOCK $N = 3$ independent experiments, $n = 5068$ observations, + ELMO-DOCK $N = 3$ independent experiments, $n = 2984$ observations. Statistical comparisons via t-test.

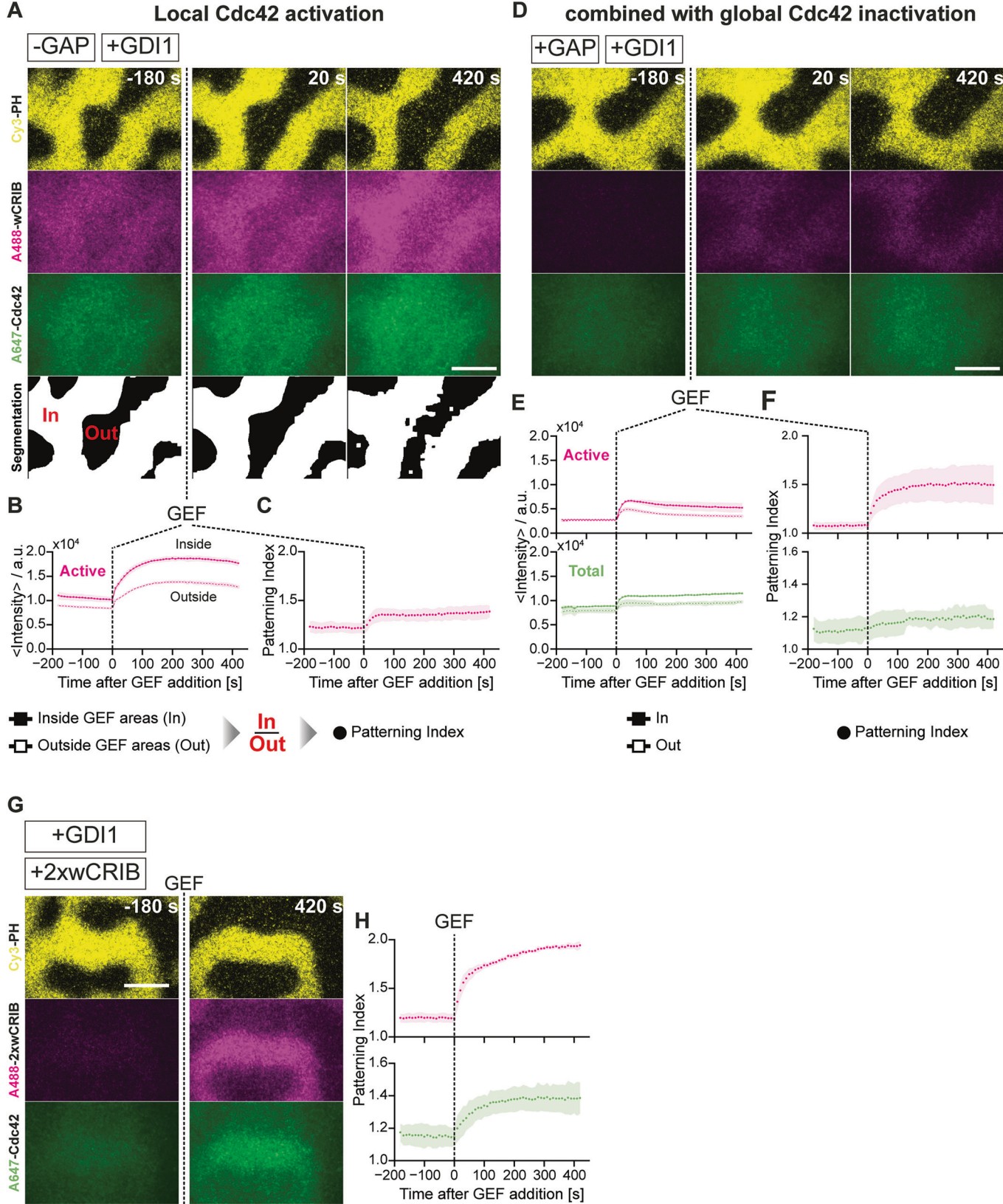

**Figure EV4.  RhoGDI1 negligibly affects the reconstitution of membrane-templated Rho GTPase activity patterns in vitro.**

(A) Time-lapse multi-color TIRFM images of Cy3-PH (2 nM, yellow), A488-wCRIB (40 nM, magenta) and A647-Cdc42:RhoGDI1 complexes (600 nM, green) on PIP patterns at indicated times before or after addition of $ITSN_{cat}$-PH (1 nM) at $t = 0$ s. Bottom row shows segmentation based on the lipid pattern. (B) Average intensities inside (filled squares, $<I_{In}>$ ) and outside (hollow squares, $<I_{Out}>$ ) of the GEF-containing membrane areas or (C) patterning index (full circle, $PI = <I_{In}> / <I_{Out}>$) of A488-wCRIB (magenta) over time. (D–F) correspond to (A–C), with the additional presence of $OPHN1_{cat}$ (20 nM). However, the segmentation is not shown in (D) (Image from (D) also used in Fig. 6B and Appendix Fig. S5A (lower). Additionally, in (E) are the average intensities inside and outside of the GEF-containing membrane areas and in (F) the patterning index of A647-Cdc42 (green) over time. (G) Time-lapse multi-color TIRFM images of the effects of the synthetic dimerization mimic 2xwCRIB on templated Rho GTPase activity patterns. Conditions as in (D) with A488-wCRIB replaced by A488-2xwCRIB (20 nM, magenta). (H) corresponds to (F) under conditions as in (G). All numeric data represent the mean from three independent experiments (symbols) ± SD (shaded areas) ($N = 3$). All scale bars are 20 µm as indicated.

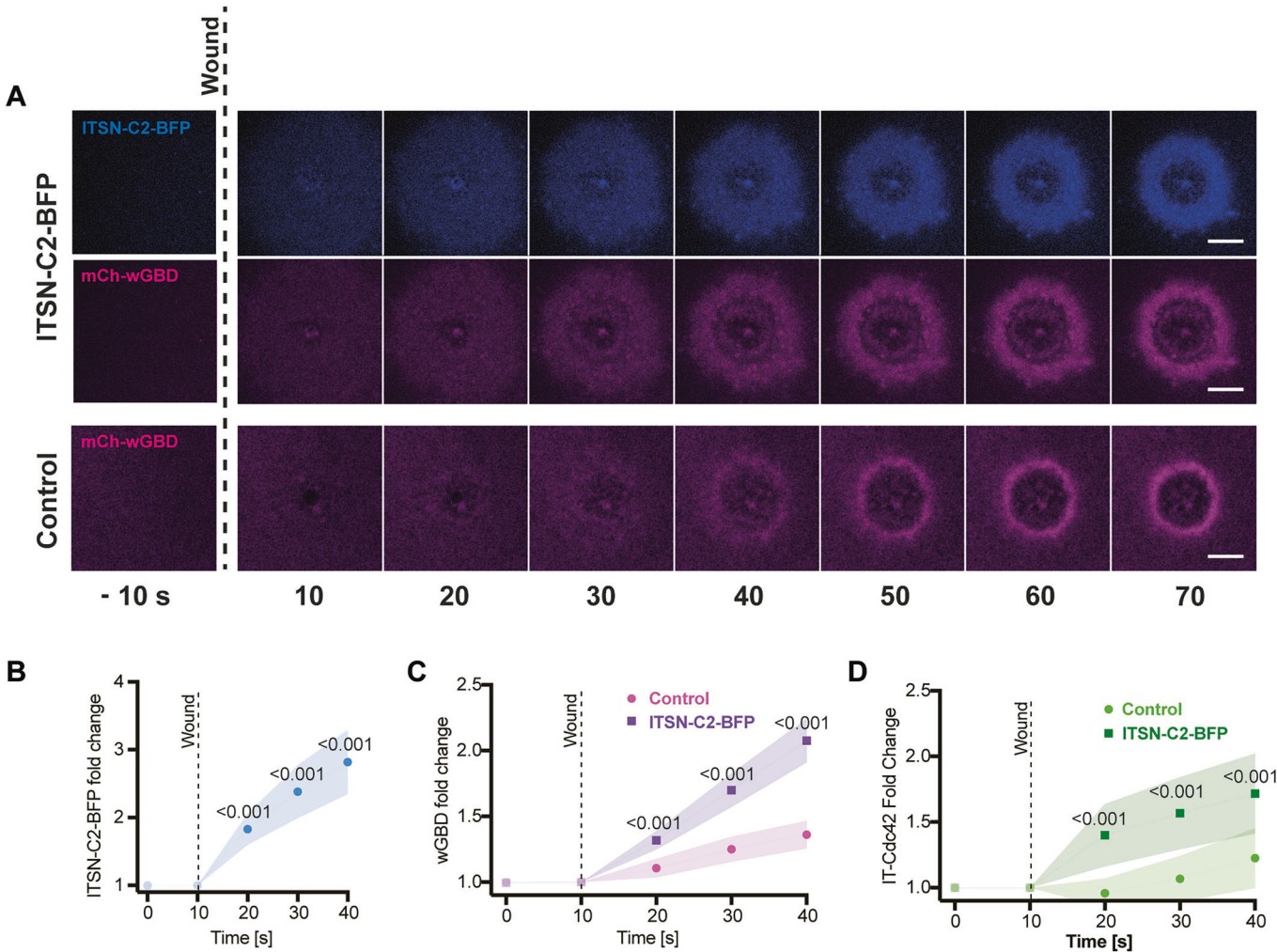

**Figure EV5. ITSN-C2-BFP is recruited to single cell wounds and increases Cdc42 activity and patterning.**

(A) Cell expressing Itsn-C2-BFP (Itsn-C2-BFP) or not (Control). Itsn-C2-BFP Top: Montage of images from confocal time-lapse movie showing accumulation of ITSN-C2-BFP around wounds. Itsn-C2-BFP Bottom: corresponding wGBD signal from images shown in top. Control: example of wGBD signal in cell not expressing ITSN-C2-BFP for comparison. Time in min:sec. Scale bar 20 μm. (B) Plot of Itsn-C2-BFP over time. Data show mean ± SD from 22 cells; relative to pre-wound signal (students t-test, *P* values 4.71E-11, 1.09-10 and 3.75E-16 for timepoints 20, 30 and 40 s respectively). (C) Plot comparing wGBD accumulation around wounds over time in control cells and cells expressing Itsn-C2-BFP. Data show mean ± SD from 17 cells; Itsn-C2-BFP versus control (students *t* test, (students t-test, *P* values 5.61E-10, 4.82-13 and 3.91E-16 for timepoints 20, 30 and 40 s respectively). (D) Plot comparing IT-Cdc42 accumulation around wounds over time in control cells and cells expressing Itsn-C2-BFP. Data show mean ± SD from 12 cells; Itsn-C2-BFP versus control (students t-test, (students t-test, *P* values 3.63E-4, 6.66E-6 and 2.11E-5 for timepoints 20, 30 and 40 s, respectively).

