## [Peer Review File · The EMBO Journal]

The biochemical mechanism of Rho GTPase membrane binding, activation and retention in activity patterning

Michael Armstrong, Yannic Weiß, Lila Hoachlander-Hobby, Ankit Roy, Ilaria Visco, Alison Moe, Adriana Golding, Scott Hansen, William M Bement, and Peter Bieling

Corresponding authors: Peter Bieling (peter.bieling@mpi-dortmund.mpg.de) , William M Bement (wmbement@wisc.edu)

Review Timeline:

Submission Date:	17th Sep 24
Editorial Decision:	4th Oct 24
Revision Received:	14th Jan 25
Editorial Decision:	14th Feb 25
Revision Received:	4th Mar 25
Accepted:	7th Mar 25

Editor: Ieva Gailite

Transaction Report:

Dear Peter,

Thank you for submitting your manuscript for consideration by the EMBO Journal. We have now received comments from three reviewers, which are included below for your information.

As you will see from the reports, all reviewers appreciate the study, and especially reviewers #2 and #3 find it of broad interest to the research field. In addition, reviewers #1 and #2 indicate several areas of clarification that would need to be implemented in the revised manuscript. Reviewer #1 also requests better integration of the findings with the existing literature. From my side, I find their comments reasonable, and therefore invite you to address these points in a revised version.

We generally allow three months as standard revision time. As a matter of policy, competing manuscripts published during this period will not negatively impact on our assessment of the conceptual advance presented by your study. However, please contact me as soon as possible upon publication of any related work to discuss the appropriate course of action. Should you foresee a problem in meeting this deadline, please let us know in advance to discuss an extension.

When preparing your letter of response to the referees' comments, please bear in mind that this will form part of the Review Process File and will therefore be available online to the community. For more details on our Transparent Editorial Process, please visit our website: <https://www.embopress.org/page/journal/14602075/authorguide#transparentprocess>. Please also see the attached instructions for further guidelines on preparation of the revised manuscript.

Please feel free to contact me if you have any further questions regarding the revision. Thank you for the opportunity to consider your work for publication. I look forward to discussing your revision.

With best wishes,

leva

leva Gailite, PhD
Senior Scientific Editor
The EMBO Journal
Meyerhofstrasse 1
D-69117 Heidelberg
Tel: +4962218891309
i.gailite@embojournal.org

- a Reagents and Tools Table as part of the Methods section, which can be downloaded from our author guidelines

We realize that it is difficult to revise to a specific deadline. In the interest of protecting the conceptual advance provided by the work, we recommend a revision within 3 months (12th Jan 2025). Please discuss the revision progress ahead of this time with the editor if you require more time to complete the revisions.

Referee #1:

This manuscript builds on many years of impressive work from the Bement laboratory using small membrane wounds in *Xenopus* oocytes to understand how localized Rho GTPase signaling facilitates wound repair. In particular, it is based on their observation 20 years ago that active Cdc42 accumulates around the wound site, as well as more recently showing the ratio of total to active Cdc42. The authors now link this to the contribution of RhoGDI in delivering Cdc42 to the wound site, using a series of elegant in vitro biochemical experiments with LUVs before returning at the end of the manuscript to the wound model. My main point for revision is that the authors should improve their statements on the conceptual novelty achieved through their development of the new in vitro reconstitution model for investigating the role of RhoGDI in Cdc42 activation, which was undoubtedly technically challenging and will be very useful to the community. The 'key findings' section and abstract should mention the development of the in vitro reconstitution methodology, and there are other appropriate areas in the text where they can emphasize the importance of its development in answering key questions relating to Cdc42.

Other points:

1. Figure 1, S1: timing of accumulation of wGBD versus IT-Cdc42. wGBD could bind to other Rho GTPases in addition to Cdc42 (e.g. RhoQ), which might accumulate earlier than active Cdc42 at the wound site. Alternatively, the timing difference could simply reflect the relative sensitivity of detecting wGBD versus IT-Cdc42.
2. Figure 2: It is not surprising that Cdc42 or Rac1 bound to RhoGDI inhibits GEF-induced activation, based on the structure of RhoGDI bound to these proteins and PMID: 12471028, among several papers, which demonstrates that RhoGDI needs to dissociate from Rac1 (or Cdc42/RhoA) to allow e.g. Tiam1 to act as GEF.
3. There is abundant evidence that RhoGDI dissociation from Rho GTPases requires phosphorylation and other post-translational mechanisms, which would then allow the Rho GTPases to insert into membranes. This is not explicitly tested in the in vitro LUV experiments and would significantly strengthen the message of the manuscript if it were tested. The authors downplay the importance of post-translational mechanisms of regulation in the Introduction and Discussion.
4. The authors should reference the extensive work on Cdc42 membrane localization and positive feedback loops in *S. cerevisiae* budding as well as molecular modeling on Rho GTPase/RhoGDI (e.g. reviewed by Garcia-Mata et al., *Nat Rev Mol Cell Biol* 2011: modeling by several groups, including Edelstein-Keshet, with whom some of the authors have collaborated). Studies in *S. cerevisiae* demonstrated clearly that activation prolongs retention of Cdc42 at the bud plasma membrane during bud extension in a positive feedback loop through a complex including the Cdc42 GEF Cdc24 and its target PAK, and that it is not activated by Cdc42 until delivered to the bud membrane in a mechanism involving yeast RhoGDI as well as vesicle trafficking (e.g. reviewed by Woods and Lew, *Small GTPases* 2019).
5. The in vitro reconstitution method the authors have developed only addresses RhoGDI/Rho GTPase dissociation and does not include membrane trafficking of Rho GTPases, which is also very important for delivering Rho GTPases such as Cdc42 to and taking them out of the plasma membrane. The authors should discuss the role of membrane trafficking.

Referee #2:

This is an important manuscript. Here the authors endeavour to analyse the time-evolution of RhoGTPase signaling, from membrane recruitment, activation to retention. These are open questions that have not yet been directly addressed by current models for Rho activation, that are based on GEF-GAP balances and often framed as steady-state rather than kinetic analyses. For their work the authors use state-of-the art reconstitution and single-molecule approaches, supported by elegant (if inevitably lower-resolution) in vivo studies. They convincingly show that Rho GTPases are activated by GEFs after they bind to the membrane, but the kinetics of membrane recruitment are not significantly altered by GEF activity. Surprisingly, they also find that

the conventional elements of Rho regulation (GEFs, GAPs and GDI) are not sufficient to generate the spatial fidelity of active Rho signaling. In other words, a significant pool of inactive/de-activated Rho remains at the membrane - which contrasts with the tight spatial coupling of Rho localization with its activated form that is often seen in vivo. Instead, spatial refinement is conferred by the engagement of effectors, which can increase dwell time (retention) of active Rho as well as reduce its diffusion out of the activation zone.

This work carries important implications that fundamentally alter how we think about Rho signaling. First, Rho effectors are not solely downstream mediators of signaling, but are integral determinants that control the spatiotemporal expression of signaling. While there have been precedents for this idea, this is the most elegant, reductionist demonstration of the concept that I am aware of. Second, it implies that the role of GDI is simpler than we conventionally think. As typically envisaged, RhoGTPases need to be released from GDI for activation. The authors data indicate that this is not essential - given a realistically large cytosolic pool of Rho-GDI complexes, basal rates of dissociation are sufficient to deliver Rho to the membrane for activation. The authors acknowledge that this does not exclude roles for modulation of Rho-GDI association/dissociation, but it means that the basal running of the system is sufficient for signaling.

Overall, these are important ideas supported by high-quality data. They have the potential to become textbook, and warrant publication to stimulate further studies in the field to test and extend this model. My comments are relatively minor: I see them as suggestions for the editors and authors, rather than deal-breakers.

General:

1. How might differences in affinity of effectors for GTP-RhoGTPases affect this model? High affinity binding might be predicted to decrease membrane off-rate more than lower-affinity interactions, unless this is offset by other factors such as effector clustering.

Tiny points:

1) Expanding on the notion of coupled spatial and catalytic cycles of RhoGTPases. This is a small point to help the general reader. The notion of catalytic cycles is clear enough from textbooks and reviews, but I think the authors could be more explicit about what they mean by spatial cycles and why they are important.

2) Fig 1G: Is this a single representative experiment or are there statistics here?

Referee #3:

The authors reconstitute in vitro the Rho GTPase cascade, and study the multi-step process by which Rho GTPases get activated. The Rho GTPase network involves dissociation of Rho GTPases from RhoGDI and their shuttling to the plasma membrane. Their GTP-loading status is then controlled by GEFs/GAPs. Active Rho GTPases then bind to effectors to actuate signaling. There has been much speculation about the dynamic set of interactions that allow for Rho GTPase activation. This problem is also difficult to study in cells. In the last decade, it got clear that the Rho GTPase network builds intracellular signaling patterns. The authors explore and discuss how the reconstituted Rho GTPase network and its dynamic interactions can contribute to such patterning. This is very timely. By a set of very elegant in vitro experiments, backed up by a classic in vivo model system for Rho GTPase patterning (frog oocyte wounding), the authors provide much needed insight about the hierarchical sequential interactions that lead to a Rho GTPase flux (activation and deactivation), that leads to patterning. Even if the reconstituted system remains (by definition) minimalistic, it captures a set of features that clarify how the Rho GTPase system works, that has been the subject of intense debate. It has to be noted that the reconstitution assays are state of the art - they include a self-organisation system that helps in dissecting patterning, as well as tackle out of equilibrium phenomena as seen in live cells. I believe this paper will become a classic, that can be cited when the Rho GTPase network is discussed. I think the paper is also very thoughtfully written. I do not have any major/minor revision request. This paper is ready to be published, and I congratulate the authors for their excellent work.

Point-to-point response to the reviewers' comments

We gratefully thank the reviewers for their positive feedback and insightful comments, which aided us to further improve the manuscript. Below is a point-by-point response to all comments and a detailed description of all changes we have made to our manuscript after considering their suggestions. The changes are highlighted in yellow in the revised text.

In addition to responding to the referee's comment, we had to make minor changes to section of our manuscript dealing with the single molecule analysis of Rho GTPase membrane recruitment. This was due to an unfortunate and inadvertent error during image acquisition for some of the data presented in Fig 3G-J and Fig 4E-J. We became aware of this issue during the process of re-submission. We therefore repeated all of these experiments, which lead to a minor change in numbers compared to the original version of the paper (see revised Fig 3G-J and Fig 4E-J), which we highlighted in the main text (see lines 272-274 and 305-314). Importantly, data from the revised experiments are still fully congruent with the conclusions of the original version.

Referee#1:

This manuscript builds on many years of impressive work from the Bement laboratory using small membrane wounds in *Xenopus* oocytes to understand how localized Rho GTPase signaling facilitates wound repair. In particular, it is based on their observation 20 years ago that active Cdc42 accumulates around the wound site, as well as more recently showing the ratio of total to active Cdc42. The authors now link this to the contribution of RhoGDI in delivering Cdc42 to the wound site, using a series of elegant in vitro biochemical experiments with LUVs before returning at the end of the manuscript to the wound model.

We appreciate the positive response and thank the reviewer for the constructive criticism that helped us to improve the manuscript.

[1.1] My main point for revision is that the authors should improve their statements on the conceptual novelty achieved through their development of the new in vitro reconstitution model for investigating the role of RhoGDI in Cdc42 activation, which was undoubtedly technically challenging and will be very useful to the community. The 'key findings' section and abstract should mention the development of the in vitro reconstitution methodology, and there are other appropriate areas in the text where they can emphasize the importance of its development in answering key questions relating to Cdc42.

Reply 1.1: We agree with the reviewer that the development of an in vitro reconstitution system is a valuable resource for the community, which deserves more emphasis in the manuscript. As requested, we now explicitly highlight this point in the 'key findings (first point)', 'abstract' (line 43), 'introduction' (line 122) and 'discussion' (line 507) section of the paper.

[1.2] Figure 1, S1: timing of accumulation of wGBD versus IT-Cdc42. wGBD could bind to other Rho GTPases in addition to Cdc42 (e.g. RhoQ), which might accumulate earlier than active Cdc42 at the wound site. Alternatively, the timing difference could simply reflect the relative sensitivity of detecting wGBD versus IT-Cdc42.

Reply 1.2: The reviewer suggests that the timing difference observed between mCh-wGBD and IT-eGFP-Cdc42 could reflect wGBD binding to other members of the Cdc42 subfamily in addition to Cdc42 and offers RhoQ (aka TC10) as a possibility. The concern is that if one of

these other potential wGBD targets is a) expressed; b) is abundant; and c) is accumulating more quickly than Cdc42 itself, this could mislead us into thinking that we are looking at active Cdc42 when in fact we are looking at some other active GTPase. However, while Cdc42 itself is abundantly expressed in *Xenopus laevis* oocytes (440 nM), neither of the other Cdc42 subfamily members -RhoJ and RhoQ (aka TC10)- are detectable at this stage of development (Wühr et al 2014 PMID: 24954049). Thus, we are confident that the signal provided by mCh-wGBD reflects active Cdc42 rather than some other GTPase.

The reviewer also suggests that the timing difference could reflect sensitivity differences in terms of detection such that mCh-wGBD is easier to detect than IT-GFP-Cdc42. However, this is unlikely in that eGFP is both brighter and more photostable than mCherry (PMID: 34862023). Moreover, the interaction of mCh-wGBD with active Cdc42 is necessarily transient while GFP is a permanent part of IT-GFP-Cdc42. Given that all of these conditions would make it easier to detect IT-GFP-Cdc42 than mCh-wGBD, we are confident that the delay seen between Cdc42 activation and recruitment does not reflect differences in sensitivity of the two probes.

As a further test of the conclusion that activation precedes accumulation, we conducted an additional experiment in which the accumulation of GFP-rGBD (a probe for active RhoA) was compared to the accumulation of IT-GFP-RhoA at wounds. The new data, which are shown in Fig. EV11 (also see line 172), demonstrate that like Cdc42, active RhoA accumulates around wounds ahead of total RhoA. Collectively, these new experiments and the above conclusions demonstrate that, just as we found *in vitro*, activation of Rho GTPases *in vivo* is followed by accumulation of the GTPases.

[1.3] Figure 2: It is not surprising that Cdc42 or Rac1 bound to RhoGDI inhibits GEF-induced activation, based on the structure of RhoGDI bound to these proteins and PMID: 12471028, among several papers, which demonstrates that RhoGDI needs to dissociate from Rac1 (or Cdc42/RhoA) to allow e.g. Tiam1 to act as GEF.

Reply 1.3: We agree with the reviewer that it is not surprising that GDIs inhibit GEFs and we note that we did not claim otherwise. However, we maintain that a clear biochemical sequence of events of GDI dissociation, Rho GTPase membrane binding and GEF-mediated activation was not established in literature. We would not have committed several years to this research if we had thought otherwise. The study mentioned by the reviewer (Robbe et al., 2003, which we cited in our previous version) shows that GTPase:GDI complexes and non-prenylated GTPases are poor substrates for GEFs in the presence of liposomes (compared to prenylated GTPases pre-bound to membranes, which have lost their association to GDI). These findings align with our reported reaction sequence, although our work builds significantly upon them.

To highlight this congruence, we have added a sentence to our results section of our bulk GEF data, which again cites the Robbe et al. paper (line 219). However, it is important to clarify that Robbe et al. does not provide conclusive evidence for a specific reaction order. Establishing this would require additional experiments like those we performed, including direct comparisons of GEF reaction rates to the kinetics of RhoGDI dissociation and efforts to co-localize GTPases and RhoGDI during membrane binding. The absence of this evidence led to alternative models, including some that were proposed long after Robbe et al. (e.g., Ugolev et al. 2005, PMID: 16702219; Ugolev et al., 2008 PMID: 18505730, Freisinger et al. 2013 PMID: 23651995), which suggested that GEFs and/or phospholipids actively dissociate GTPase complexes. Therefore, the sequence of biochemical steps has remained controversial until now, as also explicitly noted in seminal reviews focused on RhoGDI (see Garcia-Mata et al., 2011 PMID: 21779026).

[1.4] There is abundant evidence that RhoGDI dissociation from Rho GTPases requires phosphorylation and other post-translational mechanisms, which would then allow the Rho GTPases to insert into membranes. This is not explicitly tested in the in vitro LUV experiments and would significantly strengthen the message of the manuscript if it were tested. The authors downplay the importance of post-translational mechanisms of regulation in the Introduction and Discussion.

Reply 1.4: We are aware that multiple proteins, including but not limited to kinases, have been suggested to act as RhoGDI displacement factors (GDFs) and we probed multiple of these candidate GDFs during the course of this work. First, we tested for phosphorylation of RhoGDI1 by PAK1 kinase, which has been suggested to destabilize Rac1:RhoGDI1 complexes (DerMardirossian et al. 2004 PMID: 15225553). However, we could not detect kinase activity of either rat PAK1 for bovine RhoGDI1 or human PAK1 for human RhoGDI1 in the presence of active Cdc42 (Q61L) by in vitro kinase assays (see Referee Figure A-C below).

Figure for reviewers removed

Detection of phosphoproteins via gel stains (Pro-Q® Diamond, Invitrogen) only revealed a background signal for RhoGDI1 in the presence and absence of ATP. As a control we used the known PAK1 substrate Stathmin (Wittman et al 2004, 14645234), which showed a clear ATP-dependent increase in phosphostain signal in the presence of both human and rat PAK1, confirming activity of these kinases. The prominent background for RhoGDI by phosphostaining could not have resulted from pre-phosphorylation, because these proteins were expressed in E.Coli. Nevertheless, to obtain a more direct readout of phosphorylation, we turned to electrospray ionization mass spectrometry (ESI-MS). However, we found no apparent increase in the molecular weight of either bovine or human RhoGDI1 after incubation with rat or human PAK1, respectively, in the presence of ATP and active Cdc42 (Q61L) (see Referee Figure C). This was in contrast to Stathmin, which showed the expected ATP-dependent increase in mass by 80 or 160 Da (for single and dual phosphorylation) in response to incubation with PAK1. Based on unpublished observations in the *Xenopus* single cell wounding assays, we also tested for the phosphorylation of the *Xenopus* ortholog of RhoGDI1 (RhoGDI α 2) by PKC β kinase. However, we could only observe a background signal by phosphoprotein stain by kinase assays, which was not affected by the presence and absence of *Xenopus* PKC β , ATP and DAG-containing large unilamellar vesicles (LUVs). Therefore, we were unable to confirm the phosphorylation of RhoGDI by existing candidate kinases.

In addition, we tested for the proposed GDF activity by the membrane-actin crosslinking protein ezrin (Takahashi et al., 1997 PMID: 9287351). We were able to confirm that the FERM domain of human ezrin bound to supported lipid bilayers in a PI(4,5)P₂-dependent manner and that it was able to recruit RhoGDI1 to membranes (Referee Figure E). Importantly, however, we did not observe an acceleration of ITSN-mediated nucleotide exchange of Cdc42 in the presence of RhoGDI1 on membranes in response to the addition of ezrin(FERM) (Referee Figure F). This demonstrates that ezrin does not promote GDI dissociation and activation of Cdc42.

Beyond testing existing GDF candidates from literature including two kinases, we also pursued an unbiased approach and repeated our single molecule GEF experiments in the presence of an active cell lysate from porcine brain. These extracts are fully capable of supporting rapid Rho GTPase activation and downstream branched actin assembly on PI(4,5)P₂-containing membranes (Koronakis et al., 2011 PMID: 21844371; Ma et al., 1998 PMID: 9490725). The extracts should therefore likely contain such putative GDI displacement factor including kinases. However, we found that extracts supplemented with dual-labeled Cy3-Cdc42:A647-RhoGDI1 promoted neither recruitment of Cy3-Cdc42 nor the co-recruitment of A647-RhoGDI1. We introduced these results as a new appendix figure (Appendix Figure S3) in our manuscript and added text (line 319) to the results section to describe their outcome.

Despite all of these negative results, it was never our intention to downplay the possibility of post-translational modifications regulating GDI interactions. In fact, the discussion section of the previous version explicitly stated that our results "... should not be taken to mean that other factors such as GDI phosphorylation or GDFs play no role in vivo". We amended this statement by an additional sentence (line 542), which highlights and cites some of the original work suggesting a role for such mechanisms in vivo (DerMardirossian et al., 2006, 2004; Dovas et al., 2010; Takahashi et al., 1997; Yamashita and Tohyama, 2003).

[1.5] The authors should reference the extensive work on Cdc42 membrane localization and positive feedback loops in *S. cerevisiae* budding as well as molecular modeling on Rho GTPase/RhoGDI (e.g. reviewed by Garcia-Mata et al., Nat Rev Mol Cell Biol 2011: modeling by several groups, including Edelstein-Keshet, with whom some of the authors have collaborated). Studies in *S. cerevisiae* demonstrated clearly that activation prolongs retention of Cdc42 at the bud plasma membrane during bud extension in a positive feedback loop through a complex including the Cdc42 GEF Cdc24 and its target PAK, and that it is not activated by Cdc42 until delivered to the bud membrane in a mechanism involving yeast RhoGDI as well as vesicle trafficking (e.g. reviewed by Woods and Lew, Small GTPases 2019).

Reply 1.5: We agree with the reviewer and have added a paragraph to the discussion section addressing findings from yeast polarity (line 564). Previous work in budding and fission yeast has indeed suggested a preferential retention of active Cdc42 at membranes, though the underlying mechanisms remained largely unresolved. Here, we provide a biochemically founded explanation of this effect. Additionally, we note that most of these past studies used N-terminal fluorescent protein (FP) fusions, which are not functional in rescuing Cdc42 deletion. We also agree with the reviewer that it was established that Cdc42 is activated only upon reaching the membrane. However, the precise sequence of events at membrane arrival were not clear, as discussed in reply 1.3 above.

[1.6] The in vitro reconstitution method the authors have developed only addresses RhoGDI/Rho GTPase dissociation and does not include membrane trafficking of Rho GTPases, which is also very important for delivering Rho GTPases such as Cdc42 to and taking them out of the plasma membrane. The authors should discuss the role of membrane trafficking.

Reply 1.6: The reviewer alludes to studies that have previously implicated membrane trafficking in the delivery and retrieval of Cdc42 to and from the plasma membrane, respectively and asks that we discuss the role of such trafficking. We certainly agree that membrane trafficking occurs during cell repair, in part because we have documented both endo- and exocytosis around oocyte wounds (Davenport et al., 2016 PMID:27226483). Unfortunately, it is not clear to us how bringing this topic up will strengthen the current study. While there are older studies of budding yeast that led to the proposal that accumulation of Cdc42 at the nascent bud site could arise as a result of transport of Cdc42-associated vesicles to the bud site by cables of F-actin (Wedlich-Soldner et al., 2003 and 2005 PMID: 12560471 and 15353546), subsequent work challenged this claim on a variety of grounds (Layton et al. PMID: 21277209; Woods et al., PMID: 27476596; Watson et al., PMID: 25158298). As the yeast model is far and away the one in which this subject has been most thoroughly studied, and as the results are, at best, controversial, it would only make sense to weigh in on this subject if we had something to add. But we do not. The reconstitution results do not involve membrane trafficking in any way, and nothing was done in the oocyte wound experiments in this paper to monitor the effects of manipulations of membrane trafficking on accumulation of total and active Cdc42.

Referee #2:

This is an important manuscript. Here the authors endeavour to analyse the time-evolution of RhoGTPase signaling, from membrane recruitment, activation to retention. These are open questions that have not yet been directly addressed by current models for Rho activation, that are based on GEF-GAP balances and often framed as steady-state rather than kinetic analyses. For their work the authors use state-of-the art reconstitution and single-molecule approaches, supported by elegant (if inevitably lower-resolution) *in vivo* studies. They convincingly show that Rho GTPases are activated by GEFs after they bind to the membrane, but the kinetics of membrane recruitment are not significantly altered by GEF activity. Surprisingly, they also find that the conventional elements of Rho regulation (GEFs, GAPs and GDI) are not sufficient to generate the spatial fidelity of active Rho signaling. In other words, a significant pool of inactive/de-activated Rho remains at the membrane - which contrasts with the tight spatial coupling of Rho localization with its activated form that is often seen *in vivo*. Instead, spatial refinement is conferred by the engagement of effectors, which can increase dwell time (retention) of active Rho as well as reduce its diffusion out of the activation zone.

This work carries important implications that fundamentally alter how we think about Rho signaling. First, Rho effectors are not solely downstream mediators of signaling, but are integral determinants that control the spatiotemporal expression of signaling. While there have been precedents for this idea, this is the most elegant, reductionist demonstration of the concept that I am aware of. Second, it implies that the role of GDI is simpler than we conventionally think. As typically envisaged, RhoGTPases need to be released from GDI for activation. The authors data indicate that this is not essential - given a realistically large cytosolic pool of Rho-GDI complexes, basal rates of dissociation are sufficient to deliver Rho to the membrane for activation. The authors acknowledge that this does not exclude roles for modulation of Rho-GDI association/dissociation, but it means that the basal running of the system is sufficient for signaling.

Overall, these are important ideas supported by high-quality data. They have the potential to become textbook, and warrant publication to stimulate further studies in the field to test and extend this model. My comments are relatively minor: I see them as suggestions for the editors and authors, rather than deal-breakers.

We sincerely thank the reviewer for the favorable feedback and thoughtful comments. We addressed these points via textual revisions, which improved the accessibility and clarity of our manuscript.

General:

[2.1] How might differences in affinity of effectors for GTP-RhoGTPases affect this model? High affinity binding might be predicted to decrease membrane off-rate more than lower-affinity interactions, unless this is offset by other factors such as effector clustering.

Reply 2.1: We fully agree with the reviewer that this important question deserves attention and now address this point in the penultimate paragraph of the discussion (line 613). Affinity for GTPases is known to vary considerably among different effectors (Bishop and Hall, 2000 PMID: 10816416). However, this is likely not the only determinant, since effector affinity for other membrane components can also be expected to influence the degree of membrane retention. Generally, increased affinity should decrease membrane off-rate more significantly,

depending on which interaction is kinetically limiting. Exploring this issue experimentally offers an exciting opportunity for future research.

Tiny points:

1) Expanding on the notion of coupled spatial and catalytic cycles of RhoGTPases. This is a small point to help the general reader. The notion of catalytic cycles is clear enough from textbooks and reviews, but I think the authors could be more explicit about what they mean by spatial cycles and why they are important.

Reply 2.2: We appreciate this point and have revised the introduction to better explain the concept of the spatial cycle to the general reader.

2) Fig 1G: Is this a single representative experiment or are there statistics here?

Reply 2.3: We apologize for the lack of clarity in data presentation. This particular data stems from nine individual cell wounds and the plot shows the mean with standard deviation as an error indicator. The figure legend has been revised to convey this information accurately.

Referee #3:

The authors reconstitute in vitro the Rho GTPase cascade, and study the multi-step process by which Rho GTPases get activated. The Rho GTPase network involves dissociation of Rho GTPases from RhoGDI and their shuttling to the plasma membrane. Their GTP-loading status is then controlled by GEFs/GAPs. Active Rho GTPases then bind to effectors to actuate signaling. There has been much speculation about the dynamic set of interactions that allow for Rho GTPase activation. This problem is also difficult to study in cells. In the last decade, it got clear that the Rho GTPase network builds intracellular signaling patterns. The authors explore and discuss how the reconstituted Rho GTPase network and its dynamic interactions can contribute to such patterning. This is very timely. By a set of very elegant in vitro experiments, backed up by a classic in vivo model system for Rho GTPase patterning (frog oocyte wounding), the authors provide much needed insight about the hierarchical sequential interactions that lead to a Rho GTPase flux (activation and deactivation), that leads to patterning. Even if the reconstituted system remains (by definition) minimalistic, it captures a set of features that clarify how the Rho GTPase system works, that has been the subject of intense debate. It has to be noted that the reconstitution assays are state of the art - they include a self-organisation system that helps in dissecting patterning, as well as tackle out of equilibrium phenomena as seen in live cells. I believe this paper will become a classic, that can be cited when the Rho GTPase network is discussed. I think the paper is also very thoughtfully written. I do not have any major/minor revision request. This paper is ready to be published, and I congratulate the authors for their excellent work.

We greatly appreciate the reviewer's enthusiastic reception of our work and the manuscript.

Dear Peter and Bill,

Thank you for submitting a revised version of your manuscript. We have now received input from one of the original reviewers, who finds that their main concerns have been addressed satisfactorily and recommends acceptance of the manuscript.

Additionally, there remain a few editorial points that need addressing before I can extend official acceptance of the manuscript:

1. Please remove the Reagents and Tools Table from the manuscript text file and upload it as a separate file choosing the file type "Reagent Table" and using the template available in our author guidelines:
<https://www.embopress.org/page/journal/14602075/authorguide#structuredmethods>.
2. Please move the Data Availability Section to the end of Materials and Methods.
3. Please update references according to The EMBO Journal style - where there are more than 10 authors on a paper, the first 10 should be listed, followed by 'et al.' Please see further information here:
<https://www.embopress.org/page/journal/14602075/authorguide#referencesformat>
4. Please remove DOIs from references to already published articles. According to The EMBO Journal style, DOIs should be used only for preprints and datasets.
5. Please remove EV Movie legends from the manuscript text file and zip together with each movie file. Further information is available here: <https://www.embopress.org/page/journal/14602075/authorguide#expandedview>
6. Figure panel 3H is not mentioned in the manuscript text - please add the corresponding callout.
7. In the Appendix, please add the manuscript title in the front page and include page numbers in the table of contents.
8. In our standard image integrity check, we noted that the following figure panels are reused in the manuscript:
 - between Figure 3G and top row of Appendix Fig S2B;
 - between Figure 5J and Appendix Fig S5A;
 - between Figure 6B, Figure EV4D And Appendix Fig. S5A.If this was intentional, please mention the reuse in the figure legend.
9. Our data editors have flagged the following issues in figure legends that need correcting:
 - Please provide the exact p values in the legends of figures 1F, 8K, EV1 E, EV5 B-D.
 - Please indicate the statistical test used for data analysis in the legends of figures EV3 D, F.
 - Please provide information on the number and nature of replicates in the legends of figures EV1 F, EV2 C-E.
 - Please describe the nature of replicates (e.g., technical or biological) in the legends of figures 1F, 3B, EV3 F.
 - Please define the error bars in the legends of figures 1F, 8K, S; EV2 C-E.
 - Please define the measure of center for the error bars in the legends of figures 4J; EV3 F.
 - Please define the scale bar for figures 8B, E, H, M, N, P, Q.
10. Papers published in The EMBO Journal are accompanied online by a 'Synopsis' to enhance discoverability of the manuscript. It consists of A) a short (1-2 sentences) summary of the findings and their significance, B) 3-4 bullet points highlighting key results and C) a synopsis image that is 550x300-600 pixels large (width x height, jpeg or png format). You can either show a model or key data in the synopsis image. Please note that the image size is rather small and that text needs to be readable at the final size.
11. The "Key Findings" section should be removed from the manuscript text and can be uploaded as bullet points for the synopsis text.

With best wishes,

Ieva

We realize that it is difficult to revise to a specific deadline. In the interest of protecting the conceptual advance provided by the work, we recommend a revision within 3 months (15th May 2025). Please discuss the revision progress ahead of this time with the editor if you require more time to complete the revisions.

Referee #1:

I thank the authors for providing clear and well thought through replies to my comments, including changes to the text and figures. I am happy with all the responses and changes and do not require any further revisions. There are just a few minor typos in the revised text that the authors and editors will pick up.

All editorial and formatting issues were resolved by the authors.

Dear Peter,

Thank you for implementing the final formatting changes in the manuscript. I am now pleased to inform you that your manuscript has been accepted for publication in the EMBO Journal. Congratulations on a great study!

Before we forward your manuscript to our publishers, I would like to propose some edits in the manuscript abstract and synopsis (please see below and in the attached file). I have also written a short blurb that will accompany the title of your manuscript in our online table of contents. Furthermore, we would like to propose to make the title more explicit and informative, as it is rather general and seems more suitable for a review article. I have included a proposal below, but I will be happy to discuss other versions you might prefer. Please take a look and let me know if any corrections are needed.

New title:

Effector proteins mediate Rho GTPase retention at the plasma membrane for activity pattern formation

Blurb:

A biochemically defined in vitro reconstitution approach reveals mechanistic principles of Rho GTPase patterning at the plasma membrane and its stage-dependent regulators.

Synopsis:

Formation of Rho GTPase activity patterns on the plasma membrane is key to numerous morphogenic processes in eukaryotic cells. This study employs a reconstitution approach to reveal the biochemical principles of Rho GTPase pattern formation and show that these signaling proteins enrich at locations where they are activated by effector-mediated membrane retention.

- A biochemically defined reconstitution system allows investigation of Rho GTPase patterning dynamics in the presence of all major GTPase regulators.
- Rho GTPase membrane binding is limited by dissociation from RhoGDI.
- Activation of Rho GTPases occurs after membrane binding and GEFs do not recruit Rho GTPases to membranes.
- Activation prolongs retention on the membrane by effector-mediated stabilization.

If you have any questions, please do not hesitate to contact the Editorial Office. Thank you for this contribution to The EMBO Journal!

With best wishes,

Ieva
